# Methodological quality of COVID-19 clinical research

Richard G. Jung [1,2,3,13], Pietro Di Santo[1,2,4,5,13], Cole Clifford[6], Graeme Prosperi-Porta[7], Stephanie Skanes[6], Annie Hung[8], Simon Parlow[4], Sarah Visintini [9], F. Daniel Ramirez [1,4,10,11], Trevor Simard[1,2,3,4,12] & Benjamin Hibbert [2,3,4✉]

The COVID-19 pandemic began in early 2020 with major health consequences. While a need to disseminate information to the medical community and general public was paramount, concerns have been raised regarding the scientific rigor in published reports. We performed a systematic review to evaluate the methodological quality of currently available COVID-19 studies compared to historical controls. A total of 9895 titles and abstracts were screened and 686 COVID-19 articles were included in the final analysis. Comparative analysis of COVID-19 to historical articles reveals a shorter time to acceptance (13.0[IQR, 5.0–25.0] days vs. 110.0[IQR, 71.0–156.0] days in COVID-19 and control articles, respectively; $p < 0.0001$). Furthermore, methodological quality scores are lower in COVID-19 articles across all study designs. COVID-19 clinical studies have a shorter time to publication and have lower methodological quality scores than control studies in the same journal. These studies should be revisited with the emergence of stronger evidence.

[1] CAPITAL Research Group, University of Ottawa Heart Institute, Ottawa, Ontario, Canada. [2] Vascular Biology and Experimental Medicine Laboratory, University of Ottawa Heart Institute, Ottawa, Ontario, Canada. [3] Department of Cellular and Molecular Medicine, Faculty of Medicine, University of Ottawa, Ottawa, Ontario, Canada. [4] Division of Cardiology, University of Ottawa Heart Institute, Ottawa, Ontario, Canada. [5] School of Epidemiology and Public Health, University of Ottawa, Ottawa, Ontario, Canada. [6] Faculty of Medicine, University of Ottawa, Ontario, Canada. [7] Department of Medicine, Cumming School of Medicine, Calgary, Alberta, Canada. [8] Division of Internal Medicine, The Ottawa Hospital, Ottawa, Ontario, Canada. [9] Berkman Library, University of Ottawa Heart Institute, Ottawa, Ontario, Canada. [10] Hôpital Cardiologique du Haut-Lévêque, CHU Bordeaux, Bordeaux-Pessac, France. [11] L'Institut de Rythmologie et Modélisation Cardiaque (LIRYC), University of Bordeaux, Bordeaux, France. [12] Department of Cardiovascular Medicine, Mayo Clinic, Rochester, MN, USA. [13] These authors contributed equally: Richard G. Jung, Pietro Di Santo. ✉email: bhibbert@ottawaheart.ca

The severe acute respiratory syndrome coronavirus 2 (SARS-CoV-2) pandemic spread globally in early 2020 with substantial health and economic consequences. This was associated with an exponential increase in scientific publications related to the coronavirus disease 2019 (COVID-19) in order to rapidly elucidate the natural history and identify diagnostic and therapeutic tools[1].

While a need to rapidly disseminate information to the medical community, governmental agencies, and general public was paramount—major concerns have been raised regarding the scientific rigor in the literature[2]. Poorly conducted studies may originate from failure at any of the four consecutive research stages: (1) choice of research question relevant to patient care, (2) quality of research design[3], (3) adequacy of publication, and (4) quality of research reports. Furthermore, evidence-based medicine relies on a hierarchy of evidence, ranging from the highest level of randomized controlled trials (RCT) to the lowest level of case series and case reports[4].

Given the implications for clinical care, policy decision making, and concerns regarding methodological and peer-review standards for COVID-19 research[5], we performed a formal evaluation of the methodological quality of published COVID-19 literature. Specifically, we undertook a systematic review to identify COVID-19 clinical literature and matched them to historical controls to formally evaluate the following: (1) the methodological quality of COVID-19 studies using established quality tools and checklists, (2) the methodological quality of COVID-19 studies, stratified by median time to acceptance, geographical regions, and journal impact factor and (3) a comparison of COVID-19 methodological quality to matched controls.

Herein, we show that COVID-19 articles are associated with lower methodological quality scores. Moreover, in a matched cohort analysis with control articles from the same journal, we reveal that COVID-19 articles are associated with lower quality scores and shorter time from submission to acceptance. Ultimately, COVID-19 clinical studies should be revisited with the emergence of stronger evidence.

## Results

**Article selection**. A total of 14787 COVID-19 papers were identified as of May 14, 2020 and 4892 duplicate articles were removed. In total, 9895 titles and abstracts were screened, and 9101 articles were excluded due to the study being pre-clinical in nature, case report, case series <5 patients, in a language other than English, reviews (including systematic reviews), study protocols or methods, and other coronavirus variants with an overall inter-rater study inclusion agreement of 96.7% ($\kappa = 0.81$; 95% CI, 0.79–0.83). A total number of 794 full texts were reviewed for eligibility. Over 108 articles were excluded for ineligible study design or publication type (such as letter to the editors, editorials, case reports or case series <5 patients), wrong patient population, non-English language, duplicate articles, wrong outcomes and publication in a non-peer-reviewed journal. Ultimately, 686

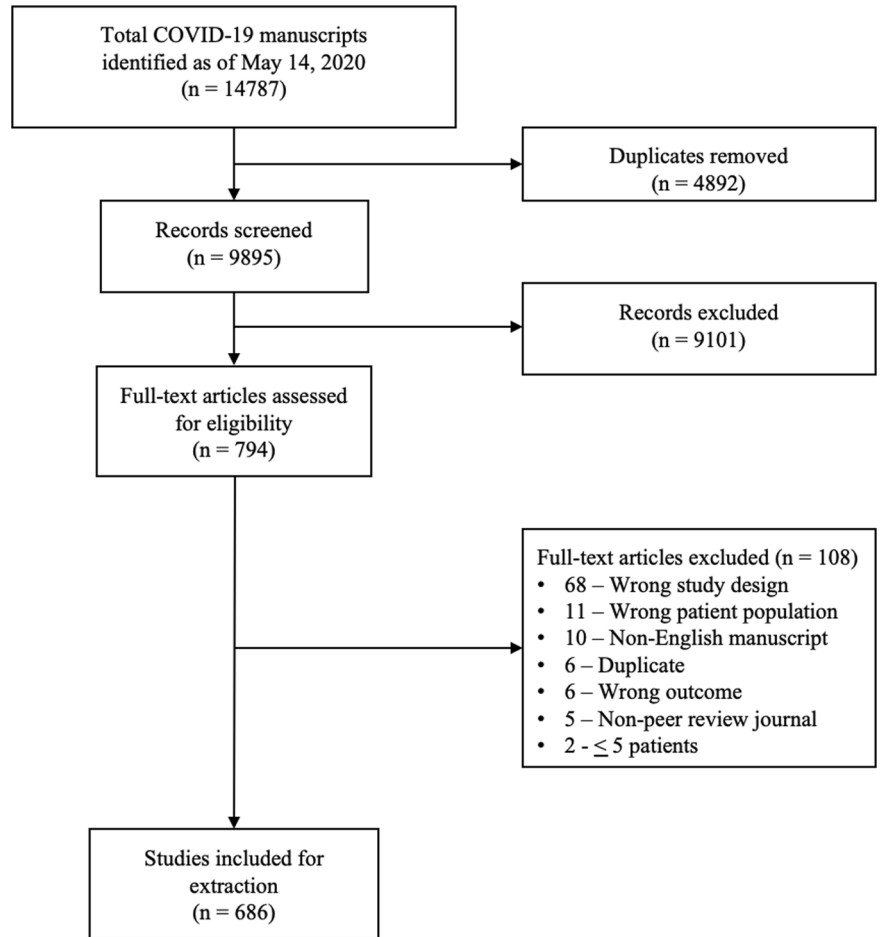

**Fig. 1 Literature search and selection of COVID-19 articles.** A total of 14787 articles were identified and 4892 duplicate articles were removed. Overall, 9895 articles were screened by title and abstract leaving 794 articles for full-text screening. Over 108 articles were excluded, leaving a total of 686 articles that underwent methodological quality assessment.

**Table 1 Characteristics of COVID-19 clinical literature until May 14, 2020.**

| | COVID-19 articles (n = 686) | Matched articles (n = 1078) | | P-value* |
| | | COVID-19 (n = 539) | Control (n = 539) | |
| --- | --- | --- | --- | --- |
| | N (%) | N (%) | N (%) | |
| *Study design* | | | | |
| Case series | 380 (55.4) | 277 (51.4) | 277 (51.4) | |
| Cohort | 199 (29.0) | 174 (32.3) | 174 (32.3) | |
| Case control | 38 (5.5) | 32 (5.9) | 32 (5.9) | |
| Diagnostic | 63 (9.2) | 53 (9.8) | 53 (9.8) | |
| Randomized controlled trial | 6 (0.9) | 3 (0.6) | 3 (0.6) | |
| *Geographic region* | | | | <0.0001 |
| Asia/Oceania | 469 (68.4) | 377 (69.9) | 167 (31.0) | |
| Europe | 139 (20.3) | 99 (18.4) | 176 (32.7) | |
| Americas | 78 (11.4) | 63 (11.7) | 196 (36.4) | |
| *Type of article* | | | | 0.91 |
| Original research | 614 (89.5) | 486 (90.2) | 484 (89.8) | |
| Letter | 35 (5.1) | 26 (4.8) | 29 (5.4) | |
| Communication | 37 (5.4) | 27 (5.0) | 26 (4.8) | |
| Retrospective study | 590 (86.0) | 459 (85.2) | 386 (71.6) | <0.0001 |
| Sex reported | 620 (90.4) | 484 (89.8) | 483 (89.6) | 0.92 |
| Sample size calculated without case series | (n = 306) | (n = 262) | (n = 262) | |
| | 7 (2.3) | 4 (1.5) | 30 (11.5) | <0.0001 |
| *Method of SARS-CoV-2 diagnosis* | | | | |
| PCR | 548 (79.9) | 437 (81.1) | — | — |
| ELISA | 2 (0.3) | 1 (0.2) | — | — |
| Physical exam only | 5 (0.7) | 4 (0.7) | — | — |
| Multiple | 3 (0.4) | 2 (0.4) | — | — |
| Unknown | 128 (18.7) | 95 (17.6) | — | — |
| *Ethics approval* | | | | 0.48 |
| Received | 556 (81.0) | 433 (80.3) | 451 (83.7) | |
| Not required/received | 91 (13.3) | 73 (13.5) | 60 (11.1) | |
| Not mentioned | 39 (5.7) | 33 (6.1) | 28 (4.1) | |
| Impact factor—median (IQR) (n = 652) | 4.7 (2.9–7.6) | — | — | — |
| Time to acceptance—median (IQR) (n = 450) | 13.0 (5.0–25.0) | 13.0 (5.0–25.0) | 110.0 (71.0–156.0) | <0.0001 |

COVID-19 articles were 1:1 matched with control articles from the same journal (identical impact factor) and study design. *PCR*: polymerase chain reaction; *ELISA*: enzyme-linked immunosorbent assay.
*Statistical analysis was conducted to compare COVID-19 to historical control articles with their corresponding *P*-values.

articles were identified with an inter-rater agreement of 86.5% (κ = 0.68; 95% CI, 0.67–0.70) (Fig. 1).

**COVID-19 literature methodological quality**. Most studies originated from Asia/Oceania with 469 (68.4%) studies followed by Europe with 139 (20.3%) studies, and the Americas with 78 (11.4%) studies. Of included studies, 380 (55.4%) were case series, 199 (29.0%) were cohort, 63 (9.2%) were diagnostic, 38 (5.5%) were case–control, and 6 (0.9%) were RCTs. Most studies (590, 86.0%) were retrospective in nature, 620 (90.4%) reported the sex of patients, and 7 (2.3%) studies excluding case series calculated their sample size a priori. The method of SARS-CoV-2 diagnosis was reported in 558 studies (81.3%) and ethics approval was obtained in 556 studies (81.0%). Finally, journal impact factor of COVID-19 manuscripts was 4.7 (IQR, 2.9–7.6) with a time to acceptance of 13.0 (IQR, 5.0–25.0) days (Table 1).

Overall, when COVID-19 articles were stratified by study design, a mean case series score (out of 5) (SD) of 3.3 (1.1), mean NOS cohort study score (out of 8) of 5.8 (1.5), mean NOS case–control study score (out of 8) of 5.5 (1.9), and low bias present in 4 (6.4%) diagnostic studies was observed (Table 2 and Fig. 2). Furthermore, in the 6 RCTs in the COVID-19 literature, there was a high risk of bias with little consideration for sequence generation, allocation concealment, blinding, incomplete outcome data, and selective outcome reporting (Table 2).

For secondary outcomes, rapid time from submission to acceptance (stratified by median time of acceptance of <13.0 days)

was associated with lower methodological quality scores for case series and cohort study designs but not for case–control nor diagnostic studies (Fig. 3A–D). Low journal impact factor (<10) was associated with lower methodological quality scores for case series, cohort, and case–control designs (Fig. 3E–H). Finally, studies originating from different geographical regions had no differences in methodological quality scores with the exception of cohort studies (Fig. 3I–L). When dichotomized by high vs. low methodological quality scores, a similar trend was observed with rapid time from submission to acceptance (34.4% vs. 46.3%, p = 0.01, Supplementary Fig. 1B), low impact factor journals (<10) was associated with lower methodological quality score (38.8% vs. 68.0%, p < 0.0001, Supplementary Fig. 1C). Finally, studies originating in either Americas or Asia/Oceania was associated with higher methodological quality scores than Europe (Supplementary Fig. 1D).

**Methodological quality score differences in COVID-19 versus historical control**. We matched 539 historical control articles to COVID-19 articles from the same journal with identical study designs in the previous year for a final analysis of 1078 articles (Table 1). Overall, 554 (51.4%) case series, 348 (32.3%) cohort, 64 (5.9%) case–control, 106 (9.8%) diagnostic and 6 (0.6%) RCTs were identified from the 1078 total articles. Differences exist between COVID-19 and historical control articles in geographical region of publication, retrospective study design, and sample size calculation (Table 1). Time of acceptance was 13.0

**Table 2 Methodological quality scores of different study types stratified by COVID-19 and historical control articles.**

|  | All COVID-19 (n = 686) | COVID-19 (n = 539) | Control (n = 539) | P-value* |
|---|---|---|---|---|
| Case series | (n = 380) | (n = 277) | (n = 277) | |
| Overall score—mean (SD) | 3.3 (1.1) | 3.3 (1.1) | 4.3 (0.8) | <0.0001 |
| Selection | 172 (45%) | 115 (42%) | 171 (62%) | |
| Ascertainment of exposure | 284 (75%) | 210 (76%) | 258 (93%) | |
| Ascertainment of outcome | 304 (80%) | 224 (81%) | 258 (93%) | |
| Outcome—was follow-up long enough | 215 (57%) | 159 (57%) | 268 (97%) | |
| Reporting—sufficient detail provided | 289 (76%) | 210 (76%) | 235 (85%) | |
| Case-control study | (n = 38) | (n = 32) | (n = 32) | |
| Overall score—mean (SD) | 5.5 (1.9) | 5.4 (1.9) | 6.6 (1.0) | 0.0027 |
| Selection—case definition | 22 (58%) | 16 (50%) | 21 (66%) | |
| Selection—representativeness of cases | 26 (68%) | 22 (69%) | 30 (94%) | |
| Selection—controls | 20 (53%) | 16 (50%) | 19 (59%) | |
| Selection—definition of controls | 25 (66%) | 20 (63%) | 29 (91%) | |
| Comparability | 25 (66%) | 22 (69%) | 26 (81%) | |
| Exposure—ascertainment | 31 (82%) | 26 (81%) | 29 (91%) | |
| Exposure—same method of ascertainment between cases and controls | 33 (87%) | 28 (88%) | 31 (97%) | |
| Exposure—non-response rate | 27 (71%) | 22 (69%) | 26 (81%) | |
| Cohort study | (n = 199) | (n = 174) | (n = 174) | |
| Overall score—mean (SD) | 5.8 (1.5) | 5.8 (1.6) | 7.1 (1.0) | <0.0001 |
| Selection—representativeness of cases | 145 (73%) | 126 (72%) | 152 (87%) | |
| Selection—non-exposed cohort | 161 (81%) | 140 (81%) | 165 (95%) | |
| Selection—ascertainment of exposure | 159 (80%) | 138 (79%) | 161 (93%) | |
| Selection—outcome not present at the beginning | 106 (53%) | 92 (53%) | 131 (75%) | |
| Comparability | 137 (69%) | 120 (69%) | 140 (81%) | |
| Outcome—ascertainment | 177 (89%) | 155 (89%) | 162 (93%) | |
| Outcome—was follow-up long enough | 113 (57%) | 98 (56%) | 161 (93%) | |
| Outcome—adequacy of follow-up of cohort | 163 (82%) | 142 (82%) | 155 (89%) | |
| Diagnostic study | (n = 63) | (n = 53) | (n = 53) | |
| *Risk of bias—low bias* | | | | |
| Patient selection | 13 (21%) | 11 (21%) | 23 (43%) | |
| Index test | 13 (21%) | 9 (17%) | 31 (59%) | |
| Reference standard | 46 (73%) | 40 (76%) | 39 (74%) | |
| Flow and timing | 41 (65%) | 36 (68%) | 41 (77%) | |
| Overall (low bias) | 4 (6%) | 12 (23%) | 24 (45%) | 0.02 |
| *Applicability concerns—low bias* | | | | |
| Patient selection | 56 (89%) | 49 (93%) | 51 (96%) | |
| Index test | 60 (95%) | 50 (94%) | 51 (96%) | |
| Reference standard | 57 (91%) | 47 (89%) | 53 (100%) | |
| Overall (low bias) | 52 (83%) | 45 (85%) | 50 (94%) | 0.11 |
| Randomized controlled trial | (n = 6) | (n = 3) | (n = 3) | |
| Sequence generation risk of bias | 3 (50%) | 2 (67%) | 2 (67%) | |
| Allocation concealment | 2 (33%) | 1 (33%) | 3 (100%) | |
| Blinding of participants and personnel to all outcomes | 1 (17%) | 1 (33%) | 2 (67%) | |
| Blinding outcome assessors for all outcomes | 1 (17%) | 1 (33%) | 3 (100%) | |
| Incomplete outcome data for all outcomes | 3 (50%) | 2 (67%) | 2 (67%) | |
| Selective outcome reporting | 5 (83%) | 3 (100%) | 3 (100%) | |
| Other sources of bias | 2 (33%) | 1 (33%) | 2 (67%) | |

*Statistical analysis was conducted to compare COVID-19 to historical control articles with their corresponding P-values.

(IQR, 5.0–25.0) days in COVID-19 articles vs. 110.0 (IQR, 71.0–156.0) days in control articles (Table 1 and Fig. 4A, p < 0.0001). Case-series methodological quality score was lower in COVID-19 articles compared to the historical control (3.3 (1.1) vs. 4.3 (0.8); n = 554; p < 0.0001; Table 2 and Fig. 4B). Furthermore, NOS score was lower in COVID-19 cohort studies (5.8 (1.6) vs. 7.1 (1.0); n = 348; p < 0.0001; Table 2 and Fig. 4C) and case–control studies (5.4 (1.9) vs. 6.6 (1.0); n = 64; p = 0.003; Table 2 and Fig. 4D). Finally, lower risk of bias in diagnostic studies was in 12 COVID-19 articles (23%; n = 53) compared to 24 control articles (45%; n = 53; p = 0.02; Table 2 and Fig. 4E). A similar trend was observed between COVID-19 and historical control articles when dichotomized by good vs. low methodological quality scores (Supplementary Fig. 2).

## Discussion

In this systematic evaluation of methodological quality, COVID-19 clinical research was primarily observational in nature with modest methodological quality scores. Not only were the study designs low in the hierarchy of scientific evidence, we found that COVID-19 articles were associated with a lower methodological quality scores when published with a shorter time of publication and in lower impact factor journals. Furthermore, in a matched cohort analysis with historical control articles identified from the same journal of the same study design, we demonstrated that COVID-19 articles were associated with lower quality scores and shorter time from submission to acceptance.

The present study demonstrates comparative differences in methodological quality scores between COVID-19 literature and

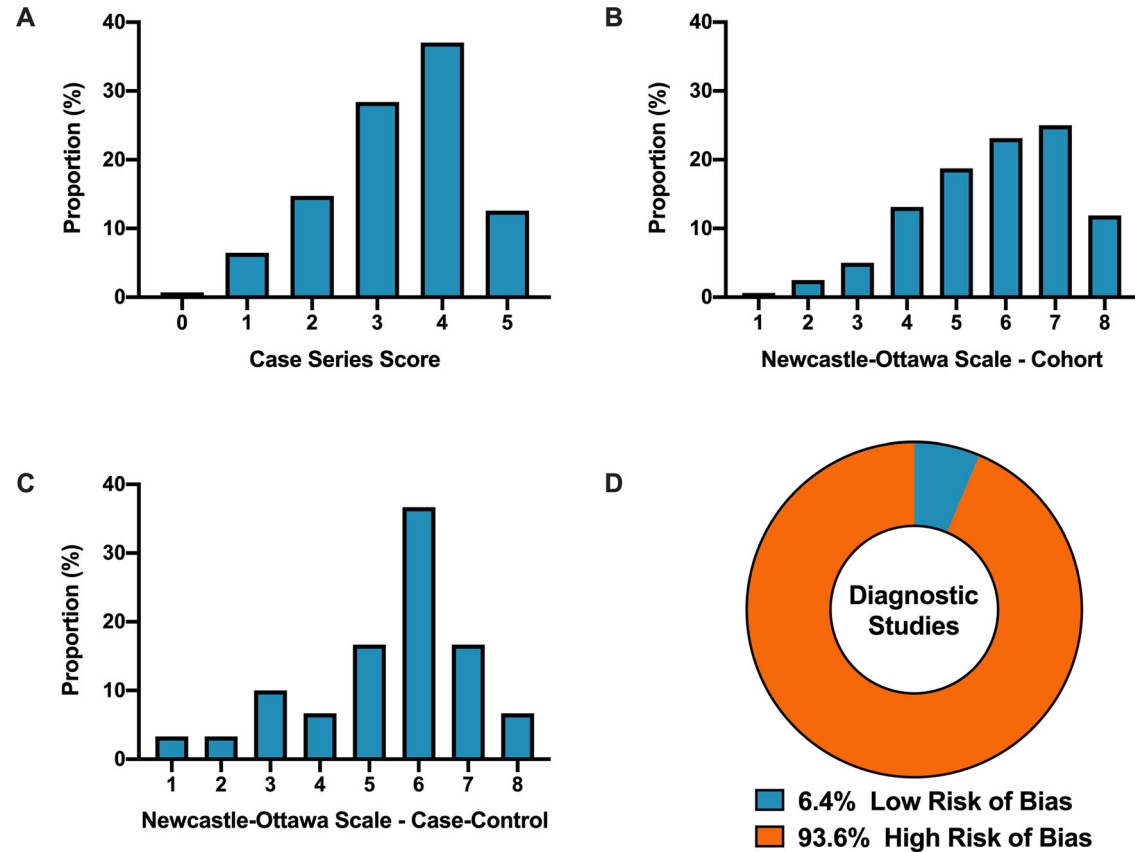

**Fig. 2 COVID-19 clinical literature quality assessment. A** Distribution of COVID-19 case series studies scored using the Murad tool ($n = 380$). **B** Distribution of COVID-19 cohort studies scored using the Newcastle–Ottawa Scale ($n = 199$). **C** Distribution of COVID-19 case–control studies scored using the Newcastle–Ottawa Scale ($n = 38$). **D** Distribution of COVID-19 diagnostic studies scored using the QUADAS-2 tool ($n = 63$). In panel **D**, blue represents low risk of bias and orange represents high risk of bias.

historical control articles. Overall, the accelerated publication of COVID-19 research was associated with lower study quality scores compared to previously published historical control studies. Our research highlights major differences in study quality between COVID-19 and control articles, possibly driven in part by a combination of more thorough editorial and/or peer-review process as suggested by the time to publication, and robust study design with questions which are pertinent for clinicians and patient management[3,6–11].

In the early stages of the COVID-19 pandemic, we speculate that an urgent need for scientific data to inform clinical, social and economic decisions led to shorter time to publication and explosion in publication of COVID-19 studies in both traditional peer-reviewed journals and preprint servers[1,12]. The accelerated scientific process in the COVID-19 pandemic allowed a rapid understanding of natural history of COVID-19 symptomology and prognosis, identification of tools including RT-PCR to diagnose SARS-CoV-2[13], and identification of potential therapeutic options such as tocilizumab and convalescent plasma which laid the foundation for future RCTs[14–16]. A delay in publication of COVID-19 articles due to a slower peer-review process may potentially delay dissemination of pertinent information against the pandemic. Despite concerns of slow peer review, major landmark trials (i.e. RECOVERY and ACTT-1 trial)[17,18] published their findings in preprint servers and media releases to allow for rapid dissemination. Importantly, the data obtained in these initial studies should be revisited as stronger data emerges as lower quality studies may fundamentally risk patient safety, resource allocation and future scientific research[19].

Unfortunately, poor evidence begets poor clinical decisions[20]. Furthermore, lower quality scientific evidence potentially undermines the public's trust in science during this time and has been evident through misleading information and high-profile retractions[12,21–23]. For example, the benefits of hydroxychloroquine, which were touted early in the pandemic based on limited data, have subsequently failed to be replicated in multiple observational studies and RCTs[5,24–30]. One poorly designed study combined with rapid publication led to considerable investment of both the scientific and medical community—akin to quinine being sold to the public as a miracle drug during the 1918 Spanish Influenza[31,32]. Moreover, as of June 30, 2020, ClinicalTrials.gov listed an astonishing 230 COVID-19 trials with hydroxychloroquine/plaquenil, and a recent living systematic review of observational studies and RCTs of hydroxychloroquine or chloroquine for COVID-19 demonstrated no evidence of benefit nor harm with concerns of severe methodological flaws in the included studies[33].

Our study has important limitations. We evaluated the methodological quality of existing studies using established checklists and tools. While it is tempting to associate methodological quality scores with reproducibility or causal inferences of the intervention, it is not possible to ascertain the impact on the study design and conduct of research nor results or conclusions in the identified reports[34]. Second, although the methodological quality scales and checklists used for the manuscript are commonly used for quality assessment in systematic reviews and meta-analyses[35–38], they can only assess the methodology without consideration for causal language and are prone to limitations[39,40]. Other tools such as the ROBINS-I and GRADE

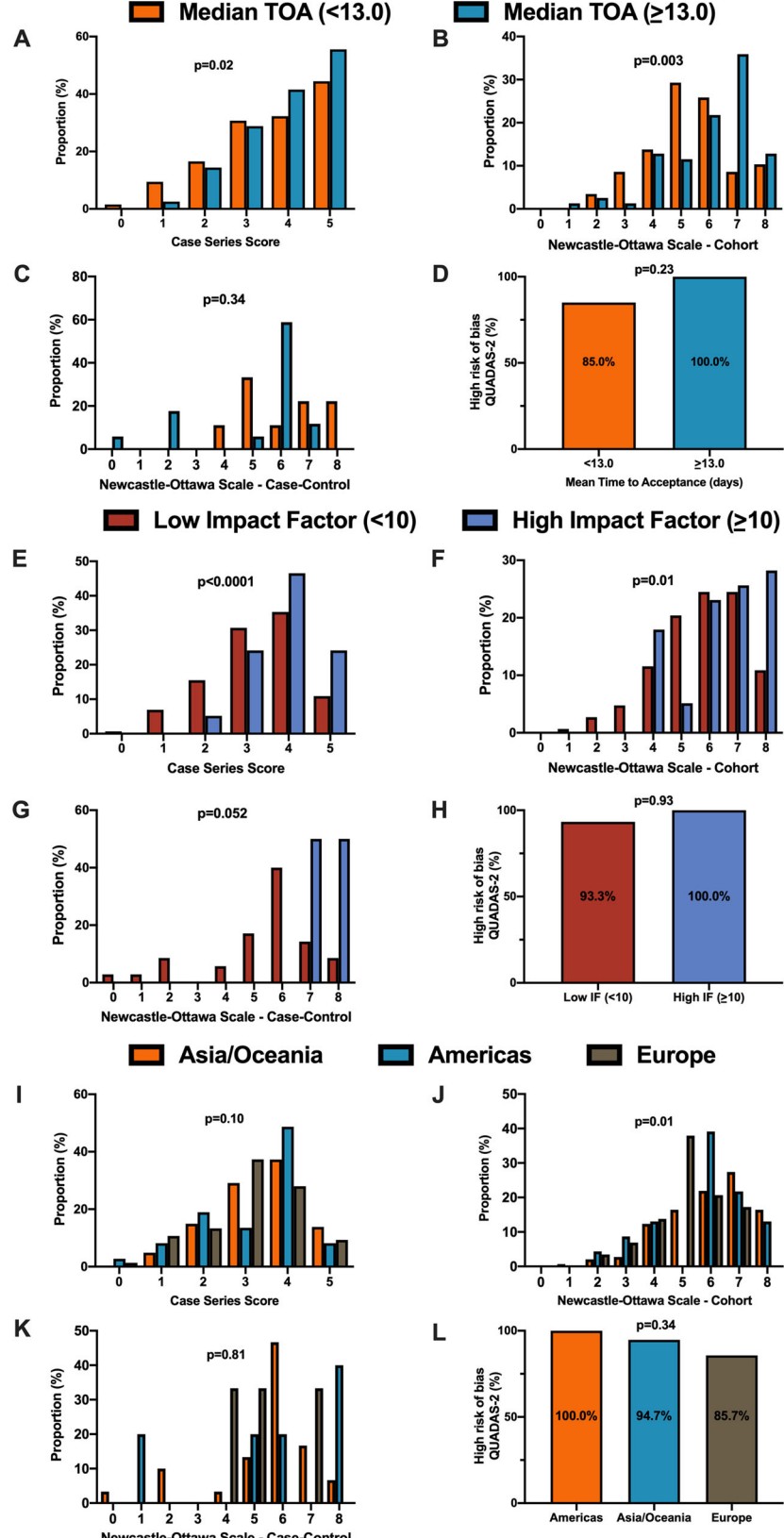

exist to evaluate methodological quality of identified manuscripts, although no consensus currently exists for critical appraisal of non-randomized studies[41–43]. Furthermore, other considerations of quality such as sample size calculation, sex reporting or ethics approval are not considered in these quality scores. As such, the quality scores measured using these checklists only reflect the patient selection, comparability, diagnostic reference standard and methods to ascertain the outcome of the study. Third, the 1:1

**Fig. 3 Differences in methodological quality scores in COVID-19 by secondary outcomes. A** When stratified by time of acceptance (13.0 days), increased time of acceptance was associated with higher case series score ($n = 186$ for <13 days and $n = 193$ for >=13 days; $p = 0.02$). **B** Increased time of acceptance was associated with higher NOS cohort score ($n = 112$ for <13 days and $n = 144$ for >=13 days; $p = 0.003$). **C** No difference in time of acceptance and case–control score was observed ($n = 18$ for <13 days and $n = 27$ for >=13 days; $p = 0.34$). **D** No difference in time of acceptance and diagnostic risk of bias (QUADAS-2) was observed ($n = 43$ for <13 days and $n = 33$ for >=13 days; $p = 0.23$). **E** When stratified by impact factor (IF ≥10), high IF was associated with higher case series score ($n = 466$ for low IF and $n = 60$ for high IF; $p < 0.0001$). **F** High IF was associated with higher NOS cohort score ($n = 262$ for low IF and $n = 68$ for high IF; $p = 0.01$). **G** No difference in IF and case–control score was observed ($n = 62$ for low IF and $n = 2$ for high IF; $p = 0.052$). **H** No difference in IF and QUADAS-2 was observed ($n = 101$ for low IF and $n = 2$ for high IF; $p = 0.93$). **I** When stratified by geographical region, no difference in geographical region and case series score was observed ($n = 276$ Asia/Oceania, $n = 135$ Americas, and $n = 143$ Europe/Africa; $p = 0.10$). **J** Geographical region was associated with differences in cohort score ($n = 177$ Asia/Oceania, $n = 81$ Americas, and $n = 89$ Europe/Africa; $p = 0.01$). **K** No difference in geographical region and case–control score was observed ($n = 37$ Asia/Oceania, $n = 13$ Americas, and $n = 14$ Europe/Africa; $p = 0.81$). **L** No difference in geographical region and QUADAS-2 was observed ($n = 49$ Asia/Oceania, $n = 28$ Americas, and $n = 28$ Europe/Africa; $p = 0.34$). In panels **A**–**D**, orange represents lower median time of acceptance and blue represents high median time of acceptance. In panels **E**–**H**, red is low impact factor and blue is high impact factor. In panels **I**–**L**, orange represents Asia/Oceania, blue represents Americas, and brown represents Europe. Differences in distributions were analysed by two-sided Kruskal–Wallis test. Differences in diagnostic risk of bias were quantified by Chi-squares test. $p < 0.05$ was considered statistically significant.

ratio to identify our historical control articles may affect the precision estimates of our findings. Interestingly, a simulation of an increase from 1:1 to 1:4 control ratio tightened the precision estimates but did not significantly alter the point estimate[44]. Furthermore, the decision for 1:1 ratio in our study exists due to limitations of available historical control articles from the identical journal in the restricted time period combined with a large effect size and sample size in the analysis. Finally, our analysis includes early publications on COVID-19 and there is likely to be an improvement in quality of related studies and study design as the field matures and higher-quality studies. Accordingly, our findings are limited to the early body of research as it pertains to the pandemic and it is likely that over time research quality will improve over time.

In summary, the early body of peer-reviewed COVID-19 literature was composed primarily of observational studies that underwent shorter peer-review evaluation and were associated with lower methodological quality scores than comparable studies. COVID-19 clinical studies should be revisited with the emergence of stronger evidence.

## Methods

A systematic literature search was conducted on May 14, 2020 (registered on June 3, 2020 at PROSPERO: CRD42020187318) and reported according to the Preferred Reporting Items for Systematic Reviews and Meta-Analyses. Furthermore, the cohort study was reported according to the Strengthening The Reporting of Observational Studies in Epidemiology checklist. The data supporting the findings of this study is available as Supplementary Data 1–2.

**Data sources and searches.** The search was created in MEDLINE by a medical librarian with expertise in systematic reviews (S.V.) using a combination of key terms and index headings related to COVID-19 and translated to the remaining bibliographic databases (Supplementary Tables 1–3). The searches were conducted in MEDLINE (Ovid MEDLINE(R) ALL 1946–), Embase (Ovid Embase Classic + Embase 1947–) and the Cochrane Central Register of Controlled Trials (from inception). Search results were limited to English-only publications, and a publication date limit of January 1, 2019 to present was applied. In addition, a Canadian Agency for Drugs and Technologies in Health search filter was applied in MEDLINE and Embase to remove animal studies, and commentary, newspaper article, editorial, letter and note publication types were also eliminated. Search results were exported to Covidence (Veritas Health Innovation, Melbourne, Australia) and duplicates were eliminated using the platform's duplicate identification feature.

**Study selection, data extraction and methodological quality assessment.** We included all types of COVID-19 clinical studies, including case series, observational studies, diagnostic studies and RCTs. For diagnostic studies, the reference standard for COVID-19 diagnosis was defined as a nasopharyngeal swab followed by reverse transcriptase-polymerase chain reaction in order to detect SARS-CoV-2. We excluded studies that were exploratory or pre-clinical in nature (i.e. in vitro or animal studies), case reports or case series of <5 patients, studies published in a

language other than English, reviews, methods or protocols, and other coronavirus variants such as the Middle East respiratory syndrome.

The review team consisted of trained research staff with expertise in systematic reviews and one trainee. Title and abstracts were evaluated by two independent reviewers using Covidence and all discrepancies were resolved by consensus. Articles that were selected for full review were independently evaluated by two reviewers for quality assessment using a standardized case report form following the completion of a training period where all reviewers were trained with the original manuscripts which derived the tools or checklists along with examples for what were deemed high scores[35–38]. Following this, reviewers completed thirty full-text extractions and the two reviewers had to reach consensus and the process was repeated for the remaining manuscripts independently. When two independent reviewers were not able reach consensus, a third reviewer (principal investigator) provided oversight in the process to resolve the conflicted scores.

First and corresponding author names, date of publication, title of manuscript and journal of publication were collected for all included full-text articles. Journal impact factor was obtained from the 2018 InCites Journal Citation Reports from Clarivate Analytics. Submission and acceptance dates were collected in manuscripts when available. Other information such as study type, prospective or retrospective study, sex reporting, sample size calculation, method of SARS-CoV-2 diagnosis and ethics approval was collected by the authors. Methodological quality assessment was conducted using the Newcastle–Ottawa Scale (NOS) for case–control and cohort studies[37], QUADAS-2 tool for diagnostic studies[38], Cochrane risk of bias for RCTs[35] and a score derived by Murad et al. for case series studies[36].

**Identification of historical control from identified COVID-19 articles.** Following the completion of full-text extraction of COVID-19 articles, we obtained a historical control group by identifying reports matched in a 1:1 fashion. From the eligible COVID-19 article, historical controls were identified by searching the same journal in a systematic fashion by matching the same study design ("case series", "cohort", "case control" or "diagnostic") starting in the journal edition 12 months prior to the COVID-19 article publication on the publisher website (i.e. COVID-19 article published on April 2020, going backwards to April 2019) and proceeding forward (or backward if a specific article type was not identified) in a temporal fashion until the first matched study was identified following abstract screening by two independent reviewers. If no comparison article was found by either reviewers, the corresponding COVID-19 article was excluded from the comparison analysis. Following the identification of the historical control, data extraction and quality assessment was conducted on the identified articles using the standardized case report forms by two independent reviewers and conflicts resolved by consensus. The full dataset has been made available as Supplementary Data 1–2.

**Data synthesis and statistical analysis.** Continuous variables were reported as mean (SD) or median (IQR) as appropriate, and categorical variables were reported as proportions (%). Continuous variables were compared using Student $t$-test or Mann–Whitney U-test and categorical variables including quality scores were compared by $\chi^2$, Fisher's exact test, or Kruskal–Wallis test.

The primary outcome of interest was to evaluate the methodological quality of COVID-19 clinical literature by study design using the Newcastle–Ottawa Scale (NOS) for case–control and cohort studies, QUADAS-2 tool for diagnostic studies[38], Cochrane risk of bias for RCTs[35], and a score derived by Murad et al. for case series studies[36]. Pre-specified secondary outcomes were comparison of methodological quality scores of COVID-19 articles by (i) median time to acceptance, (ii) impact factor, (iii) geographical region and (iv) historical comparator. Time of acceptance was defined as the time between submission to acceptance which captures peer review and editorial decisions. Geographical region

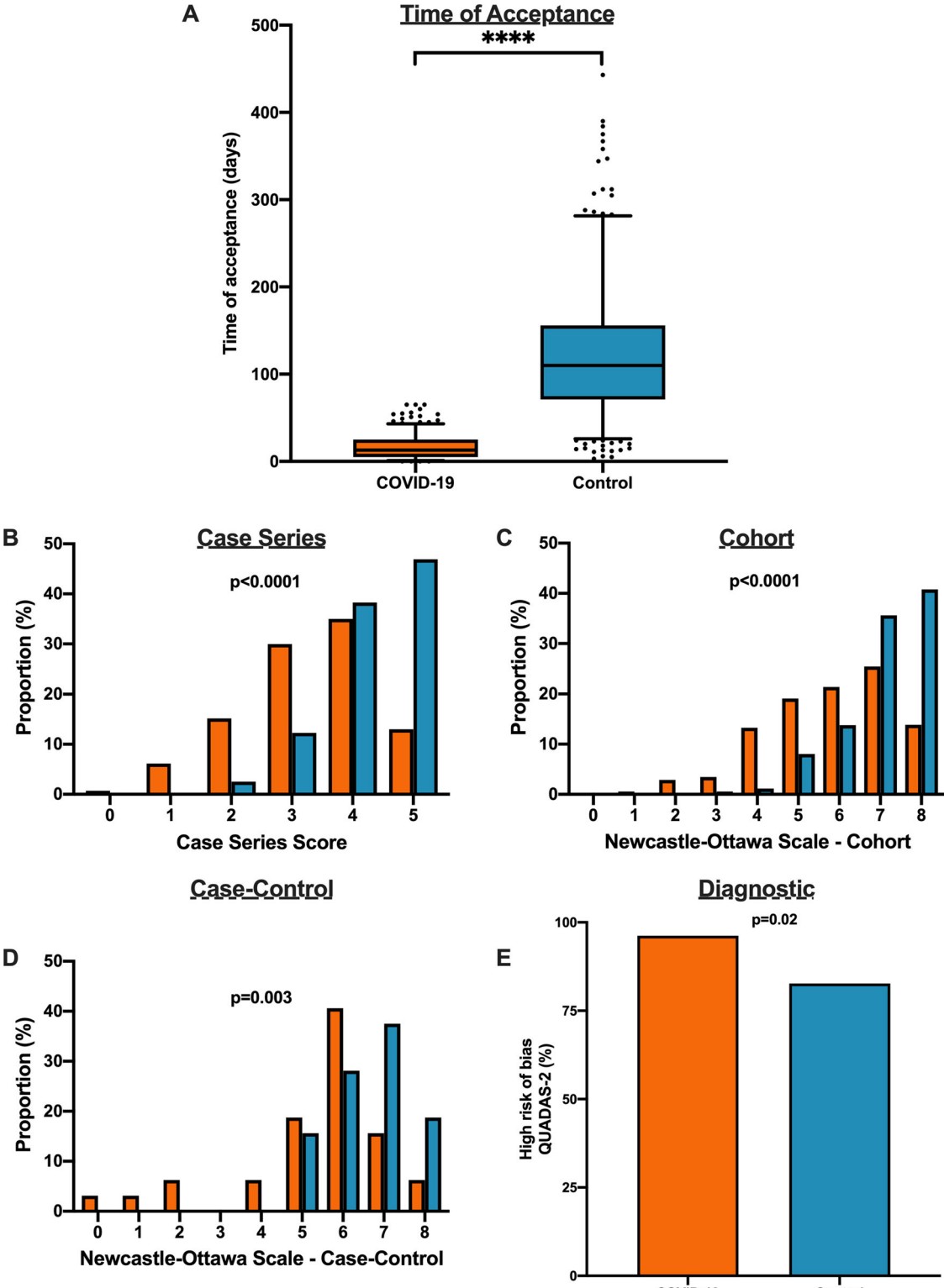

**Fig. 4 Differences in methodological quality scores in COVID-19 compared to historical control articles. A** Time to acceptance was reduced in COVID-19 articles compared to control articles (13.0 [IQR, 5.0–25.0] days vs. 110.0 [IQR, 71.0–156.0] days, $n = 347$ for COVID-19 and $n = 414$ for controls; $p < 0.0001$). **B** When compared to historical control articles, COVID-19 articles were associated with lower case series score ($n = 277$ for COVID-19 and $n = 277$ for controls; $p < 0.0001$). **C** COVID-19 articles were associated with lower NOS cohort score compared to historical control articles ($n = 174$ for COVID-19 and $n = 174$ for controls; $p < 0.0001$). **D** COVID-19 articles were associated with lower NOS case-control score compared to historical control articles ($n = 32$ for COVID-19 and $n = 32$ for controls; $p = 0.003$). **E** COVID-19 articles were associated with higher diagnostic risk of bias (QUADAS-2) compared to historical control articles ($n = 53$ for COVID-19 and $n = 53$ for controls; $p = 0.02$). For panel **A**, boxplot captures 5, 25, 50, 75 and 95% from the first to last whisker. Orange represents COVID-19 articles and blue represents control articles. Two-sided Mann–Whitney U-test was conducted to evaluate differences in time to acceptance between COVID-19 and control articles. Differences in study quality scores were evaluated by two-sided Kruskal–Wallis test. Differences in diagnostic risk of bias were quantified by Chi-squares test. $p < 0.05$ was considered statistically significant.

was stratified into continents including Asia/Oceania, Europe/Africa and Americas (North and South America). Post hoc comparison analysis between COVID-19 and historical control article quality scores were evaluated using Kruskal–Wallis test. Furthermore, good quality of NOS was defined as 3+ on selection and 1+ on comparability, and 2+ on outcome/exposure domains and high-quality case series scores were defined as a score ≥3.5. Due to a small sample size of identified RCTs, they were not included in the comparison analysis.

The finalized dataset was collected on Microsoft Excel v16.44. All statistical analyses were performed using SAS v9.4 (SAS Institute, Inc., Cary, NC, USA). Statistical significance was defined as $P < 0.05$. All figures were generated using GraphPad Prism v8 (GraphPad Software, La Jolla, CA, USA).

**Reporting summary**. Further information on research design is available in the Nature Research Reporting Summary linked to this article.

## Data availability
The authors can confirm that all relevant data are included in the paper and in Supplementary Data 1–2. The original search was conducted on MEDLINE, Embase and Cochrane Central Register of Controlled Trials.

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

## Acknowledgements
This study received no specific funding or grant from any agency in the public, commercial, or not-for-profit sectors. R.G.J. was supported by the Vanier CIHR Canada Graduate Scholarship. F.D.R. was supported by a CIHR Banting Postdoctoral Fellowship and a Royal College of Physicians and Surgeons of Canada Detweiler Travelling Fellowship. The funder/sponsor(s) had no role in design and conduct of the study, collection, analysis and interpretation of the data.

## Author contributions
R.G.J., P.D.S., S.V., F.D.R., T.S. and B.H. participated in the study conception and design. Data acquisition, analysis and interpretation were performed by R.G.J., P.D.S., C.C.,

G.P.P., S.P., S.S., A.H., F.D.R., T.S. and B.H. Statistical analysis was performed by R.G.J., P.D.S. and B.H. The manuscript was drafted by R.G.J., P.D.S., F.D.R., T.S. and B.H. All authors approved the final version of the manuscript and agree to be accountable to all aspects of the work.

## Competing interests

B.H. reports funding as a clinical trial investigator from Abbott, Boston Scientific and Edwards Lifesciences outside of the submitted work. The remaining authors declare no competing interests.
