## [Peer Review File · Nature Communications]

Reviewer comments, first round:

Reviewer #1 (Remarks to the Author):

The authors perform a systematic review and case control study investigating the methodological quality of clinical studies related to COVID-19. They identify 686 studies published to May 2020 (case series, case-controls, diagnostic studies. They then compare the 'quality' of these studies versus historical controls in the same journal. Overall, I think this is an important study as it investigates an issue that has been raised by many in the scientific community (i.e. what is the quality of COVID-19 studies published to date?). In fact, looking at #science Twitter today I noted at least two prominent voices asking this exact question. I have provided some feedback/questions on concerns below.

For context of my review, I am a physician and scientist with expertise in knowledge synthesis, reporting, and risk of bias/quality assessments of clinical literature. I have no expertise in COVID-19 or SARS-CoV-2 other than practically dealing with affected patients throughout the pandemic.

1. The study is as much a case-control study as a systematic review. This should be made clear in the abstract and methods.

2. Since this is a case-control study the STROBE guidelines for reporting should be followed as well.

3. For a study investigating methodological details, I realize the authors appreciate the importance of transparent reporting of study conduct and details. As such, I was surprised by the absence of significant details in the methods. These suggestions over the next several points must be elaborated, I believe. I fully understand the limited space that this journal offers, however there are no limitations in supplemental online methods.

4. A deviations from protocol section should be included (even as a supplemental is fine). The readers should be aware of what was planned a priori and what was not.

5. The use of the tools used to assess 'quality' is not trivial and requires content expertise, training, and oversight. How was this addressed? In addition, how were the extractors ability to use these tools appropriately assessed? This is a significant issue.

6. Although I appreciate that the authors needed to dichotomize these tools into high vs low quality using a simplified score, the tools themselves were never designed for this purpose. This is a significant limitation and should be acknowledged. In fact, there are many that suggest a simplified score should never be applied to risk of bias assessments as this approach over simplifies a very complex issue.

7. Data for individual studies assessed must be submitted for review. It was impossible for me to assess the quality of the study extraction without this granular data to review. I would encourage the authors to also provide this as supplemental online data for others to review when published.

8. The study appears to be either retrospectively registered, or incorrectly registered at best. This is a significant issue.

a) In PROSPERO it is stated that preliminary searches were not complete as of June 3, 2020. Yet the final search for this study was completed in May.

b) What else was complete prior to registration? This must be reported.

9. The analysis plan (especially with regards to regression that was presented) was not prespecified. This should be clearly labelled as post-hoc.

10. The cut-offs for good vs poor quality were not prespecified in the protocol. These should be clearly reported as post-hoc.

11. The statistical tests themselves seemed appropriate (they were all generally simple ones I am familiar with), however I am not a statistician and would defer to another reviewer with greater statistical expertise.

12. It is unclear why a 1:1 case to control ratio was used. Typically, 1:4 is thought to be optimal. For associative studies 1:2 could be argued. 1:1 must be rationalized and the potential limitations of this should be discussed.

13. Who conducted the search? If an information specialist was involved this should be stated and they should be acknowledged.

14. Was the search peer reviewed by another information specialist (i.e. Peer Review of Electronic Search Strategy, PRESS)?

15. Full search strategies for all databases should be included (I saw only MEDLINE included in the

supplemental).

16. The search filter from the Canadian Agency for Drugs and Technology in Health should be cited. I'm not entirely clear which filter this was.

17. What was the concordance between reviewers for study selection (e.g. interrater reliability). Was there piloting of screening and selection of studies? Similarly, was their piloting for study data extraction?

18. The first three boxes in the PRISMA flow diagram do not add up (i.e. records identified minus duplicates does not equal records screened).

19. Why are no details provided for the 6 RCTs identified? I realize these represents only a tiny fraction of the studies identified, however the details could be provided as online supplemental material at the very least. Also, related to this, the cut-off for high vs low quality using the RoB tool wasn't stated in the methods.

MINOR:

1. Covidence, on first use, should describe source and location.

2. The PRISMA guidelines are reporting guidelines, not conduct guidelines. Thus, it is incorrect to say this study was 'conducted according to the [PRISMA] guidelines.' I realize that this is stated in many systematic reviews, but it is simply incorrect.

Reviewer #2 (Remarks to the Author):

This paper evaluates the quality of the clinical literature on COVID-19 published up to 14th May 2020, both in absolute terms and relative to control papers matches on study type and journal. Its key finding is that the COVID papers have substantially lower quality than matched controls. The authors have assessed the quality of 686 COVID papers and 539 controls so this is an enormous piece of work.

Ironically, the paper itself bears the marks of hasty preparation for publication. Many of my comments can be addressed in a revision. I think the crucial issue is to identify the limitation of this work (3a below) that it cannot say whether the trade-off that we have made (between quality and speed) was appropriate.

Major points

1. Methods.

a. Study type is defined inconsistently. At l76-77 it is defined as case series, observational studies, diagnostic studies, and RCTs, but in the figures the RCTs disappear and the observational studies are subdivided. Further, the results for RCTs are never reported (only the fact that there were 6 RCTs).

b. I was unclear why over 9000 of the 9895 COVID-19 papers identified were rejected. Was a sample used to reduce workload? If so this must be stated.

c. The quality of the assessment of quality is crucial. It would be useful to see measures of agreement between the two reviewers who assessed quality. What measures were taken to apply comparable standard to the COVID-19 and control papers?

2. Reporting.

a. The results section includes a lot of repetition of the tables and figures – e.g. the para at l1148-161 almost exactly repeats F3. This is undesirable. Conversely some text describes findings that are not in the figures at all, e.g. the text at l128-135 does NOT match what is reported in F1.

b. T2 reports the association of other covariates (e.g. high impact factor) with methodological quality. I don't think this is relevant to the aims of this paper and therefore I think it should be dropped. Also, high methodological quality should be included in table 1. If table 2 is kept as it is

then please note that COVID-19 vs control is not a journal characteristic and the footnote does not make sense since regression adjustment applies to coefficients, not to articles.

3. Interpretation.

a. A lower methodological quality and faster time to publication has been demonstrated for COVID-related papers compared with controls. However the present evidence does not show that this was a bad thing. The authors give good arguments in the discussion that the low quality has been a problem, but they do not discuss how slower time to publication might also have been a problem.

b. Abstract (I31-32), "greater effort to appropriately weigh the existing evidence in the context of emerging high-quality research is needed" – I'm not sure I understand this conclusion. It appears to be a criticism of how the evidence is currently reviewed, and this was not the topic of this research.

c. The conclusions at LL188-190 relate to the other covariates and therefore seem inappropriate (as noted above). If the aim was to evaluate these other factors then careful assessment of confounders would be needed. This is in fact a perfect illustration of the table 2 fallacy (Westreich D, Greenland S. The Table 2 Fallacy: Presenting and Interpreting Confounder and Modifier Coefficients. *Am J Epidemiol.* 2013;177(4):292-298).

d. It would have been useful to know in what respects the COVID-19 papers are of lower quality.

Minor points

4. Language requires some attention, for example:

a. A number of prepositions are omitted e.g. I179 "shorter time from submission to acceptance COVID-19 articles", I153 "High case series quality was observed 133 COVID-19 articles".

b. "comparative" used when "comparable" meant, e.g. I29

5. I do not think this is a case-control design. A case-control design would involve taking some low-quality papers, matching them to high-quality papers, and assessing whether they addressed COVID-19. I think the design is a "matched cohort design". (This comment is only about terminology and is not a criticism of the methods.)

6. L42-44, conducting the research is also important! Ref 3 included this under point "2)".

7. Some confusion between study conduct and study reporting. e.g. I59 PRISMA is not a guideline for study conduct.

8. L67-68, I was surprised articles back to January 2019 were included. Given that COVID-19 was not defined until 2020, how many papers were retrieved from 2019? Related to this, it would be useful to see the distribution of articles by month of publication.

9. L76, "RCTs evaluating COVID-19" – I hope they were instead evaluating treatments for COVID-19.

10. L76-78, I cannot see the relevance of this.

11. L101, give a reference for the NOS.

12. L142 and elsewhere, avoid causal language like "resulted in".

13. L174-5, the study designs were low in the hierarchy of scientific evidence: this is not surprising since this was a very new science.

14. L242, median is a property of a sample, not of a single manuscript.

15. F2, is the assessment for diagnostic studies lacking the words "risk of"?
16. F2 and F3 should not report the assessment for case-control studies differently (2 categories in F3, 2 in F2).
17. F2 and F3, reporting 2 decimal places is too many.
18. T1, impact factor should not be reported to one decimal place since it is an integer. Also please note (in a footnote?) that articles are implicitly matched on this variable.

Reviewer #3 (Remarks to the Author):

Thank you for the opportunity to review this paper. I believe that several things were done very well, including the search strategy and pre-registration and the reporting of the results as specified in the methods section. I also believe this to be a critically important topic.

In order to focus on the areas of greatest concern, I have focused on critique of the methods in particular. I have largely skipped the Introduction and Discussion as a result, which I found to be well written and compelling.

Overall, I did not find this to be compelling in its current form due to three main issues: 1) The methods for evaluating "quality" and "rigor" are extremely limited. I would not refer to this as quality assessments at all, and certainly not call these studies "high quality" if they had a higher score set by an arbitrary threshold. 2) The matched sample procedures are undocumented, and unlikely appropriate for comparison. At the very least, the procedures need to be documented and made replicable. 3) The statistical model and dichotomization of the data were unnecessary and inappropriate for analysis.

Given that this study is about assessing methodological quality and rigor, the methods used to evaluate quality should meet a high standard, which I do not believe was achieved here.

A heavy revision would vastly improve this paper, including revising the language throughout to avoid calling this a "quality" or "rigor" assessment (especially avoiding suggesting "high" quality", removing the counterfactual (see below for alternative suggestions). A heavy revision of the statistical methods and dichotomization will greatly help make the main results more believable, possibly removing any linkage between score and categorical "quality" assessment.

Methods:

Data sources and searches

I appreciate the full details in the search terms; this seems very replicable and followable.

Data extraction and quality assessment:

"A historical comparator group was generated by identifying reports of the same study design matched in a 1:1 fashion."

This is the only sentence describing the construction of the counterfactual comparator on which one of the key claims is made. However, this does not contain nearly enough detail. How were these identified? What search terms? How were they matched? In what period of time were they identified? What does "same study design mean?" Without this information, I cannot verify the veracity of any claim made based on the counterfactual comparison. Some additional detail is provided in the results (same journal, previous year), but this needs to be in the methods section.

As presented, it looks like this matching was done by hand, in a non-systematic and non-replicable manner. If that is the case, it runs severe risk of selection-related biases, and is not appropriate for use in this analysis, and I believe it should be removed from the paper or redone using a

systematic method.

Data synthesis and analysis

One of my biggest concerns here is that the assessments for quality are extremely blunt instruments, particularly the NOS scale. The NOS scale is often used for purposes such as this, but is an incredibly inadequate measure of quality, and may even be counter-indicative of methodological and analytical rigor in many cases. The majority of scoring is on extremely superficial measures, without dealing with the fundamentally most important issues (often issues with causal identification).

In the case of the Risk of Bias measure, Cochrane explicitly recommends against using RoB for summary scores, for these very reasons. I am unfamiliar with the QUADAS-2 and the Murad tools, so I can't comment on those.

Secondly, I am concerned that these scores, which are questionable to begin with, are further degraded by dichotomizing the scores into "high" and "low." There is no justification, as far as I can tell, to dichotomize the data, and doing so can lead to substantially misleading inference. Either use much finer ordinal bins, or keep the scoring in its original form.

Rhetorically, labelling the quality of studies as "high quality" is also not justified; at best they are just not found to be of low quality. They may have a high score, but that scoring is not a comprehensive assessment of quality. I strongly suggest removing the word "quality" whenever in reference to the score, and perhaps from the title.

The main justification for the dichotomization seems to be to put the data in a form for plugging into logistic regression for odds ratios. However, even were the scores already binary high/low form, ORs are difficult to interpret, and models for RR should be used instead. If not dichotomizing, many other statistical model options are available that would be vastly preferable, including any number of linear models, different in means/median models, ordinal regression, etc.

Results: Within the scope of the methods as discussed, I believe this to have been relatively well-reported.

One question regards why "adjustments" were made for some analysis. What purpose did this serve? This was not described or justified in the methods, particularly the variables adjusted for. How do I usefully interpret this adjusted association? It seems very likely that this is hinting at some kind of causal model but pursuing the "just say association" strategy to sidestep having to justify it. I would strongly recommend removing the adjusted models altogether.

Discussion: I would suggest removing the "this is the first time" sentence. Our contribution should be about rigor, and not about being "first" to anything.

The limitations section needs to be vastly increased. To begin with, it needs a thorough discussion of the limitations of the review tools used related to your assessment of "quality."

REVIEWER COMMENTS

Reviewer #1 (Remarks to the Author):

The authors perform a systematic review and case control study investigating the methodological quality of clinical studies related to COVID-19. They identify 686 studies published to May 2020 (case series, case-controls, diagnostic studies. They then compare the ‘quality’ of these studies versus historical controls in the same journal. Overall, I think this is an important study as it investigates an issue that has been raised by many in the scientific community (i.e. what is the quality of COVID-19 studies published to date?). In fact, looking at #science Twitter today I noted at least two prominent voices asking this exact question. I have provided some feedback/questions on concerns below.

For context of my review, I am a physician and scientist with expertise in knowledge synthesis, reporting, and risk of bias/quality assessments of clinical literature. I have no expertise in COVID-19 or SARS-CoV-2 other than practically dealing with affected patients throughout the pandemic.

We thank the reviewer for taking the time to evaluate our manuscript where we evaluated the overall quality of recent COVID-19 literature to historical controls. We are not experts in managing COVID-19 but have previous experience in systematically evaluating the quality of preclinical literature.^{1,2} We wish to answer an important question that remains prevalent in #medtwitter, in that the methodological quality score of COVID-19 literature is lower compared to the historical control.

1. The study is as much a case-control study as a systematic review. This should be made clear in the abstract and methods.

The reviewer is correct in that the study is a systematic review. Indeed, after the recommendation of reviewer #2 along with our in-house statistician, they have mentioned that the study design is that of matched cohort and we have completed a STROBE checklist and it may be found in the **Supplemental Appendix**. This has been reflected in both the abstract and methods section and is found on **pages 3-4 and below**.

Abstract

Background: COVID-19 spread globally in early 2020 with major health consequences. While a need to disseminate information to the medical community, governmental agencies, and general

public was paramount - concerns have been raised regarding scientific rigor in published reports. We sought to evaluate the quality of reporting of currently available COVID-19 studies compared to historical controls.

Methods: MEDLINE, Embase, and Cochrane Central Register of Controlled Trials until May 14, 2020. All original clinical literature evaluating COVID-19 were identified and 1:1 historical control of the same study type in the same published journal was matched from the previous year. Two independent reviewers screened titles, abstracts, and full-texts and independently assessed methodological quality.

Results: 9895 titles and abstracts were screened, and 686 COVID-19 articles were included in the final analysis. Overall, COVID-19 articles were stratified by study design, a mean case series score (out of 5) (\pm SD) of 3.3 ± 1.1 , mean NOS cohort study score (out of 8) of 5.8 ± 1.5 , mean NOS case control study score (out of 8) of 5.5 ± 1.9 , and low bias present in 4 (6.4%) diagnostic studies and differences in quality scores were observed when dichotomized by time of submission to acceptance and journal impact factor. COVID-19 articles to historical control articles were matched in a 1:1 fashion (n=1078). The median time to acceptance was 13.0 (IQR, 5.0-25.0) days in COVID-19 articles vs. 110.0 (IQR, 71.0-156.0) days in control articles ($p < 0.0001$). Overall, methodological quality score was lower in COVID-19 articles in all study designs compared to control articles.

Conclusion: Published COVID-19 studies were accepted more quickly and were found to be of lower reported methodological quality than matched studies published in the same journal. An appreciation of the limitations of existing data and their appropriate appraisal when making clinical decisions or investing research efforts are warranted.”

2. Since this is a case-control study the STROBE guidelines for reporting should be followed as well.

We have attached the appropriate STROBE checklist of items with the appropriate page numbers for cohort studies along with the PRISMA checklist used for the systematic review and maybe found as **supplemental appendix** and below.

3. For a study investigating methodological details, I realize the authors appreciate the importance of transparent reporting of study conduct and details. As such, I was surprised by the absence of significant details in the methods. These suggestions over the next several points must be elaborated, I believe. I fully understand the limited space that this journal offers, however there are no limitations in supplemental online methods.

We thank the reviewer for suggestions in order to improve our methods section as this manuscript aims to highlight the shortcomings of COVID-19 literature and the methods have been improved and answered in the upcoming questions.

4. A deviations from protocol section should be included (even as a supplemental is fine). The readers should be aware of what was planned a priori and what was not.

We thank the reviewer for the important point that our protocol section should clarify analyses which were determined *a priori* and *post-hoc*. As per the PROSPERO registration, the primary outcome of interest was to evaluate the quality of COVID-19 literature using the Newcastle-Ottawa Scale, QUADAS-2 tool, Cochrane Risk of Bias, and the Murad et al. score.^{3, 4, 5, 6} Other pre-specified secondary outcomes of interest were to compare the COVID-19 literature quality scores by the median time to acceptance, impact factor, geographical region, and historical comparator.

For a *post-hoc* analysis, the specific categorization of geographical region was not pre-specified and we have amended the manuscript in order to reflect the categorization of the geographic region. Furthermore, the dichotomization of high vs. low quality scores or comparison by continuous/categorical variable was determined *post-hoc* in order to conduct a comparison analysis by χ^2 test or Kruskal-Wallis test. Indeed, the reviewer (and other reviewers) rightfully

comment that the dichotomization of these quality scores oversimplifies the evaluation and we have amended our manuscript to reflect those changes. The methods section has been improved and the corresponding information is found on **pages 6-9** and below.

Methods

A systematic literature search was conducted on May 14, 2020 (**PROSPERO: CRD42020187318**) and reported according to the Preferred Reporting Items for Systematic Reviews and Meta-Analyses. Furthermore, the cohort study was reported according to the Strengthening The Reporting of Observational studies in Epidemiology checklist. The data supporting the findings of this study is available in the Supplemental Appendix.

Data sources and searches

The search was created in MEDLINE by a medical librarian with expertise in systematic reviews (S.V.) using a combination of key terms and index headings related to COVID-19 and translated to the remaining bibliographic databases (**Supplemental Tables 1-3**). The searches were conducted in MEDLINE (Ovid MEDLINE(R) ALL 1946-), Embase (Ovid Embase Classic + Embase 1947-) and the Cochrane Central Register of Controlled Trials (from inception). Search results were limited to English-only publications, and a publication date limit of January 1, 2019 to present was applied. In addition, a Canadian Agency for Drugs and Technologies in Health search filter was applied in MEDLINE and Embase to remove animal studies, and comment, newspaper article, editorial, letter and note publication types were also eliminated. Search results were exported to Covidence (Veritas Health Innovation, Melbourne, Australia) and duplicates were eliminated using the platform's duplicate identification feature.

Study selection, data extraction, and quality assessment

We included all types of COVID-19 clinical studies, including case series, observational studies, diagnostic studies, and RCTs. For diagnostic studies, the reference standard for COVID-19 diagnosis was considered a nasopharyngeal swab followed by reverse transcriptase polymerase chain reaction in order to detect SARS-CoV-2. We excluded studies that were exploratory or pre-clinical in nature (ie. *in vitro* or animal studies), case reports or case series of <5 patients, studies published in a language other than English, reviews, methods or protocols, and other coronavirus variants such as the Middle East Respiratory Syndrome.

Title and abstracts were evaluated by two independent reviewers (a team of internal medicine or cardiology fellow and a medical student) using Covidence and all discrepancies were resolved by consensus. Articles that were selected for full review were independently evaluated by two reviewers for quality assessment using a standardized case report form following the completion of a training period. When two independent reviewers were not able to come to a consensus score, a third reviewer (principal investigator) provided oversight in the process to resolve the conflicted scores.

First and corresponding author names, date of publication, title of manuscript, and journal of publication were collected for all included full-text articles. Journal impact factor was obtained from the 2018 InCites Journal Citation Reports from Clarivate Analytics. Submission and acceptance dates were collected in manuscripts when available. Other information such as study type, prospective or retrospective study, sex reporting, sample size calculation, method of SARS-CoV-2 diagnosis, and ethics approval was collected by the authors. Quality assessment was

conducted using the Newcastle-Ottawa Scale (NOS) for case-control and cohort studies,⁷ QUADAS-2 tool for diagnostic studies,⁴ Cochrane Risk of Bias for RCTs,⁵ and a score derived by Murad et al. for case series were collected.⁶

Identification of historical control from identified COVID-19 articles

After full-text extraction of COVID-19 articles was completed, we obtained a historical control group by identifying reports of the same study design matched in a 1:1 fashion. From the eligible COVID-19 article, historical controls were identified by searching the same journal in a systematic fashion by matching for the same study design (“case series”, “cohort”, “case control”, or “diagnostic”) starting in the journal edition 12 months prior to the COVID-19 article publication on the publisher website (ie. COVID-19 article published on April 2020, going backwards to April 2019) and proceeding forward (or backward if a specific article type was not identified) in a temporal fashion until the first matched study was identified following abstract screening by two independent reviewers. If no comparison article was found by either reviewers, the corresponding COVID-19 article was excluded from the analysis. Following the identification of the historical control, data extraction and quality assessment was conducted on the identified articles independently using the standardized case report forms by two reviewers and conflicts resolved by consensus. The full dataset has been made available in the **Supplemental Appendix**.

Data synthesis and statistical analysis

Continuous variables were reported as mean \pm SD or median \pm IQR as appropriate, and categorical variables were reported as proportions (%). Normally distributed continuous

variables were compared using Mann-Whitney U-test and categorical variables and quality scores were compared by χ^2 , Fisher's exact test, or Kruskal-Wallis test.”

5. The use of the tools used to assess ‘quality’ is not trivial and requires content expertise, training, and oversight. How was this addressed? In addition, how were the extractors ability to use these tools appropriately assessed? This is a significant issue.

The reviewer is correct in that the use of Newcastle-Ottawa Scale³, QUADAS-2 tool,⁴ Cochrane Risk of Bias tool,⁵ and the Murad score⁶ remains limited by subjectivity of the reviewer and requires content expertise, training, and oversight. In order to address content expertise, we first trained all reviewers with the original manuscript which derived and evaluated the scores along with checklists and examples for what were deemed high scores. Furthermore, independent review was conducted by two reviewers, which consisted of a trained research staff with expertise in systematic reviews/quality assessment and one research trainee. After the completion of thirty full-text extractions, the two reviewers had to reach consensus and the process repeated throughout the remaining manuscripts independently. Furthermore, when two independent reviewers could not come to a consensus in the score, a third reviewer, the principal investigator of the study provided oversight in the process to resolve the conflicted scores. Finally, the extractors ability to assess data extraction was evaluated by generating Kappa values following the completion of data extraction.

The following information has been amended to the methods section and is found on **pages 6-9** and below.

Methods

A systematic literature search was conducted on May 14, 2020 (**PROSPERO: CRD42020187318**) and reported according to the Preferred Reporting Items for Systematic Reviews and Meta-Analyses. Furthermore, the cohort study was reported according to the Strengthening The Reporting of Observational studies in Epidemiology checklist. The data supporting the findings of this study is available in the Supplemental Appendix.

Data sources and searches

The search was created in MEDLINE by a medical librarian with expertise in systematic reviews (S.V.) using a combination of key terms and index headings related to COVID-19 and translated to the remaining bibliographic databases (**Supplemental Tables 1-3**). The searches were conducted in MEDLINE (Ovid MEDLINE(R) ALL 1946-), Embase (Ovid Embase Classic + Embase 1947-) and the Cochrane Central Register of Controlled Trials (from inception). Search results were limited to English-only publications, and a publication date limit of January 1, 2019 to present was applied. In addition, a Canadian Agency for Drugs and Technologies in Health search filter was applied in MEDLINE and Embase to remove animal studies, and comment, newspaper article, editorial, letter and note publication types were also eliminated. Search results were exported to Covidence (Veritas Health Innovation, Melbourne, Australia) and duplicates were eliminated using the platform's duplicate identification feature.

Study selection, data extraction, and quality assessment

We included all types of COVID-19 clinical studies, including case series, observational studies, diagnostic studies, and RCTs. For diagnostic studies, the reference standard for COVID-19 diagnosis was considered a nasopharyngeal swab followed by reverse transcriptase polymerase chain reaction in order to detect SARS-CoV-2. We excluded studies that were exploratory or pre-clinical in nature (ie. *in vitro* or animal studies), case reports or case series of <5 patients, studies published in a language other than English, reviews, methods or protocols, and other coronavirus variants such as the Middle East Respiratory Syndrome.

Title and abstracts were evaluated by two independent reviewers (a team of internal medicine or cardiology fellow and a medical student) using Covidence and all discrepancies were resolved by consensus. Articles that were selected for full review were independently evaluated by two reviewers for quality assessment using a standardized case report form following the completion of a training period. When two independent reviewers were not able to come to a consensus score, a third reviewer (principal investigator) provided oversight in the process to resolve the conflicted scores.

First and corresponding author names, date of publication, title of manuscript, and journal of publication were collected for all included full-text articles. Journal impact factor was obtained from the 2018 InCites Journal Citation Reports from Clarivate Analytics. Submission and acceptance dates were collected in manuscripts when available. Other information such as study type, prospective or retrospective study, sex reporting, sample size calculation, method of SARS-CoV-2 diagnosis, and ethics approval was collected by the authors. Quality assessment was conducted using the Newcastle-Ottawa Scale (NOS) for case-control and cohort studies,⁷ QUADAS-2 tool for diagnostic studies,⁴ Cochrane Risk of Bias for RCTs,⁵ and a score derived by Murad et al. for case series were collected.⁶

Identification of historical control from identified COVID-19 articles

After full-text extraction of COVID-19 articles was completed, we obtained a historical control group by identifying reports of the same study design matched in a 1:1 fashion. From the eligible COVID-19 article, historical controls were identified by searching the same journal in a systematic fashion by matching for the same study design (“case series”, “cohort”, “case

control”, or “diagnostic”) starting in the journal edition 12 months prior to the COVID-19 article publication on the publisher website (ie. COVID-19 article published on April 2020, going backwards to April 2019) and proceeding forward (or backward if a specific article type was not identified) in a temporal fashion until the first matched study was identified following abstract screening by two independent reviewers. If no comparison article was found by either reviewers, the corresponding COVID-19 article was excluded from the analysis. Following the identification of the historical control, data extraction and quality assessment was conducted on the identified articles independently using the standardized case report forms by two reviewers and conflicts resolved by consensus. The full dataset has been made available in the **Supplemental Appendix**.

Data synthesis and statistical analysis

Continuous variables were reported as mean \pm SD or median \pm IQR as appropriate, and categorical variables were reported as proportions (%). Normally distributed continuous variables were compared using Mann-Whitney U-test and categorical variables and quality scores were compared by χ^2 , Fisher’s exact test, or Kruskal-Wallis test.

The primary outcome of interest was to evaluate the quality of COVID-19 by study type using Newcastle-Ottawa Scale (**NOS**) for case-control and cohort studies, QUADAS-2 tool for diagnostic studies,⁴ Cochrane Risk of Bias for RCTs,⁵ and a score derived by Murad et al. for case series.⁶ Pre-specified secondary outcomes were comparison of quality scores by: i) median time to acceptance, ii) impact factor, iii) geographical region, and iv) historical comparator. Time of acceptance was defined as the time between submission to acceptance which captures peer

review and editorial decisions. Geographical region was stratified into continents including Asia/Oceania, Europe/Africa, and Americas (North and South America). *Post-hoc* comparison analysis between COVID-19 and historical control article quality scores were evaluated using Kruskal-Wallis test. Furthermore, good quality of NOS was defined as 3+ on selection and 1+ on comparability and 2+ on outcome/exposure domains and high quality case series scores was defined as a score ≥ 3.5 . Due to a small sample size of identified RCTs, they were not included in the comparison analysis.

All statistical analyses were performed using SAS v9.4 (SAS Institute, Inc., Cary, NC, USA). Statistical significance was defined as $P < 0.05$. All figures were generated using GraphPad Prism v8 (GraphPad Software, La Jolla, CA, USA).”

6. Although I appreciate that the authors needed to dichotomize these tools into high vs low quality using a simplified score, the tools themselves were never designed for this purpose. This is a significant limitation and should be acknowledged. In fact, there are many that suggest a simplified score should never be applied to risk of bias assessments as this approach over-simplifies a very complex issue.

We thank the reviewer with regards to the limitation and improper categorization of the quality scores for the purpose of this analysis. Indeed, the rationale for dichotomizing all the used tools was to allow a comprehensive analysis of all the scores into one analysis (NOS, QUADAS-2, and the Murad tool). As such, we moved these dichotomized scores into supplemental figures and re-ran the analysis for each individual score and stated the limitation within the limitation section of the manuscript. Indeed, when the analysis was changed to ordinal scale (as to the original scores), we saw similar results to the dichotomization into high vs. low quality. Specifically, when stratified by study design, a mean (\pm SD) case series score of 3.3 ± 1.1 , 5.8 ± 1.5 for cohort studies, 5.5 ± 1.9 for case control studies, and low bias was present in 4 (6.4%) diagnostic studies. Furthermore, in the 6 COVID-19 related RCTS identified in the search, they remained at high risk of bias with minimal consideration for sequence generation, allocation

concealment, blinding, incomplete outcome data, and selective outcome reporting (**Table 2**). Furthermore, when stratified by pre-specified secondary outcomes of median time of acceptance (<13.0 days), journal impact factor, and geographical regions, we observed that rapid time from submission to acceptance resulted in lower study quality scores for case series and cohort study designs but not case-controls or diagnostic studies (**Figure 3A-D**). Furthermore, low journal impact factor was associated with lower study quality score in case series, cohort, and case-control designs (**Figure 3E-H**). COVID-19 originating in different geographical regions had no differences in study quality scores with the exception of cohort studies (**Figure 3I-L**). We transferred the dichotomized data to **Supplemental Figure 1** which revealed similar results.

When methodological quality scores were compared between COVID-19 and historical control articles (n=1078), we observed lower case series, cohort quality score, case-control score, and diagnostic study quality in COVID-19 articles compared to the historical control (**Table 2 and Figure 4**). The results section has been changed to reflect the new findings and is found on **pages 10-12** and below.

Results

Article selection

A total of 14787 COVID-19 papers were identified as of May 14, 2020 and 4892 duplicates were removed. 9895 titles and abstracts were screened, and 9101 articles were excluded due to the study being pre-clinical in nature, case report, case series <5 patients, in an language other than English, reviews (including systematic reviews), study protocols or methods, and other coronavirus variants with an overall inter-rater study inclusion agreement of 96.7%, similar to other evaluations performed by our group.^{1, 2, 8, 9} This left a final number of 794 full texts which were reviewed for eligibility. Over 108 articles were excluded for improper study design (such as letter to the editors, editorials, case reports, or case series <5 patients), patient population, non-English language, duplicates, wrong outcomes, and publication in a non-peer reviewed journal.

Ultimately, 686 articles were identified and underwent quality assessment with an inter-rater agreement of 86.5% ($\kappa=0.68$; 95% CI, 0.67-0.70) (**Figure 1**).

COVID-19 literature quality

Most studies originated from Asia/Oceania with 469 (68.4%) followed by Europe with 139 (20.3%), and the Americas with 78 (11.4%). Of included studies, 380 (55.4%) were case series, 199 (29.0%) were cohort, 63 (9.2%) were diagnostic, 38 (5.5%) were case-control, and 6 (0.9%) were RCTs. Most studies (590, 86.0%) were retrospective in nature, 620 (90.4%) reported the sex of patients, and 7 (1.0%) studies calculated their sample size *a priori*. The method of SARS-CoV-2 diagnosis was reported in 558 studies (81.3%) and ethics approval was obtained in 556 studies (81.0%). Finally, the median journal impact factor of COVID-19 manuscripts was 4.7 (IQR, 2.9-7.6) with a median time to acceptance of 13.0 (IQR, 5.0-25.0) days (**Table 1**).

Overall, when COVID-19 articles were stratified by study design, a mean case series score (out of 5) (\pm SD) of 3.3 ± 1.1 , mean NOS cohort study score (out of 8) of 5.8 ± 1.5 , mean NOS case control study score (out of 8) of 5.5 ± 1.9 , and low bias present in 4 (6.4%) diagnostic studies was observed (**Table 2 and Figure 2**). Furthermore, of the identified 6 RCTs in COVID-19 literature, they remain at high risk of bias with little consideration for sequence generation, allocation concealment, blinding, incomplete outcome data, and selective outcome reporting (**Table 2**).

For secondary outcomes, rapid time from submission to acceptance (defined as median time of acceptance of <13.0 days) was associated with lower study quality scores for case series and cohort study designs but not case-control or diagnostic studies (**Figure 3A-D**). Low journal

impact factor (<10) was associated with lower study quality scores for case series, cohort, and case-control designs (**Figure 3E-H**). Finally, studies originating from different geographical regions had no differences in study quality scores with the exception of cohort studies (**Figure 3I-L**). When dichotomized by good vs. low study quality scores, a similar trend was observed with rapid time from submission to acceptance (34.4% vs. 46.3%, $p=0.01$, **Supplemental Figure 1B**), low impact factor journals (<10) was associated with lower study quality score (38.8% vs 68.0%, $p<0.0001$, **Supplemental Figure 1C**). Finally, studies originating in either Americas or Asia/Oceania was associated with higher quality scores than Europe (**Supplemental Figure 1D**).

Methodological quality score differences in COVID-19 versus historical control

We matched 539 historical control articles to COVID-19 articles from the same journal with identical study designs in the previous year for a final analysis of 1078 articles (**Table 1**). Overall, 554 (51.4%), 348 (32.3%) cohort, 64 (5.9%) case-control, 106 (9.8%) diagnostic, and 6 (0.6%) RCTs were identified from the 1078 total articles. Overall, the median time of acceptance was 13.0 (IQR, 5.0-25.0) days in COVID-19 articles vs. 110.0 (IQR, 71.0-156.0) days in control articles (**Figure 4A**, $p<0.0001$). Case series quality score was lower in COVID-19 articles compared to the historical control (3.3 ± 1.1 vs. 4.3 ± 0.8 ; $p<0.0001$; **Table 2 and Figure 4B**). Furthermore, NOS score was lower in COVID-19 cohort studies (5.8 ± 1.6 vs. 7.1 ± 1.0 ; $p<0.0001$; **Table 2 and Figure 4C**) and case-control studies (5.4 ± 1.9 vs. 6.6 ± 1.0 ; $p=0.003$; **Table 2 and Figure 4D**). Finally, high diagnostic study quality was observed 12 COVID-19 articles (22.6%) vs. 24 control articles (45.3%, **Table 2 and Figure 4E**, $p=0.02$). A similar trend was observed when dichotomized by good vs. low quality scores (**Supplemental Figure 2**).”

7. Data for individual studies assessed must be submitted for review. It was impossible for me to assess the quality of the study extraction without this granular data to review. I would encourage the authors to also provide this as supplemental online data for others to review when published.

We have agreed to deposit the dataset and initiated the process for this but it has not been completed at the time of the review process. We have attached the entire dataset as a supplemental excel file along for your review.

8. The study appears to be either retrospectively registered, or incorrectly registered at best. This is a significant issue.

a) In PROSPERO it is stated that preliminary searches were not complete as of June 3, 2020. Yet the final search for this study was completed in May.

We thank the reviewer in evaluating the PROSPERO registration (CRD42020187318). The reviewer is correct in that the literature search was conducted on May 14, 2020 and the start date of the systematic review as per the registration with screening done on Covidence after assembling the systematic review team. Importantly, the registration was not retroactively registered, and the search strategy and analysis plan was designed *a priori*. The reviewer is correct in that although the preliminary searches, piloting, and formal screening was started when the study was registered on PROSPERO on June 3rd, we made an oversight in not also selecting the completed boxes as this was our first submission to PROSPERO.

b) What else was complete prior to registration? This must be reported.

At the time of original submission on June 3, 2020, we completed the preliminary searches, piloting of the study selection process through Covidence, and formalized screening of the search results against our eligibility criteria. As per PROSPERO protocol, no data extraction, risk of bias assessment, or data analysis was completed at this time prior to registration as these protocols are not eligible for inclusion on the website. When the study was initially registered, the time to registration was slightly delayed as the PROSPERO administrators required us to

specify “at least one outcome of direct patient or clinical relevance” which was clarified leading to subsequent registration.

9. The analysis plan (especially with regards to regression that was presented) was not prespecified. This should be clearly labelled as post-hoc.

We thank the reviewer in rightfully specifying that our regression analysis was performed as a *post-hoc* analysis which was not pre-specified to adjust for variables which were associated with the quality scores. The statistical analysis plan which was pre-specified included the evaluation of COVID-19 articles using the NOS, QUADAS-2 tool, Cochrane Risk of Bias, and the Murad tool for case series. Furthermore, pre-specified secondary outcomes include comparison of quality scores by the median time to acceptance, impact factor, geographical region, and historical comparators. *Post-hoc* categorization of geographical regions into continental regions including Asia/Oceania, Europe/Africa, and Americas was conducted, but took place prior to data analysis by quality scores. Furthermore, the dichotomization of good vs. poor quality manuscripts was not pre-specified in the PROSPERO protocol but took place prior to data analysis being performed. However, due to the excellent recommendation in Q6 by the reviewer, we placed all the dichotomized quality data into Supplemental Figures and re-ran the analysis (now *post-hoc*) by ordinal scales and conducted Kruskal-Wallis tests. Finally, we have removed the regression analysis as per the recommendation of all three reviewers.

10. The cut-offs for good vs poor quality were not prespecified in the protocol. These should be clearly reported as post-hoc.

As question #9, the reviewer is correct that the dichotomization from good vs. poor quality was not pre-specified in the protocol. However, as per question #6, we have changed the analysis into analyzing ordinal scales instead of dichotomization. We respectfully disagree that the dichotomization was done *post-hoc*, as quality analysis for dichotomization was specified prior to formally evaluating the data using our statistical software (SAS v9.4). Since the analysis was changed from dichotomized data to ordinal scale as per the recommendation of the reviewer, we have specified this *post-hoc* analysis within our statistical analysis section and is found on **pages 8-9** and below.

“Data synthesis and statistical analysis

Continuous variables were reported as mean \pm SD or median \pm IQR as appropriate, and categorical variables were reported as proportions (%). Normally distributed continuous variables were compared using Mann-Whitney U-test and categorical variables and quality scores were compared by χ^2 , Fisher’s exact test, or Kruskal-Wallis test.

The primary outcome of interest was to evaluate the quality of COVID-19 by study type using Newcastle-Ottawa Scale (NOS) for case-control and cohort studies, QUADAS-2 tool for diagnostic studies,⁴ Cochrane Risk of Bias for RCTs,⁵ and a score derived by Murad et al. for case series.⁶ Pre-specified secondary outcomes were comparison of quality scores by: i) median time to acceptance, ii) impact factor, iii) geographical region, and iv) historical comparator. Time of acceptance was defined as the time between submission to acceptance which captures peer review and editorial decisions. Geographical region was stratified into continents including Asia/Oceania, Europe/Africa, and Americas (North and South America). *Post-hoc* comparison analysis between COVID-19 and historical control article quality scores were evaluated using Kruskal-Wallis test. Furthermore, good quality of NOS was defined as 3+ on selection and 1+ on comparability and 2+ on outcome/exposure domains and high quality case series scores was defined as a score ≥ 3.5 . Due to a small sample size of identified RCTs, they were not included in the comparison analysis.

All statistical analyses were performed using SAS v9.4 (SAS Institute, Inc., Cary, NC, USA). Statistical significance was defined as $P < 0.05$. All figures were generated using GraphPad Prism v8 (GraphPad Software, La Jolla, CA, USA).”

11. The statistical tests themselves seemed appropriate (they were all generally simple ones I am familiar with), however I am not a statistician and would defer to another reviewer with greater statistical expertise.

We thank the reviewer for the question – we agree that the statistical tests used for the manuscript were simple to answer the question of whether a difference in quality scores exist between COVID-19 and historical control articles.

12. It is unclear why a 1:1 case to control ratio was used. Typically, 1:4 is thought to be optimal. For associative studies 1:2 could be argued. 1:1 must be rationalized and the potential limitations of this should be discussed.

The reviewer raises an interesting point with regards to the ratio of case to control chosen for the purpose of evaluating the quality of COVID-19 compared to the historical control articles. Indeed, the use of 1:1 ratio in the manuscript was chosen for three main reasons: 1) large effect size between COVID-19 and control articles (effect size of 78.8% relative to COVID-19 controls), 2) due to the large sample size (n=539 in each arm), and 3) limitations in number of historical controls available in each identified journal within the restricted time period.

The reviewer is correct in that an increase from 1:1 to 1:4 ratio increases the precision of the point estimates generated and minimizes bias in treatment effect estimates,¹⁰ with diminishing returns when the case to control ratio exceeds 1:4.¹¹ When carefully examining the Hamajima paper, an increase from 1:1 to 1:4 matched controls overall did not influence the change in the point estimate (OR 2.71 to 2.59 from 1:1 to 1:4 ratio, respectively) but tightened the confidence interval (due to increased sample size). Our study demonstrates a relative change of 78.8% with absolute differences of 32.3% between COVID-19 and historical comparator cohort and the increase in sampling will not change the overall takeaway of the manuscript.

Secondly, the reason for the fixed 1:1 ratio matching remains limited by the number of historical controls available in each identified journal without duplication of historical controls. Indeed, several journals published multiple COVID-19 manuscripts in this time period (ie. Journal of Clinical Virology and Journal of Medical Virology) which restricted the number of available control manuscripts available. We opted to maximize the characteristics of the first matched

control without introducing biases due to additional matched controls which are less comparable to the original COVID-19 article.¹⁰ Indeed, we originally lost 147 articles due to the lack of an appropriate historical control and setting a fixed 1:2 or 1:4 ratio would cause a greater dropout in the total sample size which in its own will affect the point estimate and 95% confidence interval obtained in the results along with the generalizability of the ascertained results.

Propensity matching and other methods of matching in observational studies are often done in order to balance the effect of potential known confounding variable on the outcome of interest. Indeed, we determined *a priori* that identifying controls in the same journal by the same study design allow us to account for potential confounding variables which potentially influences manuscript quality (ie. similar editorial board and peer-reviewer rigor). More often than not, propensity-score matching techniques commonly utilize 1:1 matching methods.^{10, 12}

We hope that the reviewer agrees with our rationale for selection of the 1:1 match ratio due to the overwhelming effect size generated, large sample size, and the logistics related to maintaining a large sample size without further losses of COVID-19 articles. We have added the potential limitation of the 1:1 match ratio in our limitation section and is found on **pages 14-15** and below.

“Our study has important limitations. We evaluated the methodological quality of existing studies using established quality scores. While it is tempting to associate quality scores with reproducibility or causal inferences, it is not possible to ascertain the impact on the study design and conduct of research nor results or conclusions in the identified reports.¹³ Second, although the methodological quality scales and checklists used for the manuscript are commonly used for quality assessment in systematic reviews and meta-analyses,^{4, 5, 6, 7} they can only assess the methodology without consideration for causal language and prone to limitations.^{14, 15} Furthermore, other considerations of quality such as sample size calculation, sex reporting, or ethics approval is not considered in these quality scores. As such, the quality scores measured using these checklists only reflect the patient selection, comparability, diagnostic reference

standard, and methods to ascertain the outcome of the study. Third, the 1:1 ratio to identify our historical control articles may affect the precision estimates of our findings. Interestingly, a simulation of an increase from 1:1 to 1:4 control ratio tightened the precision estimates but did not significantly alter the point estimate.¹¹ Furthermore, the decision for 1:1 ratio exists due to limitations of available historical control articles from identical journal in the restricted time period combined with large effect size and sample size in the analysis. Finally, our analysis includes early publications on COVID-19 and there is likely to be an improvement in quality of related studies and study design as the field matures and higher quality studies, which take longer to design, conduct, and report are published. Accordingly, our findings are limited to the early body of research as it pertains to the pandemic and it is likely that over time research quality will improve.”

13. Who conducted the search? If an information specialist was involved this should be stated and they should be acknowledged.

We thank the reviewer for clarification on the search strategy, it was designed by our librarian, Sarah Visintini, who is a systematic review search strategy specialist and co-author in our manuscript and her initials have been added in the methods and is found on **page 6** and below.

“Data sources and searches

The search was created in MEDLINE by a medical librarian with expertise in systematic reviews (S.V.) using a combination of key terms and index headings related to COVID-19 and translated to the remaining bibliographic databases (**Supplemental Tables 1-3**). The searches were conducted in MEDLINE (Ovid MEDLINE(R) ALL 1946-), Embase (Ovid Embase Classic + Embase 1947-) and the Cochrane Central Register of Controlled Trials (from inception). Search results were limited to English-only publications, and a publication date limit of January 1, 2019

to present was applied. In addition, a Canadian Agency for Drugs and Technologies in Health search filter was applied in MEDLINE and Embase to remove animal studies, and comment, newspaper article, editorial, letter and note publication types were also eliminated. Search results were exported to Covidence (Veritas Health Innovation, Melbourne, Australia) and duplicates were eliminated using the platform's duplicate identification feature.”

14. Was the search peer reviewed by another information specialist (i.e. Peer Review of Electronic Search Strategy, PRESS)?

Unfortunately, due to a combination of the COVID-19 pandemic affecting normal library operations, time considerations related to the project, and simplicity of the search strategy, we did not have another information specialist conducting a PRESS. Furthermore, the medical librarian (S.V.) consulted numerous colleagues regarding the search strategy designed for the COVID-19 literature search strategy.

15. Full search strategies for all databases should be included (I saw only MEDLINE included in the supplemental).

We have attached the search strategies for MEDLINE, Embase, and Cochrane Central Register of Controlled Trials as Supplemental Tables in the manuscript and below.

16. The search filter from the Canadian Agency for Drugs and Technology in Health should be cited. I'm not entirely clear which filter this was.

We agree with the reviewer that the Canadian Agency for Drugs and Technology in Health (CADTH) filter should be cited. Unfortunately, the filter was not published anywhere as a standalone filter but is used in-house routinely by CADTH information specialists which was provided by the agency with permission to reuse the strategy itself.

17. What was the concordance between reviewers for study selection (e.g. interrater

reliability). Was there piloting of screening and selection of studies? Similarly, was their piloting for study data extraction?

The reviewer asks an excellent question regarding the concordance between reviewers for the initial study selection by title and abstract screening. From 9895 articles, the reviewers had a conflict in 330 texts which were resolved by consensus (96.7% agreement rate). Unfortunately, Covidence does not keep track of which reviewer voted yes and no and we are unable to calculate a Kappa-value for this aspect of study selection. On a similar theme to question 5, each reviewer initially were trained on the inclusion and exclusion criteria prior to initiation of screening of titles and abstracts. Following this, each reviewer independently reviewed 100 texts (roughly 1% of the captured papers from the search strategy) and the two reviewers discussed any conflicts which may have arisen in order to avoid systematic biases. Furthermore, independent review was conducted by two reviewers, which consisted of a trained research staff with expertise in systematic reviews/quality assessment and one research trainee. After the completion of the screening sampling, remaining titles and abstracts were screened independently. Furthermore, when two independent reviewers could not come to a consensus in the score ascertained, a third reviewer, the principal investigator of the study provided oversight in the process to resolve the conflicted scores.

A similar piloting process was captured in study data extraction as discussed on question 5. In order to address content expertise, we first trained all reviewers with the original manuscript which determined and evaluated the scores along with checklists and examples for what were deemed high scores. Furthermore, when independent review was conducted by two reviewers, the two reviewers consisted of an internal medicine resident or cardiology fellow with a medical student for content expertise. After the completion of thirty full-text extractions, the two reviewers had met in order to come to consensus and clarify miscommunication and misunderstanding of clinical concepts prior to completing the remaining manuscripts independently. Furthermore, when two independent reviewers could not come to a consensus in the score ascertained, a third reviewer, the principal investigator of the study provided oversight in the process to resolve the conflicted scores. Finally, the extractors ability to assess data extraction was evaluated by calculating Kappa values following the completion of data

extraction. For full text extraction and quality assessment, the general agreement between the reviewers for quality assessment was 86.5% ($\kappa=0.68$; 95% CI, 0.67-0.70).

We have added both the inter-rater agreement values for both the title and abstract screening process and full data extraction in the manuscript is found on **page 10** and below.

Results

Article selection

A total of 14787 COVID-19 papers were identified as of May 14, 2020 and 4892 duplicates were removed. 9895 titles and abstracts were screened, and 9101 articles were excluded due to the study being pre-clinical in nature, case report, case series <5 patients, in an language other than English, reviews (including systematic reviews), study protocols or methods, and other coronavirus variants with an overall inter-rater study inclusion agreement of 96.7%, similar to other evaluations performed by our group.^{1, 2, 8, 9} This left a final number of 794 full texts which were reviewed for eligibility. Over 108 articles were excluded for improper study design (such as letter to the editors, editorials, case reports, or case series <5 patients), patient population, non-English language, duplicates, wrong outcomes, and publication in a non-peer reviewed journal. Ultimately, 686 articles were identified and underwent quality assessment with an inter-rater agreement of 86.5% ($\kappa=0.68$; 95% CI, 0.67-0.70) (**Figure 1**).”

18. The first three boxes in the PRISMA flow diagram do not add up (i.e. records identified minus duplicates does not equal records screened).

The PRISMA flow diagram identified as of May 14, 2020, 14787 articles with 4892 duplicates which were removed to result in 9895 records to be screened. The numbers have now been updated accordingly in the manuscript and may be found on **Figure 1** and on **page 10** and below.

“Results

Article selection

A total of 14787 COVID-19 papers were identified as of May 14, 2020 and 4892 duplicates were removed. 9895 titles and abstracts were screened, and 9101 articles were excluded due to the study being pre-clinical in nature, case report, case series <5 patients, in an language other than English, reviews (including systematic reviews), study protocols or methods, and other coronavirus variants with an overall inter-rater study inclusion agreement of 96.7%, similar to other evaluations performed by our group.^{1, 2, 8, 9} This left a final number of 794 full texts which were reviewed for eligibility. Over 108 articles were excluded for improper study design (such as letter to the editors, editorials, case reports, or case series <5 patients), patient population, non-English language, duplicates, wrong outcomes, and publication in a non-peer reviewed journal. Ultimately, 686 articles were identified and underwent quality assessment with an inter-rater agreement of 86.5% ($\kappa=0.68$; 95% CI, 0.67-0.70) (**Figure 1**).”

19. Why are no details provided for the 6 RCTs identified? I realize these represents only a tiny fraction of the studies identified, however the details could be provided as online supplemental material at the very least. Also, related to this, the cut-off for high vs low quality using the RoB tool wasn’t stated in the methods.

Indeed, the reviewer is correct we failed to provide information of the six COVID-19 related RCTs identified in our search strategy. The risk of bias assessment for the six COVID-19 RCTs are now available on **Table 2**. Overall, 3 (50.0%) had low bias for sequence generation, 2 (33.3%) had considered allocation concealment, 1 (16.7%) blinded participants and personnel to outcomes, 1 (16.7%) blinded outcome assessors for the outcomes, 3 (50.0%) considered incomplete outcome data for all outcomes, 5 (83.3%) had selective outcome reporting, and 2 (33.3%) had other sources of biases (**Table 2**). Among the 6 COVID-19 RCTs, 3 historical controls were identified with the corresponding the risk of bias present on **Table 2**. We removed the RCTs from comparison analysis for three reasons: 1) Cochrane recommends against the use

of the Risk of Bias in summary scores which limits our analysis, 2) The risk of bias tool is unamenable to a collective score, and most importantly 3) the small sample size to detect any difference among the two cohorts. The information has been updated in our results section and is found on **pages 10-11** and below.

“COVID-19 literature quality

Most studies originated from Asia/Oceania with 469 (68.4%) followed by Europe with 139 (20.3%), and the Americas with 78 (11.4%). Of included studies, 380 (55.4%) were case series, 199 (29.0%) were cohort, 63 (9.2%) were diagnostic, 38 (5.5%) were case-control, and 6 (0.9%) were RCTs. Most studies (590, 86.0%) were retrospective in nature, 620 (90.4%) reported the sex of patients, and 7 (1.0%) studies calculated their sample size *a priori*. The method of SARS-CoV-2 diagnosis was reported in 558 studies (81.3%) and ethics approval was obtained in 556 studies (81.0%). Finally, the median journal impact factor of COVID-19 manuscripts was 4.7 (IQR, 2.9-7.6) with a median time to acceptance of 13.0 (IQR, 5.0-25.0) days (**Table 1**).

Overall, when COVID-19 articles were stratified by study design, a mean case series score (out of 5) (\pm SD) of 3.3 ± 1.1 , mean NOS cohort study score (out of 8) of 5.8 ± 1.5 , mean NOS case control study score (out of 8) of 5.5 ± 1.9 , and low bias present in 4 (6.4%) diagnostic studies was observed (**Table 2 and Figure 2**). Furthermore, of the identified 6 RCTs in COVID-19 literature, they remain at high risk of bias with little consideration for sequence generation, allocation concealment, blinding, incomplete outcome data, and selective outcome reporting (**Table 2**).”

MINOR:

1. Covidence, on first use, should describe source and location.

We have updated the source and location of the systematic review screening software utilized for the study on the first use and may be found on **page 6** and below.

“Data sources and searches

The search was created in MEDLINE by a medical librarian with expertise in systematic reviews (S.V.) using a combination of key terms and index headings related to COVID-19 and translated to the remaining bibliographic databases (**Supplemental Tables 1-3**). The searches were conducted in MEDLINE (Ovid MEDLINE(R) ALL 1946-), Embase (Ovid Embase Classic + Embase 1947-) and the Cochrane Central Register of Controlled Trials (from inception). Search results were limited to English-only publications, and a publication date limit of January 1, 2019 to present was applied. In addition, a Canadian Agency for Drugs and Technologies in Health search filter was applied in MEDLINE and Embase to remove animal studies, and comment, newspaper article, editorial, letter and note publication types were also eliminated. Search results were exported to **Covidence (Veritas Health Innovation, Melbourne, Australia)** and duplicates were eliminated using the platform’s duplicate identification feature.”

2. The PRISMA guidelines are reporting guidelines, not conduct guidelines. Thus, it is incorrect to say this study was ‘conducted according to the (PRISMA) guidelines.’ I realize that this is stated in many systematic reviews, but it is simply incorrect.

We thank the reviewer for this point as we certainly were not immune to this mistake and made the appropriate changes in the manuscript to reflect the reporting guidelines and is found on **page 5** and below.

Methods

A systematic literature search was conducted on May 14, 2020 (**PROSPERO: CRD42020187318**) and **reported according to the Preferred Reporting Items for Systematic Reviews and Meta-Analyses**. Furthermore, the cohort study was reported according to the

Strengthening The Reporting of Observational studies in Epidemiology checklist. The data supporting the findings of this study is available in the Supplemental Appendix.”

Reviewer #2 (Remarks to the Author):

This paper evaluates the quality of the clinical literature on COVID-19 published up to 14th May 2020, both in absolute terms and relative to control papers matches on study type and journal. Its key finding is that the COVID papers have substantially lower quality than matched controls. The authors have assessed the quality of 686 COVID papers and 539 controls so this is an enormous piece of work.

Ironically, the paper itself bears the marks of hasty preparation for publication. Many of my comments can be addressed in a revision. I think the crucial issue is to identify the limitation of this work (3a below) that it cannot say whether the trade-off that we have made (between quality and speed) was appropriate.

We thank the reviewer for taking the time to review the manuscript along with kind words and supportive suggestions in greatly improving the quality of the manuscript. We hope that we addressed the following concerns below.

Major points

1. Methods.

a. Study type is defined inconsistently. At 176-77 it is defined as case series, observational studies, diagnostic studies, and RCTs, but in the figures the RCTs disappear and the observational studies are subdivided. Further, the results for RCTs are never reported (only the fact that there were 6 RCTs).

The reviewer is correct in that we failed to provide information of the six COVID-19 related RCTs identified in our search strategy. We removed the RCTs from the comparison analysis for three reasons: 1) Cochrane recommends against the use of the Risk of Bias in summary scores which limits our analysis, 2) The risk of bias tool is unamenable to a collective score, and most importantly 3) the small sample size to detect any difference among the two cohorts. The pertinent information for the risk of bias assessment for the six COVID-19 RCTs are now available on **Table 2**. Overall, 3 (50.0%) had low bias for sequence generation, 2 (33.3%) had considered allocation concealment, 1 (16.7%) blinded participants and personnel to outcomes, 1 (16.7%) blinded outcome assessors for the outcomes, 3 (50.0%) considered incomplete outcome data for all outcomes, 5 (83.3%) had selective outcome reporting, and 2 (33.3%) had other sources of biases (**Table 2**). Among the 6 COVID-19 RCTs, 3 historical controls were identified with the corresponding the risk of bias present on **Table 2**. Our results has been updated to reflect the above changes and is found on **page 11** and below.

“Overall, when COVID-19 articles were stratified by study design, a mean case series score (out of 5) (\pm SD) of 3.3 ± 1.1 , mean NOS cohort study score (out of 8) of 5.8 ± 1.5 , mean NOS case control study score (out of 8) of 5.5 ± 1.9 , and low bias present in 4 (6.4%) diagnostic studies was observed (**Table 2 and Figure 2**). Furthermore, of the identified 6 RCTs in COVID-19 literature, they remain at high risk of bias with little consideration for sequence generation, allocation concealment, blinding, incomplete outcome data, and selective outcome reporting (**Table 2**).”

b. I was unclear why over 9000 of the 9895 COVID-19 papers identified were rejected. Was a sample used to reduce workload? If so this must be stated.

The reviewer asks an excellent question as to why 9101 articles were excluded during the title and abstract screening phase by two independent reviewers. The rationale behind the exclusion was due to the following reasons: 1) study was preclinical or exploratory in nature (ie. *in vitro* or animal study), 2) case report, 3) case series <5 patients, 4) published in language other than English, 5) reviews including systematic reviews, 6) study protocol or methods, and 7) coronavirus variants other than SARS-CoV-2. A large number of studies were narrative in nature (ie. reviews) with editorial comments or guidelines which was not the focus of this evaluation. Furthermore, pre-clinical studies are evaluated for methodological quality using different checklists.^{1,2} The information has been updated in both **Figure 1** and the results section of the manuscript and is found on **page 10** and below.

Results

Article selection

A total of 14787 COVID-19 papers were identified as of May 14, 2020 and 4892 duplicates were removed. 9895 titles and abstracts were screened, and 9101 articles were excluded due to the study being pre-clinical in nature, case report, case series <5 patients, in an language other than English, reviews (including systematic reviews), study protocols or methods, and other

coronavirus variants with an overall inter-rater study inclusion agreement of 96.7%, similar to other evaluations performed by our group.^{1, 2, 8, 9} This left a final number of 794 full texts which were reviewed for eligibility. Over 108 articles were excluded for improper study design (such as letter to the editors, editorials, case reports, or case series <5 patients), patient population, non-English language, duplicates, wrong outcomes, and publication in a non-peer reviewed journal. Ultimately, 686 articles were identified and underwent quality assessment with an inter-rater agreement of 86.5% ($\kappa=0.68$; 95% CI, 0.67-0.70) (**Figure 1**).”

c. The quality of the assessment of quality is crucial. It would be useful to see measures of agreement between the two reviewers who assessed quality. What measures were taken to apply comparable standard to the COVID-19 and control papers?

The reviewer asks an excellent question regarding the quality of the assessment of the literature between the study reviewers. Each reviewer initially were trained on the inclusion and exclusion criteria by the study leads prior to initiation of screening of titles and abstracts. Following this, each reviewer independently reviewed 100 texts (roughly 1% of the captured papers from the search strategy) and the two reviewers discussed any conflicts which may have arisen in order to avoid systematic biases. Furthermore, independent review was conducted by two reviewers, which consisted of a trained research staff with expertise in systematic reviews/quality assessment and one research trainee. After the completion of the screening sampling, remaining titles and abstracts were screened independently. Furthermore, when two independent reviewers could not come to a consensus in the score ascertained, a third reviewer, the principal investigator of the study provided oversight in the process to resolve the conflicted scores. From 9895 title and abstract screens, the reviewers had a conflict in 330 texts which were resolved by consensus (96.7% agreement rate). Unfortunately, Covidence does not keep track of the specifics of the disagreements and we are unable to calculate a corresponding kappa value.

A similar piloting process was captured in study data extraction as the title and abstract screening process. In order to address content expertise, we first trained all reviewers with the original manuscripts used to develop the quality scales/checklists used for the study along with examples

for what were deemed high scores. Similarly, independent review was conducted by two reviewers, which consisted of a trained research staff with expertise in systematic reviews/quality assessment and one research trainee. After the completion of thirty full-text extractions, the two reviewers had met in order to come to consensus and clarify miscommunication and misunderstanding of clinical concepts prior to completing the remaining manuscripts independently. When two independent reviewers could not come to a consensus in the score ascertained, a third reviewer, the principal investigator of the study provided oversight in the process to resolve the conflicted scores. Finally, the extractors ability to assess data extraction was evaluated by calculating Kappa values following the completion of data extraction. For full text extraction and quality assessment, the general agreement between the reviewers for quality assessment was 86.5% ($\kappa=0.68$; 95% CI, 0.67-0.70), which correlates to substantial agreement between the reviewers.

We have added both the inter-rater agreement values for both the title and abstract screening process and full data extraction in the manuscript is found on **page 10** and below.

Results

Article selection

A total of 14787 COVID-19 papers were identified as of May 14, 2020 and 4892 duplicates were removed. 9895 titles and abstracts were screened, and 9101 articles were excluded due to the study being pre-clinical in nature, case report, case series <5 patients, in an language other than English, reviews (including systematic reviews), study protocols or methods, and other coronavirus variants with an overall inter-rater study inclusion agreement of 96.7%, similar to other evaluations performed by our group.^{1, 2, 8, 9} This left a final number of 794 full texts which were reviewed for eligibility. Over 108 articles were excluded for improper study design (such as letter to the editors, editorials, case reports, or case series <5 patients), patient population, non-English language, duplicates, wrong outcomes, and publication in a non-peer reviewed journal.

Ultimately, 686 articles were identified and underwent quality assessment with an inter-rater agreement of 86.5% ($\kappa=0.68$; 95% CI, 0.67-0.70) (**Figure 1**).”

2. Reporting.

a. The results section includes a lot of repetition of the tables and figures – e.g. the para at l1148-161 almost exactly repeats F3. This is undesirable. Conversely some text describes findings that are not in the figures at all, e.g. the text at l1128-135 does NOT match what is reported in F1.

We apologize for the repetition and mistakes that were found in lines 128-135 and 148-161 along with Figures 1 and 3. We have carefully went over the results along with the **Figures** to ensure that the study selection flow diagram matches what is found on **Figure 1** along with presenting the inter-rater agreement values which was requested by the reviewer. Furthermore, **Figures 2-4** has been changed to remove the dichotomized quality scores to ordinal scales with the results reflecting the general findings of these **Figures**.

Indeed, when the analysis was changed to ordinal scale (as to the original scores), we saw similar results to the dichotomization into high vs. low quality. Specifically, when stratified by study design, a mean (\pm SD) case series score of 3.3 ± 1.1 , 5.8 ± 1.5 for cohort studies, 5.5 ± 1.9 for case control studies, and low bias was present in 4 (6.4%) diagnostic studies. Furthermore, in the 6 COVID-19 related RCTS identified in the search, they remained at high risk of bias with minimal consideration for sequence generation, allocation concealment, blinding, incomplete outcome data, and selective outcome reporting (**Table 2**). Furthermore, when stratified by pre-specified secondary outcomes of median time of acceptance (<13.0 days), journal impact factor, and geographical regions, we observed that rapid time from submission to acceptance resulted in lower study quality scores for case series and cohort study designs but not case-controls or diagnostic studies (**Figure 3A-D**). Furthermore, low journal impact factor lowered study quality score in case series, cohort, and case-control designs (**Figure 3E-H**). COVID-19 originating in different geographical regions had no differences in study quality scores with the exception of cohort studies (**Figure 3I-L**). We transferred the dichotomized data to **Supplemental Figure 1** which revealed similar results.

When methodological quality scores were compared between COVID-19 and historical control articles (n=1078), we observed lower case series, cohort quality score, case-control score, and diagnostic study quality in COVID-19 articles compared to the historical control (**Table 2 and Figure 4**). The results section has been changed to reflect the new findings and is found on **pages 10-12** and below.

Results

Article selection

A total of 14787 COVID-19 papers were identified as of May 14, 2020 and 4892 duplicates were removed. 9895 titles and abstracts were screened, and 9101 articles were excluded due to the study being pre-clinical in nature, case report, case series <5 patients, in an language other than English, reviews (including systematic reviews), study protocols or methods, and other coronavirus variants with an overall inter-rater study inclusion agreement of 96.7%, similar to other evaluations performed by our group.^{1, 2, 8, 9} This left a final number of 794 full texts which were reviewed for eligibility. Over 108 articles were excluded for improper study design (such as letter to the editors, editorials, case reports, or case series <5 patients), patient population, non-English language, duplicates, wrong outcomes, and publication in a non-peer reviewed journal. Ultimately, 686 articles were identified and underwent quality assessment with an inter-rater agreement of 86.5% ($\kappa=0.68$; 95% CI, 0.67-0.70) (**Figure 1**).

COVID-19 literature quality

Most studies originated from Asia/Oceania with 469 (68.4%) followed by Europe with 139 (20.3%), and the Americas with 78 (11.4%). Of included studies, 380 (55.4%) were case series, 199 (29.0%) were cohort, 63 (9.2%) were diagnostic, 38 (5.5%) were case-control, and 6 (0.9%) were RCTs. Most studies (590, 86.0%) were retrospective in nature, 620 (90.4%) reported the

sex of patients, and 7 (1.0%) studies calculated their sample size *a priori*. The method of SARS-CoV-2 diagnosis was reported in 558 studies (81.3%) and ethics approval was obtained in 556 studies (81.0%). Finally, the median journal impact factor of COVID-19 manuscripts was 4.7 (IQR, 2.9-7.6) with a median time to acceptance of 13.0 (IQR, 5.0-25.0) days (**Table 1**).

Overall, when COVID-19 articles were stratified by study design, a mean case series score (out of 5) (\pm SD) of 3.3 ± 1.1 , mean NOS cohort study score (out of 8) of 5.8 ± 1.5 , mean NOS case control study score (out of 8) of 5.5 ± 1.9 , and low bias present in 4 (6.4%) diagnostic studies was observed (**Table 2 and Figure 2**). Furthermore, of the identified 6 RCTs in COVID-19 literature, they remain at high risk of bias with little consideration for sequence generation, allocation concealment, blinding, incomplete outcome data, and selective outcome reporting (**Table 2**).

For secondary outcomes, rapid time from submission to acceptance (defined as median time of acceptance of <13.0 days) was associated with lower study quality scores for case series and cohort study designs but not case-control or diagnostic studies (**Figure 3A-D**). Low journal impact factor (<10) was associated with lower study quality scores for case series, cohort, and case-control designs (**Figure 3E-H**). Finally, studies originating from different geographical regions had no differences in study quality scores with the exception of cohort studies (**Figure 3I-L**). When dichotomized by good vs. low study quality scores, a similar trend was observed with rapid time from submission to acceptance (34.4% vs. 46.3%, $p=0.01$, **Supplemental Figure 1B**), low impact factor journals (<10) was associated with lower study quality score (38.8% vs 68.0%, $p<0.0001$, **Supplemental Figure 1C**). Finally, studies originating in either

Americas or Asia/Oceania was associated with higher quality scores than Europe (**Supplemental Figure 1D**).

Methodological quality score differences in COVID-19 versus historical control

We matched 539 historical control articles to COVID-19 articles from the same journal with identical study designs in the previous year for a final analysis of 1078 articles (**Table 1**). Overall, 554 (51.4%), 348 (32.3%) cohort, 64 (5.9%) case-control, 106 (9.8%) diagnostic, and 6 (0.6%) RCTs were identified from the 1078 total articles. Overall, the median time of acceptance was 13.0 (IQR, 5.0-25.0) days in COVID-19 articles vs. 110.0 (IQR, 71.0-156.0) days in control articles (**Figure 4A**, $p < 0.0001$). Case series quality score was lower in COVID-19 articles compared to the historical control (3.3 ± 1.1 vs. 4.3 ± 0.8 ; $p < 0.0001$; **Table 2 and Figure 4B**). Furthermore, NOS score was lower in COVID-19 cohort studies (5.8 ± 1.6 vs. 7.1 ± 1.0 ; $p < 0.0001$; **Table 2 and Figure 4C**) and case-control studies (5.4 ± 1.9 vs. 6.6 ± 1.0 ; $p = 0.003$; **Table 2 and Figure 4D**). Finally, high diagnostic study quality was observed 12 COVID-19 articles (22.6%) vs. 24 control articles (45.3%, **Table 2 and Figure 4E**, $p = 0.02$). A similar trend was observed when dichotomized by good vs. low quality scores (**Supplemental Figure 2**).”

b. T2 reports the association of other covariates (e.g. high impact factor) with methodological quality. I don't think this is relevant to the aims of this paper and therefore I think it should be dropped. Also, high methodological quality should be included in table 1. If table 2 is kept as it is then please note that COVID-19 vs control is not a journal characteristic and the footnote does not make sense since regression adjustment applies to coefficients, not to articles.

We thank the reviewer for the insightful comment regarding the logistic regression conducted to further elucidate the association of other variables with methodological quality. Indeed, this analysis is not the primary aim of the manuscript and has been removed. Instead, we have separated the pre-specified secondary outcomes into **Figure 3** (previously **Figure 2B-D**) and

created four figures to represent our findings along with two tables revealing the overall study characteristics and quality scores.

3. Interpretation.

a. A lower methodological quality and faster time to publication has been demonstrated for COVID-related papers compared with controls. However the present evidence does not show that this was a bad thing. The authors give good arguments in the discussion that the low quality has been a problem, but they do not discuss how slower time to publication might also have been a problem.

The reviewer is correct in that we have demonstrated reduced methodological quality scores and faster time of publication in COVID-19 related manuscripts compared to historical controls. The reviewer however is incorrect in that “the present evidence does not show that this was a bad thing”. Careful examination of **Figure 3A-D** demonstrates that increased time of acceptance using median TOA as the cut-off within COVID-19 manuscripts reveal differences in the overall score in both the case series score and Newcastle Ottawa Scale cohort study score ($p=0.02$ and $p=0.0003$, respectively). Although no differences were observed in the NOS case-control score and diagnostic risk of bias (QUADAS-2) when stratified by median TOA, this is an underpowered subgroup analysis due to small sample size in these categories.

Within the discussion, we highlight our findings that low-quality scores has been a problem in COVID-19 literature. Furthermore, we have updated our discussion to highlight in paragraph 3 that the urgent need to inform the public led to an increased understanding of the natural history of COVID-19 and identification of diagnostic tools to diagnose SARS-CoV-2 (such as the nasopharyngeal swab followed by RT-PCR). We do warn however that despite the initial expedition to publish to understand the disease, the data obtained in these expedited studies should be revisited when more robust or stronger data emerges (ie. higher in the hierarchy of evidence-based medicine) as lower quality studies can affect patient safety, resource allocation, and form the basis of future research. We agree with the reviewer that the initial expedition allowed us to greatly understand COVID-19, but came at a great cost of methodological rigor as we have highlighted with the hydroxychloroquine scandal in the following paragraph.

Furthermore, despite 7 million patients (as of August 31, 2020) infected with COVID-19, we have only had two landmark RCTs that allowed the development of therapeutic options (ie. dexamethasone and remdesivir). Who knows how many more questions could have been addressed with appropriate randomization. For example, the evidence for the use of anticoagulation to prevent thromboembolism and convalescent plasma remains poor and has been criticized by clinicians worldwide.

The information regarding the potential benefit for expedition of publication along with problems which may exist from early publication without proper appraisal has been updated in the discussion of the manuscript and is found on **pages 12-14** and below.

“In the early stages of the COVID-19 pandemic, an urgent need for data to inform clinical, social and economic decisions led to rapid dissemination and explosion in publication of COVID-19 studies.^{16, 17} The accelerated process allowed the understanding of natural history of COVID-19 and identification of tools to diagnose SARS-CoV-2. Despite the initial expedition to publication to better understand the disease, the data obtained in these studies should be revisited as stronger data emerges as lower quality studies can fundamentally risk patient safety, resource allocation, and future scientific research as these studies are based on flawed initial observations.² Ultimately, poor evidence begets poor clinical decisions.¹⁸ Furthermore, poor scientific evidence risks undermining the public’s trust in science during this time and has already been evidenced through misleading information and high-profile retractions.^{17, 19, 20, 21} Finally, early low quality studies can significantly decrease value of the scientific enterprise and increase waste of research funding replicating early poorly performed studies.²² Traditional peer-review processes have also been strained by the explosion of COVID-19 related articles with editorial boards revising their peer-review strategies to deal with the increase in submissions.^{23, 24}

Major breakthroughs in combating COVID-19 require properly designed studies. For example, the benefits of hydroxychloroquine, which were touted early in the pandemic based on limited data, have subsequently failed to be replicated in multiple observational studies and RCTs.^{25, 26, 27, 28, 29, 30, 31} One poorly designed study combined with rapid publication led to considerable investment of the scientific and medical community - akin to quinine being sold to the public as a miracle drug during the 1918 Spanish Influenza.^{32, 33} Moreover, as of June 30, 2020, ClinicalTrials.gov lists an astonishing 230 COVID-19 trials with hydroxychloroquine/plaquenil, and a recent living systematic review of observational studies and RCTs of hydroxychloroquine or chloroquine for COVID-19 demonstrated no evidence of benefit nor harm with concerns of severe methodological flaws in the included studies.³⁴

b. Abstract (131-32), “greater effort to appropriately weigh the existing evidence in the context of emerging high-quality research is needed” – I’m not sure I understand this conclusion. It appears to be a criticism of how the evidence is currently reviewed, and this was not the topic of this research.

We thank the reviewer for rightfully mentioning that the concluding sentence of the abstract is not the overall takeaway of the research and we have reworded our message. Specifically, we have reworded the conclusion to the overall takeaway of the manuscript such that published COVID-19 literature was accepted more quickly and of lower methodological quality than the matched manuscripts and we highlight that the appreciation of the limitations of existing data and appropriate appraisal when basing clinical decisions or investing research efforts are required. The abstract has been updated and is found on **pages 3-4** and below.

Abstract

Background: COVID-19 spread globally in early 2020 with major health consequences. While a need to disseminate information to the medical community, governmental agencies, and general public was paramount - concerns have been raised regarding scientific rigor in published reports.

We sought to evaluate the quality of reporting of currently available COVID-19 studies compared to historical controls.

Methods: MEDLINE, Embase, and Cochrane Central Register of Controlled Trials until May 14, 2020. All original clinical literature evaluating COVID-19 were identified and 1:1 historical control of the same study type in the same published journal was matched from the previous year. Two independent reviewers screened titles, abstracts, and full-texts and independently assessed methodological quality.

Results: 9895 titles and abstracts were screened, and 686 COVID-19 articles were included in the final analysis. Overall, COVID-19 articles were stratified by study design, a mean case series score (out of 5) (\pm SD) of 3.3 ± 1.1 , mean NOS cohort study score (out of 8) of 5.8 ± 1.5 , mean NOS case control study score (out of 8) of 5.5 ± 1.9 , and low bias present in 4 (6.4%) diagnostic studies and differences in quality scores were observed when dichotomized by time of submission to acceptance and journal impact factor. COVID-19 articles to historical control articles were matched in a 1:1 fashion (n=1078). The median time to acceptance was 13.0 (IQR, 5.0-25.0) days in COVID-19 articles vs. 110.0 (IQR, 71.0-156.0) days in control articles ($p < 0.0001$). Overall, methodological quality score was lower in COVID-19 articles in all study designs compared to control articles.

Conclusion: Published COVID-19 studies were accepted more quickly and were found to be of lower reported methodological quality than matched studies published in the same journal. An appreciation of the limitations of existing data and their appropriate appraisal when making clinical decisions or investing research efforts are warranted.”

c. The conclusions at LL188-190 relate to the other covariates and therefore seem inappropriate (as noted above). If the aim was to evaluate these other factors then careful assessment of confounders would be needed. This is in fact a perfect illustration of the table 2 fallacy (Westreich D, Greenland S. The Table 2 Fallacy: Presenting and Interpreting Confounder and Modifier Coefficients. Am J Epidemiol. 2013;177(4):292-298).

We thank the reviewer for the insightful suggestion that we inappropriately concluded the findings of our regression model in lines 188-190 of the manuscript. We enjoyed reading the citation provided by the reviewer to learn more regarding the table 2 fallacy. As the multivariable logistic regression model was removed as per the recommendation of the reviewers, we altered our discussion to deviate from the language that the reviewer had rightfully critiqued. We have updated the discussion section and is found on **page 13** and below.

“The present study demonstrates comparative differences in methodological quality scores between COVID-19 literature and historical control articles. Our research highlights major differences in study quality between COVID-19 and control articles, possibly driven in part by a combination of more thorough editorial and/or peer review process as suggested by the time to publication, and robust study design with questions which are pertinent for clinicians and patient management.^{35, 36, 37, 38, 39, 40, 41,,}

d. It would have been useful to know in what respects the COVID-19 papers are of lower quality.

We thank the reviewer for the clarification with regards to what respect COVID-19 papers were of lower quality. We have changed **Table 2** from the multivariable logistic regression model to a breakdown of categories of the methodological quality scores of different study types stratified by COVID-19 and historical control articles where we present the N (%) for each of these categories when appropriate.

Minor points

4. Language requires some attention, for example:

a. A number of prepositions are omitted e.g. 1179 “shorter time from submission to acceptance COVID-19 articles”, 1153 “High case series quality was observed 133 COVID-19 articles”.

We thank the reviewer for the recommendations and appropriate changes have been made to the manuscript and is found on **pages 11-13** and below.

“Methodological quality score differences in COVID-19 versus historical control

We matched 539 historical control articles to COVID-19 articles from the same journal with identical study designs in the previous year for a final analysis of 1078 articles (**Table 1**). Overall, 554 (51.4%), 348 (32.3%) cohort, 64 (5.9%) case-control, 106 (9.8%) diagnostic, and 6 (0.6%) RCTs were identified from the 1078 total articles. Overall, the median time of acceptance was 13.0 (IQR, 5.0-25.0) days in COVID-19 articles vs. 110.0 (IQR, 71.0-156.0) days in control articles (**Figure 4A**, $p < 0.0001$). Case series quality score was lower in COVID-19 articles compared to the historical control (3.3 ± 1.1 vs. 4.3 ± 0.8 ; $p < 0.0001$; **Table 2 and Figure 4B**). Furthermore, NOS score was lower in COVID-19 cohort studies (5.8 ± 1.6 vs. 7.1 ± 1.0 ; $p < 0.0001$; **Table 2 and Figure 4C**) and case-control studies (5.4 ± 1.9 vs. 6.6 ± 1.0 ; $p = 0.003$; **Table 2 and Figure 4D**). Finally, high diagnostic study quality was observed 12 COVID-19 articles (22.6%) vs. 24 control articles (45.3%, **Table 2 and Figure 4E**, $p = 0.02$). A similar trend was observed when dichotomized by good vs. low quality scores (**Supplemental Figure 2**).”

“Discussion

In this systematic evaluation of methodological quality, COVID-19 clinical research was primarily observational in nature with modest quality. Not only were the study designs low in the hierarchy of scientific evidence, we found that COVID-19 articles had low methodological quality scores when published with a shorter time of publication and in lower impact factor journals. In a matched cohort analysis with historical control articles identified from the same journal of the same study design, we demonstrate that COVID-19 articles were associated with lower quality scores and characterized by a shorter time from submission to acceptance. Overall, the accelerated publication of COVID-19 research negatively affected the study quality scores compared to previously published historical control studies.”

b. “comparative” used when “comparable” meant, e.g. l29

We thank the reviewer for the recommendations and appropriate changes have been made to the abstract and is found on **page 4** in the abstract and below.

“**Conclusion:** Published COVID-19 studies were accepted more quickly and were found to be of lower reported methodological quality than matched studies published in the same journal. An appreciation of the limitations of existing data and their appropriate appraisal when making clinical decisions or investing research efforts are warranted.”

5. I do not think this is a case-control design. A case-control design would involve taking some low-quality papers, matching them to high-quality papers, and assessing whether they addressed COVID-19. I think the design is a “matched cohort design”. (This comment is only about terminology and is not a criticism of the methods.)

The reviewer is correct in that this is indeed a matched cohort design and not a case-control study. We have made the appropriate change in the manuscript to reflect the change from case-

control to matched cohort design.

6. L42-44, conducting the research is also important! Ref 3 included this under point “2”).

We have changed reference #3 from point 4) to 2). Thank you for the thorough evaluation of our reference placement.

7. Some confusion between study conduct and study reporting. e.g. l59 PRISMA is not a guideline for study conduct.

We have made the change under the method section that we reported our findings according to PRISMA and is found on **page 6** and below.

“Methods

A systematic literature search was conducted on May 14, 2020 (**PROSPERO: CRD42020187318**) and reported according to the Preferred Reporting Items for Systematic Reviews and Meta-Analyses. Furthermore, the cohort study was reported according to the Strengthening The Reporting of Observational studies in Epidemiology checklist. The data supporting the findings of this study is available in the Supplemental Appendix.”

8. L67-68, I was surprised articles back to January 2019 were included. Given that COVID-19 was not defined until 2020, how many papers were retrieved from 2019? Related to this, it would be useful to see the distribution of articles by month of publication.

The reviewer made an interesting observation regarding our search strategy which stretched back to January 1, 2019. Indeed, the reviewer is correct in that COVID-19 was not defined until the end of 2019, but our strategy went until the inception of January 2019. The rationale behind this search strategy was to capture any potential SARS-CoV-2 or COVID-19 literature which may have pre-existed the global knowledge of the pandemic. The search strategy also contained multiple filters specific for COVID-19 or SARS-CoV-2 to filter out any erroneous manuscripts

which may have been captured with the increased date range. As for the reviewers questions, we captured zero articles from 2019 and when stratified by months, we had 3 articles from January, 39 articles from February, 129 articles from March, 299 articles from April, 113 articles from May, 5 articles from June, and 1 article from July. The 6 articles that were identified beyond our search date exists due to the discrepancy that exists between the publication date (ie. existing in a future volume and/or issue) compared to the acceptance date.

9. L76, “RCTs evaluating COVID-19” – I hope they were instead evaluating treatments for COVID-19.

The reviewer is correct and we have made the changes in the methods section to reflect this finding and is found on **pages 6-7** and below.

“Study selection, data extraction, and quality assessment

We included all types of COVID-19 clinical studies, including case series, observational studies, diagnostic studies, and RCTs. For diagnostic studies, the reference standard for COVID-19 diagnosis was considered a nasopharyngeal swab followed by reverse transcriptase polymerase chain reaction in order to detect SARS-CoV-2. We excluded studies that were exploratory or pre-clinical in nature (ie. *in vitro* or animal studies), case reports or case series of <5 patients,

studies published in a language other than English, reviews, methods or protocols, and other coronavirus variants such as the Middle East Respiratory Syndrome.”

10. L76-78, I cannot see the relevance of this.

The QUADAS-2 reporting tool requires a “reference standard” in order to compare new diagnostic tools.⁴ Since it is currently recognized that nasopharyngeal swabs with RT-PCR confirmation of SARS-CoV-2 is the “gold standard” to diagnose SARS-CoV-2, it was stated *a priori* when evaluating the quality of identified diagnostic studies in the study.

11. L101, give a reference for the NOS.

We have added the appropriate reference for the Newcastle-Ottawa Scale in the methods section and is found on **page 7** (methods) and below.

“First and corresponding author names, date of publication, title of manuscript, and journal of publication were collected for all included full-text articles. Journal impact factor was obtained from the 2018 InCites Journal Citation Reports from Clarivate Analytics. Submission and acceptance dates were collected in manuscripts when available. Other information such as study type, prospective or retrospective study, sex reporting, sample size calculation, method of SARS-CoV-2 diagnosis, and ethics approval was collected by the authors. Quality assessment was conducted using the Newcastle-Ottawa Scale (NOS) for case-control and cohort studies,⁷ QUADAS-2 tool for diagnostic studies,⁴ Cochrane Risk of Bias for RCTs,⁵ and a score derived by Murad et al. for case series were collected.⁶”

12. L142 and elsewhere, avoid causal language like “resulted in”.

We thank the reviewer for the suggestion. We removed causal language throughout the manuscript as this was largely an association study.

13. L174-5, the study designs were low in the hierarchy of scientific evidence: this is not surprising since this was a very new science.

The reviewer is correct in that COVID-19 (SARS-CoV-2) is a newly identified virus and that the study designs were low in the hierarchy of scientific evidence. Despite this understanding, it remains unfathomable that within 9895 COVID-19 related articles as of May 14, 2020, over 9000 were excluded due to the study being pre-clinical in nature, case report, case series <5 patients, in an language other than English, reviews (including systematic reviews), study protocols or methods, and other coronavirus variants. Furthermore, 55.4% of the studied articles were of case series – and while it allowed us to understand the natural history of the disease, it failed to provide prognostic indicators or therapies to target this novel disease. Ultimately, two drugs which appear to be effective against COVID-19, dexamethasone⁴² and remdesivir,⁴³ were of RCT design and not originating from the studies of low hierarchy of evidence. As such, we wish to keep the sentence that study designs were low of hierarchy of evidence as it lends credence to the discussion.

14. L242, median is a property of a sample, not of a single manuscript.

The reviewer is correct, we have made the necessary changes in the Figure captions to reflect the reviewers concerns and is found on **page 17** and below.

“Figure captions

Figure 1. Literature search and selection of COVID-19 articles.

Figure 2. COVID-19 clinical literature quality assessment. **(A)** Distribution of COVID-19 case series studies scored using the Murad tool. **(B)** Distribution of COVID-19 cohort studies scored using the Newcastle-Ottawa Scale. **(C)** Distribution of COVID-19 case-control studies scored using the Newcastle Ottawa Scale. **(D)** Distribution of COVID-19 diagnostic studies scored using the QUADAS-2 tool.

Figure 3. Differences in methodological quality scores in COVID-19 by secondary outcomes **(A)** When stratified by median time of acceptance (13.0 days), increased time of acceptance was associated with higher case series score ($p=0.02$). **(B)** Increased time of acceptance was associated with higher NOS cohort score ($p=0.003$). **(C)** No difference in time of acceptance and NOS case-control score was observed ($p=0.34$). **(D)** No difference in time of acceptance and

diagnostic risk of bias (QUADAS-2) was observed ($p=0.23$). **(E)** When stratified by journal impact factor (≥ 10), high journal impact factor was associated with higher case series score ($p<0.0001$). **(F)** High journal impact factor was associated with higher NOS cohort score ($p=0.01$). **(G)** No difference in journal impact factor and NOS case-control score was observed ($p=0.052$). **(H)** No difference in journal impact factor and diagnostic risk of bias (QUADAS-2) was observed ($p=0.93$). **(I)** When stratified by geographical region, no difference in geographical region and case series score was observed ($p=0.10$). **(J)** Geographical region was associated with differences in NOS cohort score ($p=0.01$). **(K)** No difference in geographical region and NOS case-control score was observed ($p=0.81$). **(L)** No difference in geographical region and diagnostic risk of bias (QUADAS-2) was observed ($p=0.34$). Differences in score distributions was analyzed by Kruskal-Wallis Test. Differences in diagnostic risk of bias was quantified by χ^2 test. $P<0.05$ was considered statistically significant.

Figure 4. Differences in methodological quality scores in COVID-19 compared to historical control articles. **(A)** Median time to acceptance was reduced in COVID-19 articles compared to control articles (13.0 [IQR, 5.0-25.0] days vs. 110.0 [IQR, 71.0-156.0] days, $p<0.0001$). **(B)** When compared to historical control articles, COVID-19 articles were associated with lower case series score ($p<0.0001$). **(C)** COVID-19 articles were associated with lower NOS cohort score compared to historical control articles ($p<0.0001$). **(D)** COVID-19 articles were associated with lower NOS case-control score compared to historical control articles ($p=0.003$). **(E)** COVID-19 articles were associated with higher diagnostic risk of bias (QUADAS-2) compared to historical control articles ($p=0.02$). Mann-Whitney U-test was conducted to evaluate differences in median time to acceptance between COVID-19 and control articles. Differences in study quality scores was evaluated by Kruskal-Wallis test. Differences in diagnostic risk of bias was quantified by χ^2 test. $P<0.05$ was considered statistically significant.”

15. F2, is the assessment for diagnostic studies lacking the words “risk of”?

The reviewer is correct and we have added risk of bias to the diagnostic study subpanel in **Figure 2D**.

16. F2 and F3 should not report the assessment for case-control studies differently (2 categories in F3, 2 in F2).

We thank the reviewer for the suggestion, we have changed the dichotomized outcomes to ordinal scales and double-checked the data for consistency.

17. F2 and F3, reporting 2 decimal places is too many.

We have made the changes to report 1 decimal place for each of these outcomes in the Figures.

18. T1, impact factor should not be reported to one decimal place since it is an integer. Also please note (in a footnote?) that articles are implicitly matched on this variable.

We have made the appropriate changes in **Table 1** and placed in the legends that the articles were matched by journal impact factor (the same journal) and study design.

Reviewer #3 (Remarks to the Author):

Thank you for the opportunity to review this paper. I believe that several things were done very well, including the search strategy and pre-registration and the reporting of the results as specified in the methods section. I also believe this to be a critically important topic.

In order to focus on the areas of greatest concern, I have focused on critique of the methods in particular. I have largely skipped the Introduction and Discussion as a result, which I found to be well written and compelling.

Overall, I did not find this to be compelling in its current form due to three main issues: 1) The methods for evaluating “quality” and “rigor” are extremely limited. I would not refer to this as quality assessments at all, and certainly not call these studies “high quality” if they had a higher score set by an arbitrary threshold. 2) The matched sample procedures are undocumented, and unlikely appropriate for comparison. At the very least, the procedures need to be documented and made replicable. 3) The statistical model and dichotomization of the data were unnecessary and inappropriate for analysis.

Given that this study is about assessing methodological quality and rigor, the methods used to evaluate quality should meet a high standard, which I do not believe was achieved here.

A heavy revision would vastly improve this paper, including revising the language throughout to avoid calling this a “quality” or “rigor” assessment (especially avoiding suggesting “high” quality”, removing the counterfactual (see below for alternative suggestions). A heavy revision of the statistical methods and dichotomization will greatly help make the main results more believable, possibly removing any linkage between score and categorical “quality” assessment.

We thank the reviewer for the thorough peer-review and taking the time to provide suggestions and recommendations which will greatly improve the quality of the manuscript. We hope you agree that the points below along with recommendations made by the other reviewers strengthened the overall methodology and results of the manuscript.

Methods:

Data sources and searches

I appreciate the full details in the search terms; this seems very replicable and followable.

We thank the reviewer for the kind words. As per reviewer #1 request, we have attached the search strategy for all MEDLINE, Embase, and Cochrane search strategies as **Supplemental Tables** and below.

Supplemental Table 1. Ovid MEDLINE(R) Search Strategy		
#	Searches	Results
1	exp coronavirus infections/	12186
2	coronavirus/ or exp betacoronavirus/	10086
3	(19nCoV or 2019-nCoV or 2019nCoV or corona virus 2 or coronavirus* 2 or corona* 2019 or COV 2 or COVID19* or COVID-19 or COVID or covid-19 or HCoV-19 or ncov* or nCov-2019 or novel corona* or SARS-COV-2 or SARSCOV-2 or SARSCOV2 or severe acute respiratory syndrome?).mp.	18928
4	((new or novel or "19" or "2019" or Wuhan or Hubei or China or Chinese) adj3 (betacoronavirus* or CoV or HCoV)).ti,ab,kf,ot.	956
5	(coronavirus* or corona virus* or covid).ti,ab,kf,ot.	23017
6	((Wuhan or Hubei) adj5 pneumonia).ti,ab,kf,ot.	133
7	or/1-6	30869
8	limit 7 to yr="2019 -Current"	13814
9	limit 8 to english language	12967
10	exp animals/	23161365
11	exp animal experimentation/ or exp animal experiment/	9378
12	exp models animal/	563307
13	nonhuman/	0
14	exp vertebrate/ or exp vertebrates/	22505010
15	or/10-14	23163305
16	exp humans/	18463173
17	exp human experimentation/ or exp human experiment/	12438
18	16 or 17	18463827
19	15 not 18	4700102
20	9 not 19	12685
21	(comment or newspaper article or editorial or letter or note).pt.	1857701
22	20 not 21	8796

Supplemental Table 2. Embase Classic+Embase Search Strategy		
#	Searches	Results
1	exp Coronavirus infection/	13012
2	coronavirinae/ or exp betacoronavirus/	10698
3	(19nCoV or 2019-nCoV or 2019nCoV or corona virus 2 or coronavirus* 2 or corona* 2019 or COV 2 or COVID19* or COVID-19 or COVID or covid-19 or HCoV-19 or ncov* or nCov-2019 or novel corona* or SARS-COV-2 or SARSCOV-2 or SARSCOV2 or severe acute respiratory syndrome?).mp.	19343
4	((new or novel or "19" or "2019" or Wuhan or Hubei or China or Chinese) adj3 (betacoronavirus* or CoV or HCoV)).ti,ab.	843
5	(coronavirus* or corona virus* or covid).ti,ab.	20863
6	((Wuhan or Hubei) adj5 pneumonia).ti,ab.	124
7	or/1-6	30832
8	limit 7 to yr="2019 -Current"	11318
9	limit 8 to english language	10708
10	exp animals/	27649640
11	exp animal experimentation/ or exp animal experiment/	2548806
12	exp models animal/	1373100
13	nonhuman/	6187094
14	exp vertebrate/ or exp vertebrates/	26840780
15	or/10-14	29497971
16	exp humans/	22226816
17	exp human experimentation/ or exp human experiment/	496553
18	16 or 17	22228574
19	15 not 18	7270503
20	9 not 19	9455
21	(comment or newspaper article or editorial or letter or note).pt.	2561942
22	20 not 21	5756
23	(conference abstract or conference review).pt.	3785494
24	22 not 23	5661

Supplemental Table 3. Ovid Cochrane Central Register of Controlled Trials Search Strategy		
#	Searches	Results
1	exp coronavirus infections/	157
2	coronavirus/ or exp betacoronavirus/	2
3	(19nCoV or 2019-nCoV or 2019nCoV or corona virus 2 or coronavirus* 2 or corona* 2019 or COV 2 or COVID19* or COVID-19 or COVID or covid-19 or HCoV-19 or ncov* or nCov-2019 or novel corona* or SARS-COV-2 or SARSCOV-2 or SARSCOV2 or severe acute respiratory syndrome?).mp.	395
4	((new or novel or "19" or "2019" or Wuhan or Hubei or China or Chinese) adj3 (betacoronavirus* or CoV or HCoV)).ti,ab.	35
5	(coronavirus* or corona virus* or covid).ti,ab.	384
6	((Wuhan or Hubei) adj5 pneumonia).ti,ab.	14
7	or/1-6	473
8	limit 7 to yr="2019 -Current"	330

Data extraction and quality assessment:

“A historical comparator group was generated by identifying reports of the same study design matched in a 1:1 fashion.” This is the only sentence describing the construction of the counterfactual comparator on which one of the key claims is made. However, this does not contain nearly enough detail. How were these identified? What search terms? How were they matched? In what period of time were they identified? What does “same study design mean?” Without this information, I cannot verify the veracity of any claim made based on the counterfactual comparison. Some additional detail is provided in the results (same journal, previous year), but this needs to be in the methods section. As presented, it looks like this matching was done by hand, in a non-systematic and non-replicable manner. If that is the case, it runs severe risk of selection-related biases, and is not appropriate for use in this analysis, and I believe it should be removed from the paper or redone using a systematic method.

The reviewer makes an excellent point regarding the clarification of the search strategy to identify historical control articles from the identified COVID-19 manuscripts. Following the extraction of COVID-19 articles (n=686), we generated the historical comparison group by searching the same journal in a systematic fashion by first identifying the study design using the search terms (“case series”, “cohort”, “case control”, or “diagnostic”) starting in the journal 12 months prior to publication of the COVID-19 article (ie. COVID-19 article published on April 2020, filtered backwards to April 2019 identifying the same study design for cohort for cohort articles). We then proceeded to screen the abstract of each article proceeding forward (or backward if the specific article type was not identified) in a temporal fashion until the first study

meeting the criteria was identified. If no comparison manuscript was found by either reviewers, the corresponding COVID-19 articles were excluded from the historical control analysis. Often times, the reasons for exclusion included the lack of the same study design (ie. case series are rarely published in high impact factor journals). Following the identification of the proper control manuscript, we extracted the necessary data and quality assessment using the same standardized case report form as per the COVID-19 article.

We have considered various methods of systematic search prior to conducting this analysis *a priori* as no method is perfect for this analysis including: 1) matching articles based on the top ten journals identified from the systematic search strategy as recommended by our expert librarian (Sarah Visintini, co-author), 2) filtering recent articles published by the corresponding author/principal investigator of the COVID-19 manuscript, and 3) identifying historical control articles as we have decided to do in the manuscript. When we opt for option #1, the top 10 journals which were captured in the search strategy as of May 14, 2020 are as follows:

1. BMJ Clinical Research Ed (175 articles) – Impact factor: 27.6
2. Journal of Medical Virology (167 articles) – Impact factor: 2.1
3. Lancet (135 articles) – Impact factor: 59.1
4. BMJ (91 articles) – Impact factor: 27.6
5. Journal of Infection (85 articles) – Impact factor: 5.1
6. Lancet Infectious Diseases (83 articles) – Impact factor: 27.5
7. Clinical Infectious Diseases (79 articles) – Impact factor: 9.1
8. JAMA (78 articles) – Impact factor: 51.8
9. Travel Medicine and Infectious Disease (68 articles) – Impact factor: 4.9
10. Infection Control and Hospital Epidemiology (61 articles) – Impact factor: 2.9

The problem with identifying control articles from the top 10 journals are flawed, as it is based on the total number of articles from the 9895 articles identified in our search strategy. Specifically, we excluded over 9000 articles which failed to meet our inclusion criteria, where we excluded studies which were exploratory or pre-clinical in nature (ie. *in vitro* or animal studies), case reports, case series <5 patients, studies published in a language other than English, reviews, methods or protocols, and other coronavirus variants such as the Middle East Respiratory Syndrome. Furthermore, although journal impact factor has not been associated with manuscript quality in the past, we have shown in our analysis that journal impact factor in times of COVID-19 greatly impact the overall quality score of the manuscript (**Figure 2C**). As we can

see from above, the average impact factor of the top 10 COVID-19 publishing journals was 23.9, which is the top 99.6% of all journals with impact factors according to Clarivate Analytics. Ultimately, we believe that this would bias our results to favour the historical control if we were to identify the study of the same design from these top 10 journals and the study authors elected against this method.

Method #2 was to evaluate all peer-reviewed manuscripts published by the corresponding author or principal investigator from the previous year to match the study design of their recently published COVID-19 article. The problem with this method again lies with inherent differences in the quality of the manuscript published in different journals. For example, if the COVID-19 article published by author John Smith was in JAMA but the previous non-COVID-19 manuscripts are published in a lower impact factor journal, the quality of the assessment will inherently be biased against the historical control.

We believe method #3, the method used for the analysis was the most systematic approach to minimize selection bias in order to account for inherent differences between journals and manuscript quality while permitting a direct comparison of articles based on the study design (ie. cohort for cohort study). As per the updated **Table 1**, the articles were matched in study design in a 1:1 fashion along with the journals to permit a direct comparison of the two cohorts. Furthermore, the data was collected in a systematic fashion using a standardized case report form for each article and adjudicated between two independent reviewers to reach consensus prior to any analysis being performed to prevent any biases. The updated methods containing the additional information requested has been added to the manuscript and is found on **pages 5-7** and below.

Methods

A systematic literature search was conducted on May 14, 2020 (**PROSPERO: CRD42020187318**) and conducted according to the Preferred Reporting Items for Systematic Reviews and Meta-Analyses guidelines.

Data sources and searches

The search was created in MEDLINE using a combination of key terms and index headings related to COVID-19 and translated to the remaining bibliographic databases (**Supplemental Table 1**). The searches were conducted in MEDLINE (Ovid MEDLINE(R) ALL 1946-), Embase (Ovid Embase Classic + Embase 1947-) and the Cochrane Central Register of Controlled Trials (from inception). Search results were limited to English-only publications, and a publication date limit of January 1, 2019 to present was applied. In addition, a Canadian Agency for Drugs and Technologies in Health search filter was applied in MEDLINE and Embase to remove animal studies, and comment, newspaper article, editorial, letter and note publication types were also eliminated. Search results were exported to Covidence and duplicates were eliminated using the platform's duplicate identification feature.

Study selection

We included all clinical studies from case series, observational studies, diagnostic studies, and RCTs evaluating COVID-19. For diagnostic studies, the reference standard was considered a nasopharyngeal swab followed by reverse transcriptase polymerase chain reaction in order to detect SARS-CoV-2. We excluded studies which were exploratory or pre-clinical in nature (ie. *in vitro* or animal studies), case reports, case series <5 patients, studies published in a language other than English, reviews, methods or protocols, and other coronavirus variants such as the Middle East Respiratory Syndrome.

Data extraction and quality assessment

Title and abstracts were evaluated by two independent reviewers using Covidence (Melbourne, Australia) and all discrepancies were resolved by consensus. Articles that were selected for full review were independently evaluated by two reviewers for quality assessment. A historical comparator group was generated by identifying reports of the same study design matched in a 1:1 fashion. These were identified by searching the same journal starting in the edition 12 months prior to publication and proceeding forward in a temporal fashion until the first matched study was identified. Quality assessment was similarly conducted on the identified articles. If no comparator manuscript was found, the corresponding COVID-19 article was excluded from the comparative analysis.

Data synthesis and analysis

Continuous variables were reported as mean \pm SD or median \pm IQR as appropriate, and categorical variables were reported as proportions (%). Normally distributed continuous variables were compared using Mann-Whitney U-test and categorical variables and quality scores were compared by Chi-squares, Fisher's exact test, or Kruskal-Wallis test.

The primary outcome of interest was to evaluate the quality of COVID-19 by study type using Newcastle-Ottawa Scale (**NOS**) for case-control and cohort studies, QUADAS-2 tool for diagnostic studies,⁴ Cochrane Risk of Bias for RCTs,⁵ and a score derived by Murad et al. for case series.⁶ Prespecified secondary outcomes were comparison of quality scores by: i) median time to acceptance, ii) impact factor, iii) geographical region, and iv) historical comparator. Good quality of NOS was defined as 3+ on selection and 1+ on comparability and 2+ on outcome/exposure domains. High quality case series was defined as a score ≥ 3.5 . Time to

acceptance was defined as the time between submission to acceptance which captures peer review and editorial decisions. Geographical region was stratified on a continent basis into Asia/Oceania, Europe/Africa, and Americas (North and South America).

The association of high methodological quality with COVID-19 and control studies, median time to publication, high journal impact factor, and geographical region was assessed by simple and multivariable logistic regression and was reported as odds ratio (**OR**) with 95% confidence intervals. All statistical analyses were performed using SAS v9.4 (SAS Institute, Inc., Cary, NC, USA). Statistical significance was defined as $P < 0.05$. All figures were generated using GraphPad Prism v8 (GraphPad Software, La Jolla, CA, USA).”

Data synthesis and analysis

One of my biggest concerns here is that the assessments for quality are extremely blunt instruments, particularly the NOS scale. The NOS scale is often used for purposes such as this, but is an incredibly inadequate measure of quality, and may even be counter-indicative of methodological and analytical rigor in many cases. The majority of scoring is on extremely superficial measures, without dealing with the fundamentally most important issues (often issues with causal identification).

The reviewer highlights an important limitation which are utilized to assess the quality of identified manuscripts in systematic reviews and meta-analysis. Indeed, although Newcastle-Ottawa Scale (**NOS**),⁷ QUADAS-2,⁴ Murad tool,⁶ and the Cochrane Risk of Bias⁵ are commonly utilized in systematic reviews, we agree with the reviewer that we are limited overall to superficial measures. While these measures are superficial in nature, failure to meet fundamental reporting requirements including patient selection, comparability, reporting of reference standard (for diagnostic studies), and outcome or exposure, makes it impossible to accurately assess associations or causal identification of the disease or outcome of interest from the manuscript of interest. Furthermore, quality checklists or scales such as the NOS are ubiquitously used in systematic reviews and meta-analyses for multiple reasons including excluding “low quality” manuscripts from the meta-analysis, performing meta-regression or sensitivity analysis, or weighing studies for the meta-analysis.⁴⁴

The NOS is principally used to evaluate the study quality of non-randomized studies in meta-analyses of both case-control and cohort designs. The strength of NOS lies upon the ease-of-use and applicability of the scale/checklist across multiple medical fields.¹⁵ One of the major critique of NOS lies within the comparability criteria, where one star/point is given for the “most important factor” and another for additional factors that are modified for the specific question of interest for a total of two stars.¹⁵ In order to circumvent this limitation, our NOS analysis of comparability received a point when the most important factor in both study designs being adjusted for known covariates (ie. age, gender, co-morbidities) by design or analysis (ie. adjusted regression analysis) which influence the findings of the study were considered by the authors. A major source of bias in meta-analyses is due to the limitations of the information contained within the original articles.¹⁴

Ultimately, the issues with causal inference identified from selected manuscripts cannot be ascertained by NOS nor through any of the currently available quality scales or checklists. Furthermore, quality checklists and scores can only adequately assess what has been provided within the methodology, results, and discussion of the manuscript. In the context of the limitations of NOS, Sanderson et al. has reviewed 86 quality checklists and scales (including NOS) assessing non-randomized studies and demonstrated the lack of a single gold standard tool to assess the quality of observational studies.¹⁴ Furthermore, other quality measurements such as sample size calculation or even ethics approval are not captured in these checklists. We have attempted to capture different aspects of quality and reported these findings in **Table 1**. Indeed, a high-quality report captured by the NOS may not accurately portray low susceptibility to bias and vice versa.⁴⁵ Other factors such as peer-review process and journal restrictions restrict reporting of methodology which may affect the quality of the published methods also cannot be accurately captured using checklists.

Despite the inherent limitations, no tool currently accurately captures both study quality and causal inference made from the original articles and we agree that such a tool would be valuable for future assessment of articles not only in this study but for all systematic reviews and meta-analysis. Indeed, future checklists should include scores to grade causal language used in

observational studies and appropriately penalize the field. However, this would come at a great cost of the general applicability of scores such as the NOS as content expertise is required in order to properly grade causality in the field. Most importantly, our assessment highlights the methodological shortcomings of the identified COVID-19 manuscripts due to a need for rapid dissemination of information regarding the SARS-CoV-2 virus as demonstrated by a significant decrease in time to acceptance (13.0 days vs. 110.0 days; $p < 0.0001$) with without any inference to causality or translational applicability of these identified studies.

We have stated the potential limitation of using methodological quality checklists/scales under the limitation section of our manuscript and is found on **pages 14-15** and below.

“Our study has important limitations. We evaluated the methodological quality of existing studies using established quality scores. While it is tempting to associate quality scores with reproducibility or causal inferences, it is not possible to ascertain the impact on the study design and conduct of research nor results or conclusions in the identified reports.¹³ Second, although the methodological quality scales and checklists used for the manuscript are commonly used for quality assessment in systematic reviews and meta-analyses,^{4, 5, 6, 7} they can only assess the methodology without consideration for causal language and prone to limitations.^{14, 15} Furthermore, other considerations of quality such as sample size calculation, sex reporting, or ethics approval is not considered in these quality scores. As such, the quality scores measured using these checklists only reflect the patient selection, comparability, diagnostic reference standard, and methods to ascertain the outcome of the study. Third, the 1:1 ratio to identify our historical control articles may affect the precision estimates of our findings. Interestingly, a simulation of an increase from 1:1 to 1:4 control ratio tightened the precision estimates but did not significantly alter the point estimate.¹¹ Furthermore, the decision for 1:1 ratio exists due to limitations of available historical control articles from identical journal in the restricted time

period combined with large effect size and sample size in the analysis. Finally, our analysis includes early publications on COVID-19 and there is likely to be an improvement in quality of related studies and study design as the field matures and higher quality studies, which take longer to design, conduct, and report are published. Accordingly, our findings are limited to the early body of research as it pertains to the pandemic and it is likely that over time research quality will improve.”

In the case of the Risk of Bias measure, Cochrane explicitly recommends against using RoB for summary scores, for these very reasons. I am unfamiliar with the QUADAS-2 and the Murad tools, so I can't comment on those.

The reviewer is correct in that Cochrane recommends against the use of the Risk of Bias assessment for summary scores. Indeed, for the manuscript, we identified six randomized controlled trials (**RCT**) in our full-text extraction and is important to note that none of the data analysis in **Figure 2-3** and the manuscript incorporated the six RCTs in the final analysis. Furthermore, comparator articles were identified in only 4 of the studies and no comparison was done to the historical control articles. We have opted to only describe the findings of the RoB of the RCTs in the results and clarified the misunderstanding in both the text and figure captions and is found on **pages 8-9 and 10-11** and below.

“Data synthesis and statistical analysis

Continuous variables were reported as mean \pm SD or median \pm IQR as appropriate, and categorical variables were reported as proportions (%). Normally distributed continuous variables were compared using Mann-Whitney U-test and categorical variables and quality scores were compared by χ^2 , Fisher's exact test, or Kruskal-Wallis test.

The primary outcome of interest was to evaluate the quality of COVID-19 by study type using Newcastle-Ottawa Scale (**NOS**) for case-control and cohort studies, QUADAS-2 tool for

diagnostic studies,⁴ Cochrane Risk of Bias for RCTs,⁵ and a score derived by Murad et al. for case series.⁶ Pre-specified secondary outcomes were comparison of quality scores by: i) median time to acceptance, ii) impact factor, iii) geographical region, and iv) historical comparator. Time of acceptance was defined as the time between submission to acceptance which captures peer review and editorial decisions. Geographical region was stratified into continents including Asia/Oceania, Europe/Africa, and Americas (North and South America). *Post-hoc* comparison analysis between COVID-19 and historical control article quality scores were evaluated using Kruskal-Wallis test. Furthermore, good quality of NOS was defined as 3+ on selection and 1+ on comparability and 2+ on outcome/exposure domains and high quality case series scores was defined as a score ≥ 3.5 . Due to a small sample size of identified RCTs, they were not included in the comparison analysis.”

“COVID-19 literature quality

Most studies originated from Asia/Oceania with 469 (68.4%) followed by Europe with 139 (20.3%), and the Americas with 78 (11.4%). Of included studies, 380 (55.4%) were case series, 199 (29.0%) were cohort, 63 (9.2%) were diagnostic, 38 (5.5%) were case-control, and 6 (0.9%) were RCTs. Most studies (590, 86.0%) were retrospective in nature, 620 (90.4%) reported the sex of patients, and 7 (1.0%) studies calculated their sample size *a priori*. The method of SARS-CoV-2 diagnosis was reported in 558 studies (81.3%) and ethics approval was obtained in 556 studies (81.0%). Finally, the median journal impact factor of COVID-19 manuscripts was 4.7 (IQR, 2.9-7.6) with a median time to acceptance of 13.0 (IQR, 5.0-25.0) days (**Table 1**).

Overall, when COVID-19 articles were stratified by study design, a mean case series score (out of 5) (\pm SD) of 3.3 ± 1.1 , mean NOS cohort study score (out of 8) of 5.8 ± 1.5 , mean NOS case

control study score (out of 8) of 5.5 ± 1.9 , and low bias present in 4 (6.4%) diagnostic studies was observed (Table 2 and Figure 2). Furthermore, of the identified 6 RCTs in COVID-19 literature, they remain at high risk of bias with little consideration for sequence generation, allocation concealment, blinding, incomplete outcome data, and selective outcome reporting (Table 2).”

Secondly, I am concerned that these scores, which are questionable to begin with, are further degraded by dichotomizing the scores into “high” and “low.” There is no justification, as far as I can tell, to dichotomize the data, and doing so can lead to substantially misleading inference. Either use much finer ordinal bins, or keep the scoring in its original form.

We thank the reviewer with highlighting the improper categorization of the quality scores for the purpose of this analysis. We were previously unaware that dichotomization of the aforementioned scores was improper and the rationale for dichotomizing all the used tools was to allow a comprehensive analysis of all the scores into one analysis (NOS, QUADAS-2, and the Murad tool). As such, we moved these dichotomized scores into supplemental figures and re-ran the analysis for each individual score and stated the limitation within the limitation section of the manuscript. Indeed, when the analysis was changed to ordinal scale (as to the original scores), we saw similar results to the dichotomization into high vs. low quality. Specifically, when stratified by study design, a mean (\pm SD) case series score of 3.3 ± 1.1 , 5.8 ± 1.5 for cohort studies, 5.5 ± 1.9 for case control studies, and low bias was present in 4 (6.4%) diagnostic studies. Furthermore, in the 6 COVID-19 related RCTS identified in the search, they remained at high risk of bias with minimal consideration for sequence generation, allocation concealment, blinding, incomplete outcome data, and selective outcome reporting (Table 2). Furthermore, when stratified by pre-specified secondary outcomes of median time of acceptance (<13.0 days), journal impact factor, and geographical regions, we observed that rapid time from submission to acceptance resulted in lower study quality scores for case series and cohort study designs but not case-controls or diagnostic studies (Figure 3A-D). Furthermore, low journal impact factor lowered study quality score in case series, cohort, and case-control designs (Figure 3E-H). COVID-19 originating in different geographical regions had no differences in study quality

scores with the exception of cohort studies (**Figure 3I-L**). We transferred the dichotomized data to **Supplemental Figure 1** which revealed similar results.

When methodological quality scores were compared between COVID-19 and historical control articles (n=1078), we observed lower case series, cohort quality score, case-control score, and diagnostic study quality in COVID-19 articles compared to the historical control (**Table 2 and Figure 4**). The results section has been changed to reflect the new findings and is found on **pages 10-12** and below.

Results

Article selection

A total of 14787 COVID-19 papers were identified as of May 14, 2020 and 4892 duplicates were removed. 9895 titles and abstracts were screened, and 9101 articles were excluded due to the study being pre-clinical in nature, case report, case series <5 patients, in an language other than English, reviews (including systematic reviews), study protocols or methods, and other coronavirus variants with an overall inter-rater study inclusion agreement of 96.7%, similar to other evaluations performed by our group.^{1, 2, 8, 9} This left a final number of 794 full texts which were reviewed for eligibility. Over 108 articles were excluded for improper study design (such as letter to the editors, editorials, case reports, or case series <5 patients), patient population, non-English language, duplicates, wrong outcomes, and publication in a non-peer reviewed journal. Ultimately, 686 articles were identified and underwent quality assessment with an inter-rater agreement of 86.5% ($\kappa=0.68$; 95% CI, 0.67-0.70) (**Figure 1**).

COVID-19 literature quality

Most studies originated from Asia/Oceania with 469 (68.4%) followed by Europe with 139 (20.3%), and the Americas with 78 (11.4%). Of included studies, 380 (55.4%) were case series,

199 (29.0%) were cohort, 63 (9.2%) were diagnostic, 38 (5.5%) were case-control, and 6 (0.9%) were RCTs. Most studies (590, 86.0%) were retrospective in nature, 620 (90.4%) reported the sex of patients, and 7 (1.0%) studies calculated their sample size *a priori*. The method of SARS-CoV-2 diagnosis was reported in 558 studies (81.3%) and ethics approval was obtained in 556 studies (81.0%). Finally, the median journal impact factor of COVID-19 manuscripts was 4.7 (IQR, 2.9-7.6) with a median time to acceptance of 13.0 (IQR, 5.0-25.0) days (**Table 1**).

Overall, when COVID-19 articles were stratified by study design, a mean case series score (out of 5) (\pm SD) of 3.3 ± 1.1 , mean NOS cohort study score (out of 8) of 5.8 ± 1.5 , mean NOS case control study score (out of 8) of 5.5 ± 1.9 , and low bias present in 4 (6.4%) diagnostic studies was observed (**Table 2 and Figure 2**). Furthermore, of the identified 6 RCTs in COVID-19 literature, they remain at high risk of bias with little consideration for sequence generation, allocation concealment, blinding, incomplete outcome data, and selective outcome reporting (**Table 2**).

For secondary outcomes, rapid time from submission to acceptance (defined as median time of acceptance of <13.0 days) was associated with lower study quality scores for case series and cohort study designs but not case-control or diagnostic studies (**Figure 3A-D**). Low journal impact factor (<10) was associated with lower study quality scores for case series, cohort, and case-control designs (**Figure 3E-H**). Finally, studies originating from different geographical regions had no differences in study quality scores with the exception of cohort studies (**Figure 3I-L**). When dichotomized by good vs. low study quality scores, a similar trend was observed with rapid time from submission to acceptance (34.4% vs. 46.3%, $p=0.01$, **Supplemental**

Figure 1B), low impact factor journals (<10) was associated with lower study quality score (38.8% vs 68.0%, $p<0.0001$, **Supplemental Figure 1C**). Finally, studies originating in either Americas or Asia/Oceania was associated with higher quality scores than Europe (**Supplemental Figure 1D**).

Methodological quality score differences in COVID-19 versus historical control

We matched 539 historical control articles to COVID-19 articles from the same journal with identical study designs in the previous year for a final analysis of 1078 articles (**Table 1**). Overall, 554 (51.4%), 348 (32.3%) cohort, 64 (5.9%) case-control, 106 (9.8%) diagnostic, and 6 (0.6%) RCTs were identified from the 1078 total articles. Overall, the median time of acceptance was 13.0 (IQR, 5.0-25.0) days in COVID-19 articles vs. 110.0 (IQR, 71.0-156.0) days in control articles (**Figure 4A**, $p<0.0001$). Case series quality score was lower in COVID-19 articles compared to the historical control (3.3 ± 1.1 vs. 4.3 ± 0.8 ; $p<0.0001$; **Table 2 and Figure 4B**). Furthermore, NOS score was lower in COVID-19 cohort studies (5.8 ± 1.6 vs. 7.1 ± 1.0 ; $p<0.0001$; **Table 2 and Figure 4C**) and case-control studies (5.4 ± 1.9 vs. 6.6 ± 1.0 ; $p=0.003$; **Table 2 and Figure 4D**). Finally, high diagnostic study quality was observed 12 COVID-19 articles (22.6%) vs. 24 control articles (45.3%, **Table 2 and Figure 4E**, $p=0.02$). A similar trend was observed when dichotomized by good vs. low quality scores (**Supplemental Figure 2**).”

Rhetorically, labelling the quality of studies as “high quality” is also not justified; at best they are just not found to be of low quality. They may have a high score, but that scoring is not a comprehensive assessment of quality. I strongly suggest removing the word “quality” whenever in reference to the score, and perhaps from the title.

The reviewers concerns regarding the use of the words “high quality” from the dichotomized results within the manuscript. As highlighted by the answer above, we were previously unaware that dichotomization of the aforementioned scores was improper and the rationale for dichotomizing all the used tools was to allow a comprehensive analysis of all the scores into one

analysis (NOS, QUADAS-2, and the Murad tool). As such, we moved these dichotomized scores into supplemental figures and re-ran the analysis for each individual score and stated the limitation within the limitation section of the manuscript. Furthermore, we agree that the methodological quality checklists/scales measure certain aspect of quality but is not a comprehensive assessment (as stated in the limitations and previous answer pertaining to the checklists). We have removed the use of “high quality” in the figures along with the corresponding text within the manuscript and the dichotomized data remains in the supplemental figures with “high and low quality scores”. Furthermore, we respectfully disagree with the removal of the word quality when referenced to the score along with the title as the original manuscripts describing these checklists explicitly use the word “quality” or “methodological quality” when describing these tools.^{4,5,6,7} Indeed, despite the limitations which surrounds these tools, they are used ubiquitously for quality assessment in meta-analyses and remain in the forefront to assess methodological quality until an improved quality scale and/or checklist is developed. We wish to retain the use of quality throughout the manuscript along with the title as the primary objective of the study is to evaluate the quality of COVID-19 literature.

The main justification for the dichotomization seems to be to put the data in a form for plugging into logistic regression for odds ratios. However, even were the scores already binary high/low form, ORs are difficult to interpret, and models for RR should be used instead. If not dichotomizing, many other statistical model options are available that would be vastly preferable, including any number of linear models, different in means/median models, ordinal regression, etc.

We thank the reviewer for the insightful comment regarding the dichotomization of the data for logistic regression conducted to further elucidate the association of other variables with methodological quality. Indeed, the multivariable logistic regression was not the primary aim of the manuscript and has been removed as it created misunderstanding amongst the reviewers. Instead, we have separated the pre-specified secondary outcomes into **Figure 3** (previously **Figure 2B-D**) and created four figures to represent our findings along with two tables revealing the overall study characteristics and quality scores.

Results: Within the scope of the methods as discussed, I believe this to have been relatively well-reported.

We thank the reviewer for the comments and we feel the manuscript has been much improved by their input.

One question regards why “adjustments” were made for some analysis. What purpose did this serve? This was not described or justified in the methods, particularly the variables adjusted for. How do I usefully interpret this adjusted association? It seems very likely that this is hinting at some kind of causal model but pursuing the “just say association” strategy to sidestep having to justify it. I would strongly recommend removing the adjusted models altogether.

Similar to the question above, we initially designed a multivariable logistic regression in order to adjust for known confounding variables which we collected in our analysis. Unfortunately, the use of this regression model created misunderstanding amongst the reviewers and does not appear to advance the findings of the manuscript. We agree with the reviewer’s recommendation and have decided to remove the regression model along with the associated table from the manuscript and separated the pre-specified secondary outcomes into Figure 3 to a total of four figures to present our findings.

Discussion: I would suggest removing the “this is the first time” sentence. Our contribution should be about rigor, and not about being “first” to anything.

We have removed the words “this is the first time” within the discussion section of the manuscript and agree with the reviewer that the contribution of this manuscript to the field is to evaluate rigor/quality and the corresponding changes is found on **page 13** and below.

“The present study demonstrates comparative differences in methodological quality scores between COVID-19 literature compared to historical control articles. Our research highlights major differences in study quality between COVID-19 and control articles, possibly driven in part by a combination of more thorough editorial and/or peer review process as suggested by the time to publication, and robust study design with questions which are pertinent for clinicians and patient management.^{35, 36, 37, 38, 39, 40, 41}”

The limitations section needs to be vastly increased. To begin with, it needs a thorough discussion of the limitations of the review tools used related to your assessment of “quality.”

We thank the reviewer for the suggestion to increase the limitations in our manuscript. Of course, our manuscript contains limitations which has been addressed throughout the review and added to the manuscript. We agree with the reviewer that the methodological quality scales and checklists used for the quality assessment are not perfect and prone to their own limitations. As previously described, the issues pertaining to causal inference identified from selected manuscripts cannot be ascertained by NOS nor through any of the currently available quality scales or checklists. Furthermore, quality checklists and scores can only adequately assess what has been provided within the methodology, results, and discussion of the manuscript. In the context of the limitations of NOS and use of quality scores in observational studies, Sanderson et al. has reviewed 86 quality checklists and scales (including NOS) assessing non-randomized studies and demonstrated the lack of a single gold standard tool to assess the quality of observational studies.¹⁴ Furthermore, other quality measurements such as sample size calculation or even ethics approval are not captured in these checklists. We have updated our limitation to highlight the shortcomings of these checklists along with another recommendation from reviewer #1 regarding the use of 1:1 match ratio in the study and is found on **pages 14-15** and below.

“Our study has important limitations. We evaluated the methodological quality of existing studies using established quality scores. While it is tempting to associate quality scores with reproducibility or causal inferences, it is not possible to ascertain the impact on the study design and conduct of research nor results or conclusions in the identified reports.¹³ Second, although the methodological quality scales and checklists used for the manuscript are commonly used for quality assessment in systematic reviews and meta-analyses,^{4, 5, 6, 7} they can only assess the methodology without consideration for causal language and prone to limitations.^{14, 15} Furthermore, other considerations of quality such as sample size calculation, sex reporting, or ethics approval is not considered in these quality scores. As such, the quality scores measured

using these checklists only reflect the patient selection, comparability, diagnostic reference standard, and methods to ascertain the outcome of the study. Third, the 1:1 ratio to identify our historical control articles may affect the precision estimates of our findings. Interestingly, a simulation of an increase from 1:1 to 1:4 control ratio tightened the precision estimates but did not significantly alter the point estimate.¹¹ Furthermore, the decision for 1:1 ratio exists due to limitations of available historical control articles from identical journal in the restricted time period combined with large effect size and sample size in the analysis. Finally, our analysis includes early publications on COVID-19 and there is likely to be an improvement in quality of related studies and study design as the field matures and higher quality studies, which take longer to design, conduct, and report are published. Accordingly, our findings are limited to the early body of research as it pertains to the pandemic and it is likely that over time research quality will improve.”

References

1. Ramirez FD, *et al.* Journal Initiatives to Enhance Preclinical Research: Analyses of Stroke, Nature Medicine, Science Translational Medicine. *Stroke* **51**, 291-299 (2020).
2. Ramirez FD, *et al.* Methodological Rigor in Preclinical Cardiovascular Studies: Targets to Enhance Reproducibility and Promote Research Translation. *Circ Res* **120**, 1916-1926 (2017).
3. Wells G, Shea B, O'Connell D, Peterson J, Welch V, Loso M, Tugwell P. The Newcastle-Ottawa Scale (NOS) for assessing the quality of nonrandomised studies in meta-analysis. http://www.ohrica/programs/clinical_epidemiology/oxfordasp, (2004).
4. Whiting PF, *et al.* QUADAS-2: a revised tool for the quality assessment of diagnostic accuracy studies. *Ann Intern Med* **155**, 529-536 (2011).
5. Higgins JPT, *et al.* The Cochrane Collaboration's tool for assessing risk of bias in randomised trials. *BMJ* **343**, d5928 (2011).
6. Murad MH, Sultan S, Haffar S, Bazerbachi F. Methodological quality and synthesis of case series and case reports. *BMJ Evid Based Med* **23**, 60-63 (2018).
7. Wells G SB, O'Connell D, Peterson J, Welch V, Loso M, Tugwell P. The Newcastle-Ottawa Scale (NOS) for assessing the quality of nonrandomised studies in meta-analysis. http://www.ohrica/programs/clinical_epidemiology/oxfordasp, (2004).
8. Labinaz A, *et al.* Female Authorship in Preclinical Cardiovascular Research: Temporal Trends and Influence on Experimental Design. *JACC Basic Transl Sci* **4**, 471-477 (2019).
9. Ramirez FD, *et al.* Sex Bias Is Increasingly Prevalent in Preclinical Cardiovascular Research: Implications for Translational Medicine and Health Equity for Women. *Circulation* **135**, 625-626 (2017).
10. Linden A, Samuels SJ. Using balance statistics to determine the optimal number of controls in matching studies. *Journal of Evaluation in Clinical Practice* **19**, 968-975 (2013).
11. Hamajima N, Hirose K, Inoue M, Takezaki T, Kuroishi T, Tajima K. Case-control studies: matched controls or all available controls? *J Clin Epidemiol* **47**, 971-975 (1994).
12. Austin PC. A critical appraisal of propensity-score matching in the medical literature between 1996 and 2003. *Statistics in Medicine* **27**, 2037-2049 (2008).
13. Glasziou P, Chalmers I. Research waste is still a scandal—an essay by Paul Glasziou and Iain Chalmers. *BMJ*, k4645 (2018).

14. Sanderson S, Tatt ID, Higgins JP. Tools for assessing quality and susceptibility to bias in observational studies in epidemiology: a systematic review and annotated bibliography. *Int J Epidemiol* **36**, 666-676 (2007).
15. Stang A. Critical evaluation of the Newcastle-Ottawa scale for the assessment of the quality of nonrandomized studies in meta-analyses. *Eur J Epidemiol* **25**, 603-605 (2010).
16. Chen Q, Allot A, Lu Z. Keep up with the latest coronavirus research. *Nature* **579**, 193 (2020).
17. Bauchner H. The Rush to Publication: An Editorial and Scientific Mistake. *JAMA* **318**, 1109-1110 (2017).
18. Heneghan C, Mahtani KR, Goldacre B, Godlee F, Macdonald H, Jarvies D. Evidence based medicine manifesto for better healthcare. *BMJ*, j2973 (2017).
19. Mehra MR, Desai SS, Ruschitzka F, Patel AN. RETRACTED: Hydroxychloroquine or chloroquine with or without a macrolide for treatment of COVID-19: a multinational registry analysis. *The Lancet*, (2020).
20. Servick K, Enserink M. The pandemic's first major research scandal erupts. *Science* **368**, 1041-1042 (2020).
21. Mehra MR, Desai SS, Kuy S, Henry TD, Patel AN. Retraction: Cardiovascular Disease, Drug Therapy, and Mortality in Covid-19. *N Engl J Med*. DOI: 10.1056/NEJMoa2007621. *New England Journal of Medicine* **382**, 2582-2582 (2020).
22. Macleod MR, *et al*. Biomedical research: increasing value, reducing waste. *The Lancet* **383**, 101-104 (2014).
23. Bauchner H, Fontanarosa PB, Golub RM. Editorial Evaluation and Peer Review During a Pandemic: How Journals Maintain Standards. *JAMA*, (2020).
24. The Lancet Global H. Publishing in the time of COVID-19. *The Lancet Global Health* **8**, e860 (2020).
25. Boulware DR, *et al*. A Randomized Trial of Hydroxychloroquine as Postexposure Prophylaxis for Covid-19. *New England Journal of Medicine*, (2020).
26. Gautret P, *et al*. Clinical and microbiological effect of a combination of hydroxychloroquine and azithromycin in 80 COVID-19 patients with at least a six-day follow up: A pilot observational study. *Travel medicine and infectious disease* **34**, 101663-101663 (2020).
27. Geleris J, *et al*. Observational Study of Hydroxychloroquine in Hospitalized Patients with Covid-19. *New England Journal of Medicine* **382**, 2411-2418 (2020).

28. Alexander PE, *et al.* COVID-19 coronavirus research has overall low methodological quality thus far: case in point for chloroquine/hydroxychloroquine. *Journal of Clinical Epidemiology* **123**, 120-126 (2020).
29. Borba MGS, *et al.* Effect of High vs Low Doses of Chloroquine Diphosphate as Adjunctive Therapy for Patients Hospitalized With Severe Acute Respiratory Syndrome Coronavirus 2 (SARS-CoV-2) Infection: A Randomized Clinical Trial. *JAMA Network Open* **3**, e208857-e208857 (2020).
30. Mercurio NJ, *et al.* Risk of QT Interval Prolongation Associated With Use of Hydroxychloroquine With or Without Concomitant Azithromycin Among Hospitalized Patients Testing Positive for Coronavirus Disease 2019 (COVID-19). *JAMA Cardiology*, (2020).
31. Molina JM, *et al.* No evidence of rapid antiviral clearance or clinical benefit with the combination of hydroxychloroquine and azithromycin in patients with severe COVID-19 infection. *Médecine et Maladies Infectieuses* **50**, 384 (2020).
32. Shors T, McFadden SH. 1918 influenza: a Winnebago County, Wisconsin perspective. *Clin Med Res* **7**, 147-156 (2009).
33. Stolberg S. A Mad Scramble to Stock Millions of Malaria Pills, Likely for Nothing. In: *The New York Times* (2020).
34. Hernandez AV, Roman YM, Pasupuleti V, Barboza JJ, White CM. Hydroxychloroquine or Chloroquine for Treatment or Prophylaxis of COVID-19: A Living Systematic Review. *Annals of Internal Medicine*, (2020).
35. Chalmers I, Glasziou P. Avoidable waste in the production and reporting of research evidence. *Lancet* **374**, 86-89 (2009).
36. Barakat AF, Shokr M, Ibrahim J, Mandrolia J, Elgendy IY. Timeline from receipt to online publication of COVID-19 original research articles. *medRxiv*, 2020.2006.2022.20137653 (2020).
37. Chan A-W, *et al.* Increasing value and reducing waste: addressing inaccessible research. *The Lancet* **383**, 257-266 (2014).
38. Ioannidis JPA, *et al.* Increasing value and reducing waste in research design, conduct, and analysis. *The Lancet* **383**, 166-175 (2014).
39. Chalmers I, *et al.* How to increase value and reduce waste when research priorities are set. *The Lancet* **383**, 156-165 (2014).

40. Salman RA-S, *et al.* Increasing value and reducing waste in biomedical research regulation and management. *The Lancet* **383**, 176-185 (2014).
41. Glasziou P, *et al.* Reducing waste from incomplete or unusable reports of biomedical research. *The Lancet* **383**, 267-276 (2014).
42. Group RC, *et al.* Dexamethasone in Hospitalized Patients with Covid-19 - Preliminary Report. *N Engl J Med*, (2020).
43. Beigel JH, *et al.* Remdesivir for the Treatment of Covid-19 - Preliminary Report. *N Engl J Med*, (2020).
44. Deeks JJ, *et al.* Evaluating non-randomised intervention studies. *Health Technol Assess* **7**, iii-x, 1-173 (2003).
45. Huwiler-Muntener K, Juni P, Junker C, Egger M. Quality of reporting of randomized trials as a measure of methodologic quality. *JAMA* **287**, 2801-2804 (2002).

Reviewer comments, second round:

Reviewer #1 (Remarks to the Author):

I thank the authors for the comprehensive replies to my comments, questions and suggestions.

1. Original point #5. I'm not sure where this information provided has been inserted into the methods? I found the response to be incredibly difficult to navigate as the authors pasted whole sections of the article, rather than simply highlighting the specific text that had been revised. In the future, I would suggest simply highlighting the revised text.

Again, I recognize that there are limitation in word count for the manuscript, however this information could be included in supplemental methods online. The level of training suggested by the authors speaks to rigor of the study and should be transparently reported.

2. Original point #8. Timing of the registration must be transparently reported, especially since the authors suggest it was incorrectly registered (i.e. the oversight they report in the response).

3. Original point #17: Thank-you for clarifying. As per original point #5 these details should be reported somewhere. I'm not sure why the authors are self-citing their previous studies in relation to a kappa score? There is no need for this as this has nothing to do with previous studies by the authors.

4. Original point #19: Reasons for omission of RCT analysis should be reported in the paper so the reader understands what happened. I'm not clear how reasons #1 ("summary score") and #2 ("collective score") are different? Also, it's unclear why the authors then chose this tool to begin with if the initial plan was to perform a statistical analysis.

5. Page 13: I would disagree with the sentence, "Overall, the accelerated publication of COVID-19 research negatively affected the study quality scores compared to previously published historical control studies." This sentence implies causation. Association is the strongest conclusion that can be made with the observational design of this study.

MINOR:

1. Page 10: "improper study design" – I'm not sure 'improper' is the correct term (i.e. case series might have been the correct/proper design for some of the studies you screened), and editorials are not studies. As such, I suggest "...ineligible study design or publication type..."

2. Page 10: "1.0% calculated sample size" – Not all these study types would be expected to have calculated sample sizes. For instance, case series would not. The denominator for this particular statistic should only be the study designs that would be expected to be able to perform a power calculation.

3. The lack of ethics approval (only 81% had approval, or at least reported it) seems low. Would all these study types require ethics approval?

4. Page 11: sentence does not seem to be grammatically correct? "Furthermore, of the identified 6 RCTs ...they remain at high risk of bias..."

5. Page 15: "they can only assess the methodology without consideration for causal language and prone to limitations." I wasn't clear what 'causal language' meant in this context. Also, I believe there should be an "are" inserted before "prone".

Reviewer #2 (Remarks to the Author):

I thank the authors for their substantial revision and rebuttal.

1. The responses to my comments on methods are ok, although I thought it a bit weak that kappa for inter-rater agreement could not be calculated because of failures of the Covidence software.

2. The responses to my comments on reporting are broadly ok. The text in Results still restates what can be read in the tables and figures, but there is now a good match between text and tables/figures.

3. The responses to my comments on interpretation are not adequate.

a. For my point 3a, I thought I was quite clear: a lower methodological quality and faster time to publication is not necessarily a bad thing. The authors respond that I am wrong because their data show that lower methodological quality is associated with faster time to publication. This misses the point, for two reasons. First, it is a correlation and does not imply causation. Second, even if it did imply causation, so that we could improve methodological quality by longer times to publication, it does not follow that this would have improved the world's response to covid-19. To put it very crudely, is it better to have poor quality evidence or no evidence at all? So the authors' conclusion that "the data obtained in these studies should be revisited" is entirely valid, but their implication that the scientific community was wrong to accept lower-quality rapidly-published research is not valid.

b. My point 3b criticised the writing of the abstract, and I find the revised abstract even harder to read.

i. The 2nd sentence of the results part in particular makes little sense.

ii. The results part twice repeats the methods, and the conclusion partly repeats the results.

iii. The aim stated in the background is just the 3rd of the 3 aims stated in the main paper Introduction, yet the results part gives almost no detail of this 3rd aim and lots of detail about the 1st and 2nd aims, so that it reads as not answering the stated aim.

iv. There remain a number of grammatical errors.

4. The question of the overall aims is difficult. When I first read the paper, I thought it was mainly about comparing covid-19 research with other research. But I now see this is just the 3rd of three aims. So my previous comments 2b and 3c were wrong (sorry). However I don't find aims 1 and 2 particularly interesting, and I would have placed aim 3 as primary. So the paper remains largely focussed on aims that are not very interesting. Further, the (commendable) shift of emphasis from dichotomising quality to quantifying quality makes the reporting of aim 1 even less interesting – most readers (like me) will be unable to interpret e.g. "a mean case series score of 3.3" as good or bad, and will only be able to understand it from the comparison with the matched control papers.

5. Table 2 has big problems.

a. The mean and SD of the quality scores are reported in columns wrongly headed "N" and "%". One approach here is to drop the column heading and label the rows "N (%)" or "mean (SD)".

b. For the yes/no characteristics, it is not enough to report N when the denominator is not given. E.g. the first N is 172 but the denominator is not the 686 stated at the top. (It's actually 380, found in table 1, but table 2 should be self-sufficient.)

c. the study types should be listed as well as their associated quality scores

d. "+/-" should be avoided, here and elsewhere. "Mean (SD)" is better.

6. Smaller points

- a. My previous point 14 – median is still wrongly used at line 212 (just delete “median” here).
- b. I was amused by “The primary outcome of interest was to evaluate the quality of COVID-19 ...” (L163) – but this is also another example of poor writing.

Reviewer #3 (Remarks to the Author):

This overall has been a very thorough effort by the original authors to make very substantial changes to the manuscript, and respond very clearly to the reviewers. In fact, with one exception, I believe that all of my and the rest of the reviewers’ major concerns here have been thoroughly and completely addressed. Really excellent job. This is a major improvement over previous versions.

However, I believe that the one remaining issue in this paper is a major problem, and likely an unfixable one: scales like NOS are simply not minimally adequate measures of study quality. At best they are measures of reporting, not study strength or quality. I recognize that they are often used in the literature as such and are relatively quick, as the authors have noted, but commonly performed and quick do not indicate quality analysis.

I should note that the authors cite a paper for systematic review guides and tools from 2007 as an indication that no tools exist for the job. That is clearly incorrect; there have been at least 2 such tools published and used in systematic review for study strength since then, the best known of which is the ROBINS-I tool. They require a great deal more time and expertise to use than the scales used in this paper, but they are the standard for comparison. Quality review is extremely difficult, and I do not believe we are well-served by calling an index of superficial reporting as indicating quality.

At minimum, I would strongly recommend reframing this paper from being one about quality, strength, etc, and to being about minimal reporting guidelines. If the authors wish to review quality or strength and make that claim, tools appropriate for the task should be used.

of “quality.”

REVIEWER COMMENTS

Reviewer #1 (Remarks to the Author):

I thank the authors for the comprehensive replies to my comments, questions and suggestions.

We thank the reviewer for the kind words.

1. Original point #5. I'm not sure where this information provided has been inserted into the methods? I found the response to be incredibly difficult to navigate as the authors pasted whole sections of the article, rather than simply highlighting the specific text that had been revised. In the future, I would suggest simply highlighting the revised text. Again, I recognize that there are limitation in word count for the manuscript, however this information could be included in supplemental methods online. The level of training suggested by the authors speaks to rigor of the study and should be transparently reported.

We apologize for the confusion created by inserting the entire methods section of the article. We have added the additional information provided in original point #5 and updated the methods section on **page 6** and below.

“The review team consisted of a trained research staff with expertise in systematic reviews and one trainee. Title and abstracts were evaluated by two independent reviewers using Covidence and all discrepancies were resolved by consensus. Articles that were selected for full review were independently evaluated by two reviewers for quality assessment using a standardized case report form following the completion of a training period where all reviewers were trained with the original manuscripts which derived the tools or checklists along with examples for what were deemed high scores.^{1, 2, 3, 4} Following this, reviewers completed thirty full-text extractions and the two reviewers had to reach consensus and the process was repeated for the remaining manuscripts independently. When two independent reviewers were not able reach consensus, a third reviewer (principal investigator) provided oversight in the process to resolve the conflicted scores.”

2. Original point #8. Timing of the registration must be transparently reported, especially since the authors suggest it was incorrectly registered (i.e. the oversight they report in the response).

We thank the reviewer for the suggestion of including the PROSPERO registration in the manuscript. We have included the registration date (June 3, 2020) in the methods section on **page 5** and below.

“Methods

A systematic literature search was conducted on May 14, 2020 (registered on June 3, 2020 at **PROSPERO: CRD42020187318**) and reported according to the Preferred Reporting Items for Systematic Reviews and Meta-Analyses. Furthermore, the cohort study was reported according to the Strengthening The Reporting of Observational studies in Epidemiology checklist. The data supporting the findings of this study is available in the Supplemental Appendix.”

3. Original point #17: Thank-you for clarifying. As per original point #5 these details should be reported somewhere. I’m not sure why the authors are self-citing their previous studies in relation to a kappa score? There is no need for this as this has nothing to do with previous studies by the authors.

We thank the reviewer for the response. We have added additional detail with regards to the level of training of all the independent reviewers for identification of COVID-19 along with methodological quality assessment of the manuscripts. Furthermore, we had initially provided the kappa-values in reference to our other manuscripts as evidence for consistency of the kappa-values but have removed the references to avoid self-citation. The additional method provided is found on **page 6** and below.

“The review team consisted of a trained research staff with expertise in systematic reviews and one trainee. Title and abstracts were evaluated by two independent reviewers using Covidence and all discrepancies were resolved by consensus. Articles that were selected for full review were independently evaluated by two reviewers for quality assessment using a standardized case

report form following the completion of a training period where all reviewers were trained with the original manuscripts which derived the tools or checklists along with examples for what were deemed high scores.^{1, 2, 3, 4} Following this, reviewers completed thirty full-text extractions and the two reviewers had to reach consensus and the process was repeated for the remaining manuscripts independently. When two independent reviewers were not able reach consensus, a third reviewer (principal investigator) provided oversight in the process to resolve the conflicted scores.”

4. Original point #19: Reasons for omission of RCT analysis should be reported in the paper so the reader understands what happened. I’m not clear how reasons #1 (“summary score”) and #2 (“collective score”) are different? Also, it’s unclear why the authors then chose this tool to begin with if the initial plan was to perform a statistical analysis.

The reviewer correctly recommends the inclusion of rationale for not analyzing randomized controlled trials in the study. The main reason for not comparing the quality in RCTs was largely due to sample size limitations (n=6). The second reason (“summary scores”) was recommended by Reviewer 2 as a reason to not conduct a comparison analysis. The primary outcome of interest was to evaluate the quality of COVID-19 clinical literature using the four tools corresponding to the study design. As such, we believed it would be important to highlight the findings of study design whereas the comparison analyses were secondary outcomes. We revised the statistical analysis section in the manuscript to reflect that RCTs were not included due to the small sample size (n=6) and is found on **page 8** and below.

“The primary outcome of interest was to evaluate the quality of COVID-19 by study type using Newcastle-Ottawa Scale (**NOS**) for case-control and cohort studies, QUADAS-2 tool for diagnostic studies,⁴ Cochrane Risk of Bias for RCTs,¹ and a score derived by Murad et al. for case series.² Pre-specified secondary outcomes were comparison of quality scores by: i) median time to acceptance, ii) impact factor, iii) geographical region, and iv) historical comparator. Time of acceptance was defined as the time between submission to acceptance which captures peer

review and editorial decisions. Geographical region was stratified into continents including Asia/Oceania, Europe/Africa, and Americas (North and South America). *Post-hoc* comparison analysis between COVID-19 and historical control article quality scores were evaluated using Kruskal-Wallis test. Furthermore, good quality of NOS was defined as 3+ on selection and 1+ on comparability and 2+ on outcome/exposure domains and high-quality case series scores was defined as a score ≥ 3.5 . Due to a small sample size of identified RCTs, they were not included in the comparison analysis.”

5. Page 13: I would disagree with the sentence, “Overall, the accelerated publication of COVID-19 research negatively affected the study quality scores compared to previously published historical control studies.” This sentence implies causation. Association is the strongest conclusion that can be made with the observational design of this study.

We thank the reviewer for the suggestion of changing the summative sentence of the discussion regarding our findings. We have revised the sentence to reflect the reviewers recommendations with the following:

“Overall, the accelerated publication of COVID-19 research was associated with lower study quality scores compared to previously published historical control studies.”

MINOR:

1. Page 10: “improper study design” – I’m not sure ‘improper’ is the correct term (i.e. case series might have been the correct/proper design for some of the studies you screened), and editorials are not studies. As such, I suggest “...ineligible study design or publication type...”

The reviewer is correct and we have revised the sentence to “ineligible study design or publication type”.

2. Page 10: “1.0% calculated sample size” – Not all these study types would be expected to have calculated sample sizes. For instance, case series would not. The denominator for this particular statistic should only be the study designs that would be expected to able to perform a power calculation.

The reviewer is correct, and we apologize for the oversight. We have recalculated the sample size calculations without case series and the results with appropriate sample size is found on **Table 1**.

3. The lack of ethics approval (only 81% had approval, or at least reported it) seems low. Would all these study types require ethics approval?

We agree with the reviewer when they mention that 81.0% ethics approval appears to be low in COVID-19 clinical literature. Indeed, we had previously sub-categorized the information for “not mentioned, not received, and not required”. Indeed, 556 (81.0%) received ethical approval, 91 (13.3%) did not mention ethical approval, and 39 (5.7%) did not require or receive ethics (ie. epidemiological/population studies). Oddly enough, many case series did not report their ethical approval, which also lends credence to the low methodological quality reporting evident in COVID-19 manuscripts.

When COVID-19 articles were compared to historical controls (n=1078), the following was observed:

- COVID-19 articles:
 - o 433 (80.3%) received ethical approval
 - o 73 (13.5%) did not require/receive ethical approval

- 33 (6.1%) did not mention ethical approval
- Historical control articles:
 - 451 (83.7%) received ethical approval
 - 60 (11.1%) did not require/receive ethical approval
 - 28 (4.1%) did not mention ethical approval

4. Page 11: sentence does not seem to be grammatically correct? “Furthermore, of the identified 6 RCTs ...they remain at high risk of bias...”

We have revised the sentence for grammatical purposes and is the following:

“Furthermore, in the 6 RCTs in the COVID-19 literature, there was a high risk of bias with little consideration for sequence generation, allocation concealment, blinding, incomplete outcome data, and selective outcome reporting (**Table 2**).”

5. Page 15: “they can only assess the methodology without consideration for causal language and prone to limitations.” I wasn’t clear what ‘causal language’ meant in this context. Also, I believe there should be an “are” inserted before “prone”.

We thank the reviewer for the clarification. Indeed for “causal inference”, we are inferring to the inherent limitation present within methodological quality tools which cannot evaluate the causal language used in studies evaluating the effect of these interventions (ie. case series evaluating drug X but the conclusion of the study contains causal language). We have added the word “are” before the word prone in the limitations as well.

Reviewer #2 (Remarks to the Author):

I thank the authors for their substantial revision and rebuttal.

1. The responses to my comments on methods are ok, although I thought it a bit weak that kappa for inter-rater agreement could not be calculated because of failures of the Covidence software.

The reviewer raises an interesting point regarding the lack of kappa values for inter-rater agreement for title and abstract screening for identification of COVID-19 clinical manuscripts. Generally, kappa values for inter-rater agreements of title and abstract screening are not generated for systematic reviews. As previously mentioned, from screening of 9895 title and abstracts, the reviewers selected 794 texts for full-review and had conflicts in 330 texts which were resolved by consensus (96.7% agreement rate). Since we do not know the direction of the disagreement, we opted out in reporting the kappa values within the manuscript. If we assume all the disagreement came from one reviewer, it would generate a kappa value of 0.81 (95% CI, 0.79-0.83) which falls in the range of “almost perfect agreement”. With equal disagreement (155 in each reviewer), this would generate a kappa value of 0.809 (95% CI, 0.790-0.829), also considered “almost perfect agreement”. Most importantly, we report agreement between two independent reviewers for quality assessment. For full text extraction and quality assessment, the general agreement between the reviewers for quality assessment was 86.5% ($\kappa=0.68$; 95% CI, 0.67-0.70), which correlates to substantial agreement between the reviewers. We have added both kappa values for inter-rater agreements in the manuscript and is found on **page 9** and below.

Results

Article selection

A total of 14787 COVID-19 papers were identified as of May 14, 2020 and 4892 duplicate articles were removed. 9895 titles and abstracts were screened, and 9101 articles were excluded due to the study being pre-clinical in nature, case report, case series <5 patients, in an language other than English, reviews (including systematic reviews), study protocols or methods, and other coronavirus variants with an overall inter-rater study inclusion agreement of 96.7%

($\kappa=0.81$; 95% CI, 0.79-0.83). A total number of 794 full texts were reviewed for eligibility. Over 108 articles were excluded for ineligible study design or publication type (such as letter to the editors, editorials, case reports, or case series <5 patients), patient population, non-English language, duplicates, wrong outcomes, and publication in a non-peer reviewed journal. Ultimately, 686 articles were identified and underwent methodological quality assessment with an inter-rater agreement of 86.5% ($\kappa=0.68$; 95% CI, 0.67-0.70) (**Figure 1**).”

2. The responses to my comments on reporting are broadly ok. The text in Results still restates what can be read in the tables and figures, but there is now a good match between text and tables/figures.

We thank the reviewer for the comment.

3. The responses to my comments on interpretation are not adequate.

a. For my point 3a, I thought I was quite clear: a lower methodological quality and faster time to publication is not necessarily a bad thing. The authors respond that I am wrong because their data show that lower methodological quality is associated with faster time to publication. This misses the point, for two reasons. First, it is a correlation and does not imply causation. Second, even if it did imply causation, so that we could improve methodological quality by longer times to publication, it does not follow that this would have improved the world's response to covid-19. To put it very crudely, is it better to have poor quality evidence or no evidence at all? So the authors' conclusion that "the data obtained in these studies should be revisited" is entirely valid, but their implication that the scientific community was wrong to accept lower-quality rapidly-published research is not valid.

The reviewer asks whether it is better to have poor quality evidence or no evidence at all and questions our implication that the scientific community was wrong to accept the lower-quality, rapidly published COVID-19 research in the early phases of the pandemic. This is undoubtedly to be debated but we would argue, strongly, that COVID will again demonstrate that rapidly published poor quality studies simply distracts, wastes resources, and in some instances directly harms patients.

While it is tempting to cut scientific corners when faced with situations such as pandemics, remains a need to maintain quality in order to inform future research. We have added a paragraph discussing the reviewers view, in that rapid time to publication allowed us to understand the symptomology and prognosis of COVID-19, identification of tools to diagnose SARS-CoV-2,⁵ and identify potential therapeutic options (ie. tocilizumab and convalescent plasma) which laid the foundation for RCTs (current topics of the RECOVERY Trial group).^{6, 7, 8} We hope the reviewer agrees with the new paragraph and our final concluding sentences that while initial expedition to publication exists, these data should serve largely to inform further research rather than practice and should be revisited as stronger data emerges (ie. hydroxychloroquine now fully debunked with the latest RECOVERY trial findings). Interestingly, major findings which dramatically impacted our clinical care were initially presented in preprint servers (ie. Dexamethasone in RECOVERY Trial) or through media release (ie. Remdesivir for ACTT-1 Trial) to allow dissemination of clinical findings while permitting a thorough traditional peer review process to take place.^{9, 10, 11} We hope that we have presented

both sides of the argument and that the reviewer agrees with the revised discussion found on **pages 12-13** and below.

Where we disagree with the reviewer is in that we should strive for high-quality research even in the times of difficulty, but we understand the ethical dilemma present in this situation. History often repeats itself, for example during the Ebola epidemic in West Africa six years ago. Public health interventions such as quarantine, social distancing, and travel restrictions which are prevalent in the COVID-19 pandemic was also in place during the Ebola epidemic.¹² During the Ebola epidemic, ZMapp was an immunotherapy of three human monoclonal antibodies produced in *Nicotinia* plants against the Ebola virus and was scarcely available.¹³ Indeed, Dr. Joffe posed an interesting ethical dilemma at the time of what the best use of this limited ZMapp antibody cocktail would be. He states individuals may be tempted to use this experimental drug first in a case series (low quality evidence) to identify the safety and efficacy of the drug against Ebola. A randomized trial would allow us to definitively establish if this drug works (or causes harm) and to appropriately allocate resources.

Similar situations are present in the COVID-19 pandemic, where numerous observational studies were published evaluating therapeutic options against SARS-CoV-2. As we have highlighted in the manuscript, hydroxychloroquine is the best example of a poorly designed observational study gone awry requiring to numerous observational studies and randomized controlled trials to debunk its efficacy. Furthermore, the largest study (35,322 patients) evaluating convalescent plasma has drawn a lot of criticisms due to its observational nature which does not allow for any conclusive statement of the true efficacy of convalescent plasma in this population which could have been answered with randomization of even a fraction of the 35,322 patients enrolled in the study.¹⁴ The best therapeutic option against COVID-19 remains dexamethasone, which to our knowledge, was not preceded by lower quality observational study, but rather the RECOVERY Trial definitely answered the therapeutic efficacy of dexamethasone in the critically ill COVID-19 population with a properly designed randomized controlled trial with rapid dissemination through preprint servers.¹¹

“In the early stages of the COVID-19 pandemic, an urgent need for scientific data to inform clinical, social and economic decisions led to shorter time to publication and explosion in publication of COVID-19 studies in both traditional peer-reviewed journals and preprint servers.^{15, 16} The accelerated scientific process in the COVID-19 pandemic allowed a rapid understanding of natural history of COVID-19 symptomology and prognosis, identification of tools including RT-PCR to diagnose SARS-CoV-2,⁵ and identification of potential therapeutic options such as tocilizumab and convalescent plasma which laid the foundation for future RCTs.^{6, 7, 8} During the worldwide pandemic, a delay in publication due to a slower peer review process potentially delays the dissemination of information against the COVID-19 pandemic. Despite concerns of slow peer review, the trial groups of major landmark trials (ie. RECOVERY and ACTT-1 trial),^{10, 11} published their findings in preprint servers and media releases to allow for rapid dissemination while permitting a traditional peer review process to take place. Furthermore, despite the initial expedition to publication to better understand the pandemic, the data obtained in these initial studies are best revisited as stronger data emerges as lower quality studies can fundamentally risk patient safety, resource allocation, and future scientific research.^{17”}

b. My point 3b criticised the writing of the abstract, and I find the revised abstract even harder to read.

We apologize for the confusion and hope that the revised abstract found on **page 3** and below presents our findings in a coherent fashion.

“Abstract

Background: COVID-19 spread globally in early 2020 with major health consequences. While a need to disseminate information to the medical community and general public was paramount -

concerns have been raised regarding the scientific rigor in published reports. We sought to evaluate the methodological quality of currently available COVID-19 studies compared to historical controls.

Methods: We performed a systematic search of MEDLINE, Embase, and Cochrane Central Register of Controlled Trials until May 14, 2020. All clinical literature evaluating COVID-19 were identified and 1:1 historical control of the same study design in the identical journal was matched from the previous year. Two independent reviewers screened titles, abstracts, full-texts, and independently assessed the methodological quality.

Results: 9895 titles and abstracts were screened, and 686 COVID-19 articles were included in the final analysis. Overall, COVID-19 articles had a mean case series quality score (out of 5) (\pm SD) of 3.3 (1.1), mean NOS cohort study score (out of 8) of 5.8 (1.5), mean NOS case control study score (out of 8) of 5.5 (1.9), and low risk of bias present in 4 (6.4%) diagnostic studies. Comparative analysis of COVID-19 to historical articles reveal a difference in time to acceptance (13.0 [IQR, 5.0-25.0] days vs. 110.0 [IQR, 71.0-156.0] days in COVID-19 and control articles, respectively; $p < 0.0001$). Furthermore, methodological quality scores were lower in COVID-19 articles in all study designs compared to control articles.

Conclusion: COVID-19 clinical studies had a shorter time to publication and had lower methodological quality scores than control studies published in the same journal. These studies should be revisited with the emergence of stronger evidence.”

i. The 2nd sentence of the results part in particular makes little sense.

We apologize for the confusion and hope that the revised results section presents our findings in a coherent fashion.

ii. The results part twice repeats the methods, and the conclusion partly repeats the results.

We have removed redundancy present in the abstract and hope the reviewer agrees.

iii. The aim stated in the background is just the 3rd of the 3 aims stated in the main paper Introduction, yet the results part gives almost no detail of this 3rd aim and lots of detail about the 1st and 2nd aims, so that it reads as not answering the stated aim.

We thank the reviewer for the suggestion to further elaborate on the results of aim 3 in the introduction of the manuscript. Aims 1-3 each received one paragraph of our findings where we highlight overall COVID-19 quality in paragraph 3 of the results (aim 1), secondary outcomes by impact factor, time to publication, and geographical region in paragraph 4 of the results (aim 2), and comparison to historical control in paragraph 5 of the results (aim 3). While we agree aim 3 is the most important finding to this manuscript, we believe all the pertinent results has been presented in aim 3 in paragraph 5 of the results.

iv. There remain a number of grammatical errors.

We have made changes in the manuscript and hope you agree that we have made significant grammatical changes.

4. The question of the overall aims is difficult. When I first read the paper, I thought it was mainly about comparing covid-19 research with other research. But I now see this is just the 3rd of three aims. So my previous comments 2b and 3c were wrong (sorry). However I don't find aims 1 and 2 particularly interesting, and I would have placed aim 3 as primary. So the paper remains largely focussed on aims that are not very interesting. Further, the (commendable) shift of emphasis from dichotomising quality to quantifying quality makes the reporting of aim 1 even less interesting – most readers (like me) will be unable to interpret e.g. “a mean case series score of 3.3” as good or bad, and will only be able to understand it from the comparison with the matched control papers.

We agree with the reviewer that aim 3 is the most important/interesting findings from the manuscript. Indeed, we had reported the findings in this fashion as it was determined *a priori* following our protocol registered on PROSPERO. The reviewer is also correct in that the shift from dichotomized quality score to a numerical score does not allow for a direct interpretation of COVID-19 quality and that comparison to historical control articles demonstrates reduced

quality scores of COVID-19 manuscripts. We have reported all the findings of primary and secondary outcomes pre-specified in PROSPERO within the results section but revised the abstract to highlight the comparison of methodological quality scores between COVID-19 to historical control articles. We hope you agree with the changes in the abstract and results section.

5. Table 2 has big problems.

We thank the reviewer for all the concerns regarding Table 2 and hope we have addressed all the concerns.

a. The mean and SD of the quality scores are reported in columns wrongly headed “N” and “%”. One approach here is to drop the column heading and label the rows “N (%)” or “mean (SD)”.

We have removed the row heading of N (%) and have specified the findings within the row title.

b. For the yes/no characteristics, it is not enough to report N when the denominator is not given. E.g. the first N is 172 but the denominator is not the 686 stated at the top. (It’s actually 380, found in table 1, but table 2 should be self-sufficient.)

The reviewer is correct, we have added rows specifying the sample size of each.

c. the study types should be listed as well as their associated quality scores

We have added the study types with the associated quality scores on the left.

d. “+/-“ should be avoided, here and elsewhere. “Mean (SD)” is better.

We have revised the +/- with mean (SD).

6. Smaller points

a. My previous point 14 – median is still wrongly used at line 212 (just delete “median” here).

We have revised the manuscript to delete median from median time to acceptance at line 212.

b. I was amused by “The primary outcome of interest was to evaluate the quality of COVID-19 ...” (L163) – but this is also another example of poor writing.

We have revised the manuscript and hope the reviewer agrees with the improved quality.

Reviewer #3 (Remarks to the Author):

This overall has been a very thorough effort by the original authors to make very substantial changes to the manuscript, and respond very clearly to the reviewers. In fact, with one exception, I believe that all of my and the rest of the reviewers' major concerns here have been thoroughly and completely addressed. Really excellent job. This is a major improvement over previous versions.

We thank the reviewer for the kind words. We hope that the following response adequately answers the remaining issue.

However, I believe that the one remaining issue in this paper is a major problem, and likely an unfixable one: scales like NOS are simply not minimally adequate measures of study quality. At best they are measures of reporting, not study strength or quality. I recognize that they are often used in the literature as such and are relatively quick, as the authors have noted, but commonly performed and quick do not indicate quality analysis.

I should note that the authors cite a paper for systematic review guides and tools from 2007 as an indication that no tools exist for the job. That is clearly incorrect; there have been at least 2 such tools published and used in systematic review for study strength since then, the best known of which is the ROBINS-I tool. They require a great deal more time and expertise to use than the scales used in this paper, but they are the standard for comparison. Quality review is extremely difficult, and I do not believe we are well-served by calling an index of superficial reporting as indicating quality.

At minimum, I would strongly recommend reframing this paper from being one about quality, strength, etc, and to being about minimal reporting guidelines. If the authors wish to review quality or strength and make that claim, tools appropriate for the task should be used.

The reviewer highlights the limitations which exists in the field of evaluating study quality and risk of biases from identified manuscripts in a systematic review. Furthermore, the reviewer states that Newcastle-Ottawa Scale is a measure of reporting and not study strength nor quality and recommends the use of the ROBINS-I (Risk Of Bias In Non-randomized Studies – of Interventions) tool as the only tool available for the assessment of literature quality.¹⁸ ROBINS-I is more comprehensive than NOS, including points given for blinding of personnel, greater weighing of outcomes/exposures, and selective reporting of outcomes. Despite this, these two tools also have great overlap in patient selection, assessment of outcomes, and follow-up or handling of missing data (part of outcomes in NOS). In the paragraphs below, we will highlight our rationale for not using the ROBINS-I tool for COVID-19 and we have reframed the manuscript title/abstract to reflect methodological reporting quality, although we respectfully

disagree with the reviewer with findings from previous manuscripts evaluating critical appraisal in that Newcastle-Ottawa Scale is not usable as a measure of methodological quality.

First, the reviewer recommends the use of the ROBINS-I tool for quality assessment. The reviewer is correct in that other quality tools have been generated since the critical assessment by Sanderson et al.¹⁹ in 2007 including the GRADE and ROBINS-I tool.^{20, 21} The ROBINS-I tool designed to evaluate the risk of bias in non-randomized studies which compares 2 or more interventions (from the detailed guidance). The risk of bias is assessed in seven domains including confounding variables, patient selection, interventions, missing data, outcomes, and selection of reported results. Indeed, the risk of bias assessed from these non-randomized studies include cohort studies, case-control studies, interrupted time-series studies, controlled before-and-after studies, and quasi-randomized studies. Unfortunately, the ROBINS-I tool does not appear to allow for the assessment of case series studies (without a comparison group and often lack interventions). In our study, case-series studies comprise over 380/686 (55.4%) of COVID-19 articles and 554 (51.4%) of the comparison analysis in our manuscript which would lead to the exclusion of over half of our identified manuscripts. Secondly, ROBINS-I requires pre-specification of key confounders, comparisons, and interventions prior to initiation of the study itself. At the time of the study inception in May 2020, we had minimal clinical knowledge of the natural history of COVID-19, confounding variables, and intervention options in order to pre-specify these findings that we now know in hindsight (ie. impact of obesity, age, and other comorbidities on COVID-19 outcomes and dexamethasone now being the gold-standard therapy against critically ill COVID-19 patients).

Although there are purported advantages of the ROBINS-I tool, it remains a relatively new tool without validation unlike that of the Newcastle-Ottawa Scale.²² Furthermore, a recent critical appraisal by Quigley et al. demonstrates that Newcastle-Ottawa Scale was the most commonly used tool for evaluation of methodological quality and concludes that no consensus currently exists within health technology assessment agencies for the critical appraisal of evidence of non-randomized studies.²² Indeed, in their assessment of 686 systematic reviews, 142 (21%) studies utilized the Newcastle-Ottawa Scale and the ACROBAT-NRSI/ROBINS-I tool was used in 18 (3%) of the identified studies.²² Moreover, concerns have been raised regarding the poor inter-

rater reliability of ROBINS-I even amongst experienced public health researchers with moderate to extensive experience in critical appraisal.²³ Specifically, although ROBINS-I allowed a systematic identification of the risk of biases in these studies, the lack of consensus in patient selection and deviations from intended interventions was concerning with authors specifically noting that the tool requires complex epidemiological knowledge that is beyond the capacity of many systematic review author groups.²³

The reviewer is concerned with the use of the word “quality” in our manuscript as we have assessed. Several appraisals of methodological quality tools refer to the Newcastle-Ottawa Scale as a potential and the most popular option for assessing quality in systematic reviews.^{19, 22} Ironically, even the original ROBINS-I article refers to the Newcastle-Ottawa scale as a tool to assess methodological quality of observational studies.²¹ Assessment of quality continues to be contested and there is no established method for any study type deemed to be a gold standard (although many groups/authors are striving to establish specific tools as such).²² However, even the proposed ROBINS-I would have excluded most studies, lacks robust validation and has poor reproducibility.

We have added to the discussion the nuanced differences in the limitations on **pages 14-15** and below and hope that the reviewer can accept his acknowledgement.

“Our study has important limitations. We evaluated the methodological quality of existing studies using established checklists and tools. While it is tempting to associate methodological quality scores with reproducibility or causal inferences of the intervention, it is not possible to ascertain the impact on the study design and conduct of research nor results or conclusions in the identified reports.²⁴ Second, although the methodological quality scales and checklists used for the manuscript are commonly used for quality assessment in systematic reviews and meta-analyses,^{1, 2, 3, 4} they can only assess the methodology without consideration for causal language and are prone to limitations.^{19, 25} Other tools such as the ROBINS-I and GRADE exist to evaluate

methodological quality of identified manuscripts, although no consensus currently exists for critical appraisal of non-randomized studies.^{20, 21, 22} Furthermore, other considerations of quality such as sample size calculation, sex reporting, or ethics approval is not considered in these quality scores. As such, the quality scores measured using these checklists only reflect the patient selection, comparability, diagnostic reference standard, and methods to ascertain the outcome of the study. Third, the 1:1 ratio to identify our historical control articles may affect the precision estimates of our findings. Interestingly, a simulation of an increase from 1:1 to 1:4 control ratio tightened the precision estimates but did not significantly alter the point estimate.²⁶ Furthermore, the decision for 1:1 ratio exists due to limitations of available historical control articles from the identical journal in the restricted time period combined with large effect size and sample size in the analysis. Finally, our analysis includes early publications on COVID-19 and there is likely to be an improvement in quality of related studies and study design as the field matures and higher quality studies, which take longer to design, conduct, and report are published. Accordingly, our findings are limited to the early body of research as it pertains to the pandemic and it is likely that over time research quality will improve.”

References

1. Higgins JPT, *et al.* The Cochrane Collaboration's tool for assessing risk of bias in randomised trials. *BMJ* **343**, d5928 (2011).
2. Murad MH, Sultan S, Haffar S, Bazerbachi F. Methodological quality and synthesis of case series and case reports. *BMJ Evid Based Med* **23**, 60-63 (2018).
3. Wells G SB, O'Connell D, Peterson J, Welch V, Loso M, Tugwell P. The Newcastle-Ottawa Scale (NOS) for assessing the quality of nonrandomised studies in meta-analysis. http://www.ohrica/programs/clinical_epidemiology/oxfordasp, (2004).
4. Whiting PF, *et al.* QUADAS-2: a revised tool for the quality assessment of diagnostic accuracy studies. *Ann Intern Med* **155**, 529-536 (2011).
5. He X, *et al.* Temporal dynamics in viral shedding and transmissibility of COVID-19. *Nature Medicine* **26**, 672-675 (2020).
6. Duan K, *et al.* Effectiveness of convalescent plasma therapy in severe COVID-19 patients. *Proceedings of the National Academy of Sciences* **117**, 9490-9496 (2020).
7. Guaraldi G, *et al.* Tocilizumab in patients with severe COVID-19: a retrospective cohort study. *The Lancet Rheumatology* **2**, e474-e484 (2020).
8. Shen C, *et al.* Treatment of 5 Critically Ill Patients With COVID-19 With Convalescent Plasma. *JAMA* **323**, 1582-1589 (2020).
9. Effect of Hydroxychloroquine in Hospitalized Patients with Covid-19. *New England Journal of Medicine*, (2020).
10. Beigel JH, *et al.* Remdesivir for the Treatment of Covid-19 — Final Report. *New England Journal of Medicine*, (2020).
11. Group RC, *et al.* Dexamethasone in Hospitalized Patients with Covid-19 - Preliminary Report. *N Engl J Med*, (2020).
12. Gostin LO, Lucey D, Phelan A. The Ebola Epidemic: A Global Health Emergency. *JAMA* **312**, 1095-1096 (2014).
13. Joffe S. Evaluating Novel Therapies During the Ebola Epidemic. *JAMA* **312**, 1299-1300 (2014).
14. Joyner MJ, *et al.* Effect of Convalescent Plasma on Mortality among Hospitalized Patients with COVID-19: Initial Three-Month Experience. *medRxiv*, 2020.2008.2012.20169359 (2020).

15. Chen Q, Allot A, Lu Z. Keep up with the latest coronavirus research. *Nature* **579**, 193 (2020).
16. Bauchner H. The Rush to Publication: An Editorial and Scientific Mistake. *JAMA* **318**, 1109-1110 (2017).
17. Ramirez FD, *et al.* Methodological Rigor in Preclinical Cardiovascular Studies: Targets to Enhance Reproducibility and Promote Research Translation. *Circ Res* **120**, 1916-1926 (2017).
18. Hartling L, *et al.* Risk of bias versus quality assessment of randomised controlled trials: cross sectional study. *BMJ* **339**, b4012 (2009).
19. Sanderson S, Tatt ID, Higgins JP. Tools for assessing quality and susceptibility to bias in observational studies in epidemiology: a systematic review and annotated bibliography. *Int J Epidemiol* **36**, 666-676 (2007).
20. Guyatt G, *et al.* GRADE guidelines: 1. Introduction-GRADE evidence profiles and summary of findings tables. *J Clin Epidemiol* **64**, 383-394 (2011).
21. Sterne JA, *et al.* ROBINS-I: a tool for assessing risk of bias in non-randomised studies of interventions. *BMJ* **355**, i4919 (2016).
22. Quigley JM, Thompson JC, Halfpenny NJ, Scott DA. Critical appraisal of nonrandomized studies-A review of recommended and commonly used tools. *Journal of Evaluation in Clinical Practice* **25**, 44-52 (2019).
23. Thomson H, Craig P, Hilton-Boon M, Campbell M, Katikireddi SV. Applying the ROBINS-I tool to natural experiments: an example from public health. *Syst Rev* **7**, 15 (2018).
24. Glasziou P, Chalmers I. Research waste is still a scandal—an essay by Paul Glasziou and Iain Chalmers. *BMJ*, k4645 (2018).
25. Stang A. Critical evaluation of the Newcastle-Ottawa scale for the assessment of the quality of nonrandomized studies in meta-analyses. *Eur J Epidemiol* **25**, 603-605 (2010).
26. Hamajima N, Hirose K, Inoue M, Takezaki T, Kuroishi T, Tajima K. Case-control studies: matched controls or all available controls? *J Clin Epidemiol* **47**, 971-975 (1994).

Reviewer comments, third round:

Reviewer #1 (Remarks to the Author):

Thank-you for addressing all my queries and suggestions. I only have two minor points to raise.

1. NOS appears in the abstract and is not defined. I would simply remove NOS and keep the score nameless in the abstract (which would be inline with the other scores mentioned in the abstract).

2. The paragraph in the discussion, modified to address reviewers 2/3 is somewhat problematic.
- "In the early stages of the COVID-19 pandemic, an urgent need for scientific data to inform clinical, social and economic decisions led to shorter time to publication and explosion in publication of COVID-19 studies in both traditional peer-reviewed journals and preprint servers.^{1, 16}"

Although I do believe the authors are correct about this, the reasons provided are speculative. This should be plainly stated.

- "During the worldwide pandemic, a delay in publication due to a slower peer review process potentially delays the dissemination of information against the COVID-19 pandemic.: - this read a bit awkwardly.

- "...while permitting a traditional peer review process to take place." This could be perceived as an insinuation that peer review didn't take place in the other articles. Moreover, the RCTs cited (21, 22) were accepted very rapidly (e.g. NEJM trial: April 20 trial completed, May 22 accepted and first version published by journal).

- "Furthermore, despite the initial expedition to publication to better understand the pandemic, the..." This sentence is rather long and difficult to navigate. Suggest editing.

Reviewer #2 (Remarks to the Author):

My remaining comments refer to the abstract and Table 2 only.

Abstract

1. The hyphen on line 46 should be a comma.

2. L52, "1:1 historical control" is not logical. I suggest "Each clinical paper evaluating COVID-19 was matched 1:1 to a historical control of the same study design in the same journal from the previous year."

3. L56-58 wrongly implies that every COVID-19 article had a mean case series quality score (etc.). I suggest "COVID-19 articles reporting case series had a mean quality score (out of 5) (SD) of 3.3 (1.1); cohort studies, mean NOS score (out of 8) of 5.8 (1.5); case control studies, mean NOS score (out of 8) of 5.5 (1.9); and diagnostic studies, low risk of bias present in 4 (6.4%)." By the way "±" is discouraged and unnecessary.

Table 2

4. "All" is unclear in the heading, need to say it's "all COVID-19": it would be best to use exactly the same headings as in table 1.

5. The use of "no. (%)" with "mean (SD)" beneath it is of minimally acceptable quality. It would be much clearer if the authors just put % as appropriate e.g. in row 2 change (45.3) to (45.3%).

6. Percentages should be rounded to the nearest 1%, because they don't have even this level of accuracy (due to small samples): so the result above would be best as (45%).

7. "Case-series design" can just be "Case series". Later also "Case-control study", "Cohort study", "Diagnostic study" (why plural?), "Randomised controlled trial".

8. In both tables 1 and 2, please make clear what the p-value represents e.g. "P-value, COVID-19 vs Control".

9. "n (%)" isn't needed & could be confusing in the subheadings under "Diagnostic studies - no. (%)". Please remove it.

Reviewer #3 (Remarks to the Author):

While I retain the majority of my previous concerns regarding the use of the Newcastle-Ottawa scale, I have found the language changes to be sufficient for publication. The Newcastle-Ottawa scale is not justifiable just because it is more common and faster. We cannot and should not excuse weaker methods just because they are common and fast does not improve methodological rigor, as is clearly true for COVID-19 studies. I would strongly have preferred use of a more sensible scale, and lacking that a much more details accounting of the limitations of the scales used.

However, the revised manuscript no longer refers to studies as being high quality, and focuses almost entirely on the differences in scores between COVID and non-COVID research. Comparing differences in scores, rather than assigning meaning to the scores themselves, is more broadly acceptable in this regard. As the paper is now written and framed, it is no longer an existential threat to the validity of the claims being made.

I have generally found the responses to the reviewers to be well-written, and the changes made appropriate. I would also like to strongly disagree with Reviewer #2's response to 3a, which appears to be largely based in a philosophical stance regarding what is acceptable about evidence, and not a reflection of the findings of this research. R2 appears to be taking the stance that there is no use to this research because it is acceptable to have weaker methods in emerging situations (personally, I believe the opposite to be true, where we should increase the standards of our methodological rigor when the decisions are more important, not decrease). Regardless of that preference, it is critical to document the strength of that research so that decision-makers are better able to determine whether they are useful for decision-making. I strongly recommend the editors and authors disregard that comment.

REVIEWERS' COMMENTS

Reviewer #1 (Remarks to the Author):

Thank-you for addressing all my queries and suggestions. I only have two minor points to raise.

- 1. NOS appears in the abstract and is not defined. I would simply remove NOS and keep the score nameless in the abstract (which would be inline with the other scores mentioned in the abstract).**

We thank the reviewer for the feedback which has greatly strengthened the manuscript overall. We have greatly revised the abstract to meet the requirements of the journal and can be seen below and on page 3 of the manuscript.

“The COVID-19 pandemic began in early 2020 with major health consequences. While a need to disseminate information to the medical community and general public was paramount - concerns have been raised regarding the scientific rigor in published reports. We performed a systematic review to evaluate the methodological quality of currently available COVID-19 studies compared to historical controls. A total of 9895 titles and abstracts were screened, and 686 COVID-19 articles were included in the final analysis. Comparative analysis of COVID-19 to historical articles reveals a shorter time to acceptance (13.0[IQR, 5.0-25.0] days vs. 110.0[IQR, 71.0-156.0] days in COVID-19 and control articles, respectively; $p < 0.0001$). Furthermore, methodological quality scores were lower in COVID-19 articles in all study designs. COVID-19 clinical studies had a shorter time to publication and had lower methodological quality scores than control studies in the same journal. These studies should be revisited with the emergence of stronger evidence.”

- 2. The paragraph in the discussion, modified to address reviewers 2/3 is somewhat problematic.**

- "In the early stages of the COVID-19 pandemic, an urgent need for scientific data to inform clinical, social and economic decisions led to shorter time to publication and explosion in publication of COVID-19 studies in both traditional peer-reviewed journals and preprint servers.1, 16" Although I do believe the authors are correct about this, the reasons provided are speculative. This should be plainly stated.

We have added the phrase “we speculate that” within the sentence as per the recommendation of the reviewer.

- "During the worldwide pandemic, a delay in publication due to a slower peer review process potentially delays the dissemination of information against the COVID-19 pandemic.: - this read a bit awkwardly.

We have revised the sentence.

-"...while permitting a traditional peer review process to take place." This could be perceived as an insinuation that peer review didn't take place in the other articles. Moreover, the RCTs cited (21, 22) were accepted very rapidly (e.g. NEJM trial: April 20 trial completed, May 22 accepted and first version published by journal).

We have removed the sentence “while permitting a traditional peer review process to take place”.

-"Furthermore, despite the initial expedition to publication to better understand the pandemic, the..." This sentence is rather long and difficult to navigate. Suggest editing.

We have revised the sentence. For all the recommendation of the paragraph modified to address reviewer 2/3 concerns, see the following paragraph below and on pages 8-9 of the manuscript.

“In the early stages of the COVID-19 pandemic, we speculate that an urgent need for scientific data to inform clinical, social and economic decisions led to shorter time to publication and explosion in publication of COVID-19 studies in both traditional peer-reviewed journals and preprint servers.^{1, 12} The accelerated scientific process in the COVID-19 pandemic allowed a rapid understanding of natural history of COVID-19 symptomology and prognosis, identification of tools including RT-PCR to diagnose SARS-CoV-2,¹³ and identification of potential therapeutic options such as tocilizumab and convalescent plasma which laid the foundation for future RCTs.^{14, 15, 16} A delay in publication of COVID-19 articles due to a slower peer review process may potentially delay dissemination of pertinent information against the pandemic. Despite concerns of slow peer review, major landmark trials (ie. RECOVERY and ACTT-1

trial)^{17, 18} published their findings in preprint servers and media releases to allow for rapid dissemination. Importantly, the data obtained in these initial studies should be revisited as stronger data emerges as lower quality studies may fundamentally risk patient safety, resource allocation, and future scientific research.¹⁹”

Reviewer #2 (Remarks to the Author):

My remaining comments refer to the abstract and Table 2 only.

Abstract

- 1. The hyphen on line 46 should be a comma.**
- 2. L52, "1:1 historical control" is not logical. I suggest "Each clinical paper evaluating COVID-19 was matched 1:1 to a historical control of the same study design in the same journal from the previous year."**
- 3. L56-58 wrongly implies that every COVID-19 article had a mean case series quality score (etc.). I suggest "COVID-19 articles reporting case series had a mean quality score (out of 5) (SD) of 3.3 (1.1); cohort studies, mean NOS score (out of 8) of 5.8 (1.5); case control studies, mean NOS score (out of 8) of 5.5 (1.9); and diagnostic studies, low risk of bias present in 4 (6.4%)." By the way "±" is discouraged and unnecessary.**

We thank the reviewer for the suggestions in revising the abstract. We had to revise the abstract greatly to meet the word requirement of the journal and can be found below and on page 3 of the manuscript.

“The COVID-19 pandemic began in early 2020 with major health consequences. While a need to disseminate information to the medical community and general public was paramount - concerns have been raised regarding the scientific rigor in published reports. We performed a systematic review to evaluate the methodological quality of currently available COVID-19 studies compared to historical controls. A total of 9895 titles and abstracts were screened, and 686 COVID-19 articles were included in the final analysis. Comparative analysis of COVID-19 to historical articles reveals a shorter time to acceptance (13.0[IQR, 5.0-25.0] days vs. 110.0[IQR, 71.0-156.0] days in COVID-19 and control articles, respectively; $p < 0.0001$). Furthermore, methodological quality scores were lower in COVID-19 articles in all study designs. COVID-19 clinical studies had a shorter time to publication and had lower methodological quality scores than control studies in the same journal. These studies should be revisited with the emergence of stronger evidence.”

Table 2

- 3. "All" is unclear in the heading, need to say it's "all COVID-19": it would be best to use exactly the same headings as in table 1.**

Thank you, this has been revised.

- 4. The use of "no. (%)" with "mean (SD)" beneath it is of minimally acceptable quality. It would be much clearer if the authors just put % as appropriate e.g. in row 2 change (45.3) to (45.3%).**

We have revised the table to reflect the recommended changes.

- 5. Percentages should be rounded to the nearest 1%, because they don't have even this level of accuracy (due to small samples): so the result above would be best as (45%).**

We have revised the table to reflect the recommended changes.

- 6. "Case-series design" can just be "Case series". Later also "Case-control study", "Cohort study", "Diagnostic study" (why plural?), "Randomised controlled trial".**

We have revised the table to reflect the recommended changes.

- 7. In both tables 1 and 2, please make clear what the p-value represents e.g. "P-value, COVID-19 vs Control".**

We have added a statement in the legend representing the p-value from comparison of COVID-19 and control articles.

- 8. "n (%)" isn't needed & could be confusing in the subheadings under "Diagnostic studies - no. (%)". Please remove it.**

We have revised the table to reflect the recommended changes.

Reviewer #3 (Remarks to the Author):

While I retain the majority of my previous concerns regarding the use of the Newcastle-Ottawa scale, I have found the language changes to be sufficient for publication. The Newcastle-Ottawa scale is not justifiable just because it is more common and faster. We cannot and should not excuse weaker methods just because they are common and fast does not improve methodological rigor, as is clearly true for COVID-19 studies. I would strongly have preferred use of a more sensible scale, and lacking that a much more details accounting of the limitations of the scales used.

However, the revised manuscript no longer refers to studies as being high quality, and focuses almost entirely on the differences in scores between COVID and non-COVID research. Comparing differences in scores, rather than assigning meaning to the scores themselves, is more broadly acceptable in this regard. As the paper is now written and framed, it is no longer an existential threat to the validity of the claims being made.

I have generally found the responses to the reviewers to be well-written, and the changes made appropriate. I would also like to strongly disagree with Reviewer #2's response to 3a, which appears to be largely based in a philosophical stance regarding what is acceptable about evidence, and not a reflection of the findings of this research. R2 appears to be taking the stance that there is no use to this research because it is acceptable to have weaker methods in emerging situations (personally, I believe the opposite to be true, where we should increase the standards of our methodological rigor when the decisions are more important, not decrease). Regardless of that preference, it is critical to document the strength of that research so that decision-makers are better able to determine whether they are useful for decision-making. I strongly recommend the editors and authors disregard that comment.

We thank the reviewer for their kind words and strengthening the overall manuscript. It is our hope that the manuscript will generate as much vigorous discussion and debate as it did amongst the reviewers. We have revised the discussion paragraph as per Reviewers 1-3 recommendations and hope that we can portray both sides of the debate well in the discussion.